# GLOBAL CONVERGENCE OF FOUR-LAYER MATRIX FACTORIZATION UNDER RANDOM INITIALIZATION

## ABSTRACT

Gradient descent dynamics on the deep matrix factorization problem is extensively studied as a simplified theoretical model for deep neural networks. Although the convergence theory for two-layer matrix factorization is well-established, no global convergence guarantee for general deep matrix factorization under random initialization has been established to date. To address this gap, we provide a polynomial-time global convergence guarantee for randomly initialized gradient descent on four-layer matrix factorization, given certain conditions on the target matrix and a standard balanced regularization term. Our analysis employs new techniques to show saddle-avoidance properties of gradient decent dynamics, and extends previous theories to characterize the change in eigenvalues of layer weights.

## 1 INTRODUCTION

This paper investigates matrix factorization, a fundamental non-convex optimization problem, which in its canonical form seeks to optimize the following objective:

$$\mathcal{L}(W_1, \ldots, W_N) := \frac{1}{2} \|W_N \cdots W_1 - \Sigma\|_F^2 + \mathcal{L}_{\text{reg}}(W_1, \ldots, W_N), \tag{1}$$

where $W_j \in \mathbb{F}^{d \times d}$ denotes the $j^{\text{th}}$ layer weight matrix, $\Sigma \in \mathbb{F}^{d \times d}$ denotes the target matrix and $\mathcal{L}_{\text{reg}}$ is a (optional) regularizer, $d \in \mathbb{N}^*$ is the size of matrices which can be arbitrary positive integers (for $d = 1$ it reduces to scalars). Here $\mathbb{F} \in \{\mathbb{C}, \mathbb{R}\}$ as we consider both real and complex matrices in this paper. Following a long line of works (Arora et al., 2019a; Jiang et al., 2023; Ye & Du, 2021; Chou et al., 2024), we aim to understand the dynamics of gradient descent (GD) on this problem:

$$j = 1, \ldots, N : W_j(t+1) = W_j(t) - \eta \nabla_{W_j} \mathcal{L}(W_1(t), \ldots, W_N(t)), \tag{2}$$

where $\eta \in \mathbb{R}^+$ is the learning rate.

While global convergence guarantee for the case of two-layer matrix factorization ($N = 2$) is well studied (Du et al., 2018; Ye & Du, 2021; Jiang et al., 2023), the deep matrix factorization problem, *i.e.*, the $N > 2$ case is less explored. While the model representation power is independent of depth $N$, the deep matrix factorization problem is naturally motivated by the goal of understanding benefits of depth in deep learning (see, *e.g.*, Arora et al. (2019b)). A long line of previous works (Hardt & Ma, 2016; Arora et al., 2019b;a; Wang & Jacot, 2023) studies this regime as it directly captures Deep Linear Networks (DLN), the simplest type of deep neural networks. However, a general global convergence guarantee is still missing. Therefore, the following open research question can be naturally asked:

*Can we prove global convergence of GD for matrix factorization problem (1) with $N > 2$ layers?*

In this paper, we take a positive step towards answering the question above. Specifically, we consider 4-layer matrix factorization ($N = 4$) with the standard balancing regularization term (see Park et al. (2017); Ge et al. (2017); Zheng & Lafferty (2016)) as

$$\mathcal{L}(W_1, W_2, W_3, W_4) := \frac{1}{2} \|W_4 W_3 W_2 W_1 - \Sigma\|_F^2 + \frac{1}{4} a \left( \sum_{j=1}^{3} \left\| W_j W_j^H - W_{j+1}^H W_{j+1} \right\|_F^2 \right),$$

where $W_j^H$ denotes the Hermitian transpose of $W_j$ and $a \in \mathbb{R}^+$ is a hyperparameter. We consider both real ($\mathbb{F} = \mathbb{R}$) and complex ($\mathbb{F} = \mathbb{C}$) setting with random Gaussian initialization and prove global convergence of gradient descent. Our main result can be summarized as follows:

**Theorem 1** (Main theorem, informal). *Consider four-layer matrix factorization for target matrix $\Sigma$ with identical singular values $\sigma_1 > 0$, under gradient descent and random Gaussian initialization with small scaling factor $\epsilon \ll \sigma_1^{1/4}$, then with sufficient small learning rate $\eta$ and large regularization factor $a$, (1) with high probability $1 - \delta$ over the complex initialization and complex $\Sigma$, or (2) with probability $\frac{1}{2}(1 - \delta)$ over the real initialization and real $\Sigma$, loss function $\mathcal{L}(t) \leq \epsilon_{\mathrm{conv}}$ for*

$$t > T(\epsilon_{\mathrm{conv}}, \eta) = \eta^{-1}\sigma_1^{-1}\epsilon^{-2}\mathrm{poly}\left(1/\delta, d\right) + O\left(\eta^{-1}\sigma_1^{-3/2}\ln\left(d\sigma_1^2/\epsilon_{\mathrm{conv}}\right)\right), \textit{for any } \epsilon_{\mathrm{conv}} > 0.$$

The formal version of Theorem 1 is stated in Theorem 47 in Appendix, where we specify the polynomial degrees for $\epsilon, a, \eta, T(\epsilon_{\mathrm{conv}}, \eta)$. Below we provide a simple example to illustrate the result.

**Example for tightness.** We show the convergence rate is nearly tight by the toy example of $d = 1$, where all the weight matrices degenerate into scalars. Consider identical initialization $w_{j:j\in[4]}(t = 0) = \epsilon$ and gradient flow, then all $w_j$ remain identical and the dynamics become $\frac{\mathrm{d}w_j}{\mathrm{d}t} = (\sigma_1 - w_j^4)w_j^3$. By solving the differential equation, it takes time $\Theta\left(\sigma_1^{-1}\epsilon^{-2}\right)$ for product weight $w := w_4 w_3 w_2 w_1$ to increase from $\epsilon^4$ to $\Theta(\sigma_1)$, then time $\Theta\left(\sigma_1^{-3/2}\ln\left(\sigma_1^2/\epsilon_{\mathrm{conv}}\right)\right)$ to reach local convergence. Theorem 1 exactly reduces to this result when the dimension $d = 1$. Calculation details are provided in Appendix J.1.

For further explanation on the exponents of $\sigma_1$ in $\epsilon$ and $T(\epsilon_{\mathrm{conv}}, \eta)$, please refer to Appendix J.2.

**Remark 1.** *A natural question is why the convergence guarantee in the real case holds only with probability close to $\frac{1}{2}$, but not 1. For the other $\frac{1}{2}$ probability, Theorem 2 presents a special case - considering gradient flow under the strict balance condition (which can be viewed as the limit as $a \to +\infty$), showing that the optimization process does not converge to a global minimum in finite time (and hence converges to a saddle point).*

**Main contributions.** Our major contributions can summarized as follows:

- We prove global convergence of GD for 4-layer matrix factorization under random Gaussian initialization. To the best of our knowledge, this is the first global convergence result for general deep linear networks under random initialization beyond the NTK regime in Du & Hu (2019). This result helps provide new insights towards understanding the training dynamics of general deep neural networks.

- We construct a novel three-stage convergence analysis of gradient descent dynamics, consisting of an alignment stage, a saddle-avoidance stage, and a local convergence stage. We also develop new techniques to show GD dynamics avoids saddle points and to characterize layer matrix eigenvalue changes, which we believe are of independent interest for deep linear networks analysis.

**Challenges and techniques.** Our analysis employs the following key techniques:

- Initialization analysis. To guarantee that gradient descent makes progress, it is necessary to establish a monotonically increasing lower bound for the singular values of the weight matrices. This, in turn, requires analyzing the smallest singular value of a newly introduced term (namely $W + (WW^H)^{1/2}$, where $W = W_4 W_3 W_2 W_1$), at initialization. This analysis utilizes tools from random matrix theory, particularly the concept of Circular Ensembles. The detailed proof is given in Appendix C.

- Regularity condition of each layer. To bridge the initialization with the subsequent training dynamics, we need to ensure that key matrix properties evolve in a controlled manner even during the rapid changes in the alignment stage. We prove that despite significant updates, the weight matrices retain certain spectral properties from their initial state. A delicate analysis of the smooth evolution of the extreme singular values and the behavior of the Hermitian term after the regularization term converges is provided in Section 5.2.1 and 5.2.2.

- Saddle avoidance. To avoid convergence to a saddle point, it is essential to prevent the smallest singular values of the weight matrices from decaying to zero, as such decay would cause the gradient norm to vanish. To this end, we construct a hermitian term providing lower-bounds for these singular values, along with a skew-hermitian error. During the optimization, the skew-hermitian error is approximately non-increasing, which in turn ensures that the minimum singular value of the hermitian term is non-decreasing. This mechanism provides a persistent lower bound, thereby effectively avoiding saddle points.

- Bound of eigenvalue change. Finally, to translate the continuous-time intuition into rigorous guarantees for the discrete gradient descent algorithm, we develop new perturbation bounds for eigenvalues. In continuous time, the time derivatives of eigenvalues are directly characterized by the derivatives of the matrix. In discrete time, however, eigenvalue changes depend on the spectral gap in general, requiring a fine-grained, problem-specific analysis. Similar challenge are noted in Lemma 3.2 of Ye & Du (2021). We address this issue in Lemma 19 and 20 in Appendix D.2.

These techniques form a cohesive proof strategy: the initialization analysis provides a favorable starting point; the regularity analysis ensures controlled dynamics throughout training; the saddle avoidance mechanism guarantees persistent progress; and the discrete-time perturbation bounds rigorously translate these insights into a full global convergence proof.

**Paper Roadmap.** Section 3 introduces basic notations. To provide a intuitive framework of the convergence analysis, we first establish the result under a special initialization (namely balanced Gaussian initialization) and gradient flow in Section 4, then generalize the proof strategy into general random Gaussian initialization and gradient descent in Section 5, which consists of three stages. Some of our supporting theorems can be applied to more general setting of target matrix $\Sigma$ and depth $N$, where we specify in Table 1 below (identical means the singular values are the same):

| Theorem | Initialization | Depth $N$ | Target |
|---|---|---|---|
| Thm 3: balanced Gaussian initialization | balanced Gaussian | $\geq 2$ | - |
| Thm 6: random Gaussian initialization | random Gaussian | $\geq 2$ | - |
| Thm 4: bounded skew-Hermitian error | balanced Gaussian | $\geq 2$ | arbitrary |
| Thm 5: increasing rate of main term | balanced Gaussian | $4$ | identical |
| Thm 7: convergence rate of regularization term $\mathcal{L}_{\mathrm{reg}}$ | - | $4^{1}$ | arbitrary |
| Thm 8: max/min singular value changes under $\mathcal{L}_{\mathrm{reg}}$ | - | $\geq 2$ | arbitrary |

Table 1: Summary of the supporting theorems and their assumptions.

## 2 RELATED WORKS

For two-layer matrix factorization, the global convergence of symmetric case has been established under various settings (Jain et al., 2017; Li et al., 2019; Chen et al., 2019). For asymmetric matrix factorization case with objective $\mathcal{L} = \frac{1}{2}\|UV^\top - \Sigma\|_F^2$, the following homogeneity issue occurs: the prediction result remains the same if one layer is multiplied by a positive constant while the other is divided by the same, introducing significant challenges in convergence analyzing (Lee et al. (2016), Proposition 4.11). Tu et al. (2016) and Ge et al. (2017) tackles this problem by manually adding a regularization term on the objective function. Du et al. (2018) discovers that gradient descent automatically balances the magnitudes of layers under small initialization, providing analysis of global convergence with polynomial time under decayed learning rate, while removing the regularization term. Ye & Du (2021) extends the convergence analysis to constant learning rate. Wang et al. (2022) demonstrates the convergence for constant large learning rates and exhibits that the optimization converges to a approximately balanced optimum. Xu et al. (2024) adopts an unbalanced initialization, under which they proved that NAG achieves an accelerated convergence rate.

Kawaguchi (2016) analyzes landscape for general DLN, showing there exists saddle points with no negative eigenvalues of Hessian for depth over three. Bartlett et al. (2018) analyzes the dynamic under identity initialization, proving polynomial convergence with target matrix near initialization

---

[1]This can be generalized to arbitrary $N \geq 2$. An arbitrary $N$ version for gradient flow is provided in Theorem 28 in the Appendix.

or symmetric positive definite, but such initialization fails to converge when target matrix is symmetric and has a negative eigenvalue. Arora et al. (2019a) provides global convergence proof under specific deep linear neural network structures and initialization scheme, requiring the initial loss to be smaller than the loss of any rank-deficient solution. Ji & Telgarsky (2019) conducted the proof of convergence on general deep neural networks with similar requirements on the initial loss. Arora et al. (2019b) simplifies the training dynamics of deep linear neural network into the dynamic of singular values and singular vectors of product matrix under balanced initialization, providing theoretical illustration of local convergence when singular vectors are stationary. Nguegnang et al. (2024) proves that for general depth linear networks, under appropriate gradient scheduling and initialization the optimization converges to a critical point. Du & Hu (2019) proves global convergence for wide linear networks under the neural tangent kernel (NTK) regime. More recent works focus on GD dynamics under (approximately) balanced initialization schemes (Min et al., 2023) or the 2-layer case (Min et al., 2021; Xiong et al., 2023; Tarmoun et al., 2021). Chizat et al. (2024) studies the infinite-width limit of DLN in the mean field regime. However, none of these results imply a global convergence guarantee for general DLN with $N > 2$ under random initialization.

## 3 PRELIMINARIES

**Notation.** Denote the complex conjugate of $M$ as $\bar{M}$ and adjoint of $M$ as $M^H$, $\mathbb{N}$ as the set of non-negative integers, and $\mathbb{N}^*$ as the set of positive integers. $\sigma_k(\cdot)$ denotes the $k^{th}$ largest singular value of the matrix. For $k_1 < k_2 \in \mathbb{N}$, $\prod_{j=k_2}^{k_1} M_j = M_{k_2} M_{k_2-1} \cdots M_{k_1}$. $x \sim \mathcal{N}(0,1)_{\mathbb{C}}$ means that the real and imaginary parts are independently sampled from Gaussian distribution with variance $\frac{1}{2}$: $\Re x, \Im x \overset{\text{i.i.d.}}{\sim} \mathcal{N}(0, 1/2)$. $Q \sim U(d, \mathbb{C})$ or $O(d, \mathbb{R})$ means $Q$ is drawn from the unique uniform distribution (Haar measure) on the unitary or orthogonal group, implying its distribution is unitarily/orthogonally invariant.

Consider general $N$-layer matrix factorization, for simplicity we define the following notations:

$$W_{\prod_L, j} := \prod_{k=N}^{j} W_k, \; W_{\prod_R, j} := \prod_{k=j}^{1} W_k, \; W := \prod_{k=N}^{1} W_k = W_{\prod_L, 1} = W_{\prod_R, N}, \tag{3}$$

$$\Delta_{j,j+1} := \begin{cases} W_j W_j^H - W_{j+1}^H W_{j+1} & , j \in \{1, 2, \cdots, N-1\} \\ O^{d \times d} & , j \in \{0, N\} \end{cases}. \tag{4}$$

$W$ is referred to as *product matrix*. The loss is written by $\mathcal{L}(W_1, \cdots, W_N) = \mathcal{L}_{\text{ori}} + \mathcal{L}_{\text{reg}}$, where $\mathcal{L}_{\text{ori}} = \frac{1}{2} \|\Sigma - W\|_F^2$, $\mathcal{L}_{\text{reg}} = \frac{1}{4} a \left( \sum_{j=1}^{N-1} \|\Delta_{j,j+1}\|_F^2 \right)$.

**Algorithmic setup.** For the real case ($W_j \in \mathbb{R}^{d \times d}$), GD dynamics is canonical and described by equation 2. Under complex field ($W_j \in \mathbb{C}^{d \times d}$), for simplicity and coherence we define $\nabla_M = \frac{\partial}{\partial \Re M} + i \frac{\partial}{\partial \Im M}$, which is two times of Wirtinger derivative with $\bar{M}$: $\frac{\partial}{\partial \bar{M}} = \frac{1}{2} \left( \frac{\partial}{\partial \Re M} + i \frac{\partial}{\partial \Im M} \right)$. By following the updating rule of complex neural networks (see Guberman (2016)), the gradient can be uniformly represented by

$$\begin{aligned} \nabla_{W_j} \mathcal{L} &= \nabla_{W_j} \mathcal{L}_{\text{ori}} + \nabla_{W_j} \mathcal{L}_{\text{reg}} \\ \nabla_{W_j} \mathcal{L}_{\text{ori}} &= -W_{\prod_L, j+1}^H (\Sigma - W) W_{\prod_R, j-1}^H, \; \nabla_{W_j} \mathcal{L}_{\text{reg}} = -a W_j \Delta_{j-1,j} + a \Delta_{j,j+1} W_j, \end{aligned} \tag{5}$$

Under gradient flow, $\frac{dW_j}{dt} = -\nabla_{W_j} \mathcal{L}$; under gradient descent, $W_j(t+1) = W_j(t) - \eta \nabla_{W_j} \mathcal{L}(t)$.

**Reduction to diagonal target.** Following the simplification process of Section 2.1 in Ye & Du (2021), suppose the singular value decomposition of $\Sigma$ is $\Sigma = U_\Sigma \Sigma' V_\Sigma^H$, by applying the following transformation $W_1 \leftarrow W_1 V_\Sigma$ and $W_N \leftarrow U_\Sigma^H W_N$, the dynamics remain the same form, while the distributions of $W_j$ under our initialization schemes remain the same. Hence without loss of generality, we assume the target matrix is *diagonal with real and non-negative entries* throughout our analysis. Detailed analysis is presented in Appendix B.

For some of the results, we further require target matrix to be *an identity matrix scaled by a positive constant* $\Sigma = \sigma_1(\Sigma)I$, which is equivalent to *requiring that the singular values of target matrix are identical*.

**Balancedness.** Following a long line of works (Arora et al., 2019a;b; Du et al., 2018), we define the balance difference between layer $j$ and $j+1$ as $\Delta_{j,j+1}$ (refer to 4). As discussed in Definition 1 of Arora et al. (2019a), the weights are approximately balanced (namely $\|\Delta_{j,j+1}\|_F$ are small) throughout the iterations of gradient descent under approximate balancedness at initialization and small learning rate. Notice that approximate balancedness holds for small initialization near origin (small variance for Gaussian initialization).

Specifically, under *gradient flow* the balanced condition (defined as $\|\Delta_{j,j+1}(t)\|_F \equiv 0$ or equivalently $\Delta_{j,j+1}(t) \equiv O, \forall j \in \{1, 2, \cdots, N-1\}$) *holds strictly at arbitrary time under balanced initialization*, which is defined as $\Delta_{j,j+1}(t=0) \equiv O, \forall j \in \{1, 2, \cdots, N-1\}$.

**Remark 2.** *As previously discussed, balance condition holds approximately under small initialization, so such regularization's affect on the training process is relatively weak, especially when weight matrices grow larger and be away from origin.*

# 4 TRAINING DYNAMICS UNDER BALANCED GAUSSIAN INITIALIZATION

We denote the initialization satisfying strict balancedness as balanced initialization. Generally, strict balancedness yields a clean form of dynamics, where the dynamic of product matrix $W$ depends on $W$ itself solely and is irrelevant to layers $W_{1,2,\cdots,N}$ (Arora et al., 2019b). However, random Gaussian initialization does not satisfy strict balancedness. To adapt the random Gaussian initialization to ensure balanced condition, we introduce a *balanced Gaussian initialization* scheme for the analysis below. The procedure is defined as follows:

(1) Sample $G$ with entries $G_{ij} \overset{\text{i.i.d.}}{\sim} \mathcal{N}(0,1)_{\mathbb{F}}$, $Q_{k,k+1;k\in\{0,1,\cdots,N\}} \overset{\text{i.i.d.}}{\sim}$ Haar on $U(d,\mathbb{C})$ for $\mathbb{F} = \mathbb{C}$ (or $O(d,\mathbb{R})$ for $\mathbb{F} = \mathbb{R}$). $s_{j,j\in\{1,2,\cdots,N\}} \in \mathbb{F}$ are arbitrary constants with modulus/absolute value 1.

(2) For scaling factor $\epsilon \in \mathbb{R}^+$, which is a small positive constant, set the weight matrices by:

$$W_j = \begin{cases} s_j \epsilon Q_{j,j+1} G Q_{j-1,j}^H & , 2 \nmid j \\ s_j \epsilon Q_{j,j+1} G^H Q_{j-1,j}^H & , 2 \mid j \end{cases} . \tag{6}$$

Intuitively, $Q_{k,k+1;k\in\{0,1,\cdots,N\}}$ are i.i.d. uniformly distributed unitary/orthogonal matrices. By Corollary 13 in the Appendix, each matrix is a $\epsilon$-scaled Gaussian random matrix ensemble (but not independent of the others), while satisfying balanced condition $\Delta_{j,j+1}(0) = O, \forall j \in \{1, 2, \cdots, N-1\}$.

To exhibit the convergence dynamics clearly, we present the global convergence under the simplified scenario of balanced Gaussian initialization and gradient flow. Notice that the adjacent matrices remain balanced due to the non-increasing property of regularization term (Lemma 26).

**Theorem 2.** *(Informal) Global convergence bound under balanced Gaussian initialization, gradient flow. For four-layer matrix factorization under gradient flow, balanced Gaussian initialization with scaling factor $\epsilon \leq \sigma_1^{1/4}(\Sigma)/\text{poly}(1/\delta, d)$, then for target matrix with identical singular values,*

*1. For $\mathbb{F} = \mathbb{R}$, with probability at least $\frac{1}{2}$ the loss does not converge to zero.*

*2. For $\mathbb{F} = \mathbb{C}$ with high probability at least $1 - \delta$ and for $\mathbb{F} = \mathbb{R}$ with probability at least $\frac{1}{2}(1 - \delta)$, there exists $T(\epsilon_{\text{conv}}) = \sigma_1^{-1}\epsilon^{-2}\text{poly}(1/\delta, d) + O\left(\sigma_1^{-3/2} \ln\left(d\sigma_1^2/\epsilon_{\text{conv}}\right)\right)$, such that for any $\epsilon_{\text{conv}} > 0$, when $t > T(\epsilon_{\text{conv}})$, $\mathcal{L}(t) < \epsilon_{\text{conv}}$.*

The formal version is stated in Theorem 35 in the Appendix, where we specify the polynomial degrees of $\epsilon$ and $T(\epsilon_{\text{conv},\eta})$.

## 4.1 BALANCED GAUSSIAN INITIALIZATION

This section establishes the properties for balanced Gaussian initialization.

**Theorem 3.** *Under $\epsilon$-scaled balanced Gaussian initialization, suppose $W$ is $W = U\Sigma_w^N V^H$, where $U$, $V$ are unitary/orthogonal matrices, $\Sigma_w$ is positive semi-definite and diagonal, denote $s := \prod_{j=1}^N s_j$, then for some $f_1 = O\left(\frac{1}{\delta}\right)$, $f_2' = O\left(\frac{1}{\delta^2}\right)$:*

*1. If $\mathbb{F} = \mathbb{C}$, at the initialization the following inequalities hold with probability at least $1 - \delta$:*

$$\|\Sigma_w\|_{op} \leq f_1(\delta)\sqrt{d}\epsilon, \ \|(U - V)\Sigma_w\|_F|_{t=0} \leq 2f_1(\delta)d\epsilon$$
$$\sigma_{\min}((U + V)\Sigma_w)|_{t=0} \geq f_2'(\delta)^{-1} d^{-3/2}\epsilon. \tag{7}$$

*2. If $\mathbb{F} = \mathbb{R}$, at the initialization we have $\Pr(s\det(Q_{N,N+1})\det(Q_{01}) = 1) = \Pr(s\det(Q_{N,N+1})\det(Q_{01}) = -1) = \frac{1}{2}$. If the initialization satisfies $s\det(Q_{N,N+1})\det(Q_{01}) = -1$, then $\sigma_{\min}((U + V)\Sigma_w)|_{t=0}$; otherwise $s\det(Q_{N,N+1})\det(Q_{01}) = 1$, then the following inequalities hold with probability at least $1 - \delta$:*

$$\|\Sigma_w\|_{op} \leq f_1(\delta)\sqrt{d}\epsilon, \ \|(U - V)\Sigma_w\|_F|_{t=0} \leq 2f_1(\delta)d\epsilon$$
$$\sigma_{\min}((U + V)\Sigma_w)|_{t=0} \geq f_2'(\delta)^{-1} d^{-3/2}\epsilon. \tag{8}$$

Proof is presented in Appendix C.3. One may question the motivation of analyzing $\sigma_{\min}((U + V)\Sigma_w)|_{t=0}$. We later show that this term acts as a crucial lower bound with a relatively simple dynamics in Section 4.3.

### 4.2 Non-increasing Skew-Hermitian Error

As presented in Lemma 24 in the Appendix, the product matrix can be factorized in to the form of $W(t) = U(t)\Sigma_w(t)^N V(t)^H$, where $\Sigma_w(t)$ is positive semi-definite and diagonal (consequently real-valued), $U$ and $V$ are unitary/orthogonal matrices, $U$, $V$ and $\Sigma_w$ are analytic. For simplicity, we denote $\sigma_{w,j}$ as the $j^{th}$ diagonal entry of $\Sigma_w$, and $u_j$, $v_j$ as the $j^{th}$ column of $U$, $V$. Under this representation of product matrix, we obtain a *non-increasing Skew-Hermitian/Symmetric term*:

**Theorem 4.** *(Informal) Skew-Hermitian error term is non-increasing. Under balanced initialization with product matrix $W(t) = U(t)\Sigma_w(t)^N V(t)^H$, for depth $N \geq 2$, if the singular values of the product matrix at initial $W(0)$ are non-zero and distinct, then the following skew-Hermitian error $\left\|\Sigma^{1/2}(U - V)\Sigma_w\right\|_F^2$ is non-increasing:*

$$\frac{d}{dt}\left\|\Sigma^{1/2}(U - V)\Sigma_w\right\|_F^2 \leq 0. \tag{9}$$

*Proof sketch. Proof of the Theorem 4 involves technical and lengthy calculations. The formal version is stated in Theorem 31, while a special version for even $N$ is separately discussed in Theorem 32. For the proof of Theorem 31, the idea is to decompose the derivative of this term into the derivative of $\sigma_{w,j}$ and $u_j$, $v_j$, which have been characterized by Theorem 3 and Lemma 2 in Arora et al. (2019b) respectively. This method is hard to generalize into imbalanced setting. For Theorem 32, this term is directly derived from derivative of $W_N W_N^H$, $W_1^H W_1$ and $W$. This approach is straight forward and can be extended to imbalanced initialization, but encounters difficulty under odd depth $2 \nmid N$.*

**Remark 3.** *This result is established under the reduction to target matrix (refer to Section 3 and Appendix B). For general target matrix, suppose its SVD is $\Sigma = U_\Sigma \Sigma' V_\Sigma^H$, then Theorem 4 becomes:*

$$\frac{d}{dt}\left\|\Sigma'^{1/2}(U_\Sigma^H U - V_\Sigma^H V)\Sigma_w\right\|_F^2 \leq 0. \tag{10}$$

**Explanation of the result.** This theorem provides an intrinsic non-increasing term of *general deep matrix factorization*. (Under initialization close to origin, this term is already small at initial. ) Although the result is accurately derived under strictly balanced initialization and gradient flow, one may expect similar property to hold under small initialization and gradient descent.

Moreover, this theorem characterizes when $U$ and $V$ become aligned. The product matrix can be expressed as $W = \sum_{i=1}^d \sigma_{w,j}^N u_j v_j^H$, while the error can be rewritten as

$\sum_{j=1}^{d} \sigma_{w,j}^2 \left\| \Sigma^{1/2}(u_j - v_j) \right\|_F^2$. Each term $\sigma_{w,j}^N u_j v_j^H$ of the product matrix can be interpreted as *a "feature" of the linear neural network*, containing one "value" $\sigma_{w,j}^N$ and two "directions" $u_j$, $v_j$. When the loss converges, each feature converges to $\sigma_j u_{\Sigma,j} u_{\Sigma,j}^H$, where $\Sigma = \sum_{j=1}^{d} \sigma_j u_{\Sigma,j} u_{\Sigma,j}^H$ is a SVD of $\Sigma$. This shows that under initialization near origin, once a "value" of the $j^{th}$ feature *increases to a relatively large value* (comparing to initialization), the directions of this feature *automatically align with each other* (i.e. $\langle u_j, v_j \rangle \approx 1$). Followed by Theoretical illustration part of Arora et al. (2019b), Section 3, generally the alignment of $U$, $V$ leads to convergence.

As shown in the proof sketch, the analysis for odd $N$ encounters difficulty when generalized to the imbalanced case, thus this intrinsic non-increasing term becomes considerably more challenging to characterize. This is why we have developed the convergence proof for the four-layer case rather than the three-layer architecture.

### 4.3 Non-Decreasing Hermitian Main Term

This section shows the dynamics of the minimum singular value of Hermitian main term $(U+V)\Sigma_w$.

The motivation of studying this specific term is that it provides both lower and upper bounds for $\sigma_k(\Sigma_w)$, $k \in \{1, 2, \cdots, N-1\}$, especially tight bounds for $\sigma_{\min}(\Sigma_w)$ (refer to Lemma 18):

$$
\begin{aligned}
\frac{1}{2}\sigma_k\left((U+V)\Sigma_w\right) &\leq \sigma_k(\Sigma_w) \leq \frac{\sqrt{2}}{2}\sqrt{\sigma_k^2\left((U+V)\Sigma_w\right) + \left\|(U-V)\Sigma_w\right\|_{op}^2} \\
\frac{1}{2}\sigma_{\min}\left((U+V)\Sigma_w\right) &\leq \sigma_{\min}(\Sigma_w) \leq \frac{1}{2}\sqrt{\sigma_{\min}^2\left((U+V)\Sigma_w\right) + \left\|(U-V)\Sigma_w\right\|_{op}^2}.
\end{aligned}
\tag{11}
$$

Notice that the extra term in the upper bound is bounded by the skew-Hermitian error term discussed in the previous section.

Although the evolution of $\sigma_k((U+V)\Sigma_w)$ is difficult to characterize in general, we find that in the special case of $\Sigma = \sigma_1(\Sigma)I$ and $N = 4$, it exhibits a monotonically increasing pattern before local convergence:

**Theorem 5.** *Dynamics of minimum singular value of Hermitian term. Under balanced initialization with product matrix $W(t) = U(t)\Sigma_w(t)^N V(t)^H$, for target matrix with identical singular values (reduces to $\Sigma = \sigma_1(\Sigma)I$) and depth $N = 4$, the time derivative of the $k^{th}$ singular value of the Hermitian term $x_k := \frac{1}{2}\sigma_k((U+V)\Sigma_w)$ is bounded by:*

$$
\left(2\sigma_1(\Sigma) - x_k^4 - \frac{1}{2}\|\Sigma_w\|_{op}^2\|((U-V)\Sigma_w)|_{t=0}\|_F^2\right)x_k^4 - \frac{1}{16}x_k^2\|\Sigma_w\|_{op}^2\|((U-V)\Sigma_w)|_{t=0}\|_F^4
$$
$$
\leq \frac{\mathrm{d}}{\mathrm{d}t}x_k^2 \leq \sigma_1(\Sigma)\left(2\|\Sigma_w\|_{op}^2 + \|((U-V)\Sigma_w)|_{t=0}\|_F^2\right)x_k^2.
\tag{12}
$$

Detailed proof is presented in E.2.

**Discussion on $1/2$ failure probability.** This theorem implies that under small initialization, if all singular values $\sigma_k((U+V)\Sigma_w)$ are initially non-zero, they increase monotonically to relatively large values, leading to subsequent local convergence. However, if any singular value is initialized to zero (which occurs with probability at least $1/2$ for $\mathbb{F} = \mathbb{R}$, as shown in Theorem 3), it *remains zero throughout the optimization* (see Corollary 34), thereby explaining the $1/2$ convergence probability in Theorem 2. Numerical simulations under the identity target setting are provided in Figure 1.

**Discussion on target matrix with spectral gaps (singular values are different from each other).** We also conduct additional simulations for non-identical targets (i.e. non-zero spectral gaps) in Figure 2, which we do not cover in Theorem 5. From these results, we exhibit that while the lower bounds constructed in equation (11) still hold under general target matrix with spectral gap, they suffer from sudden change when one singular value converges, so the monotonicity in Theorem 5 does not hold anymore. More detailed discussions are presented in Appendix K.1.

**A short note on incremental learning.** Although the proof of incremental learning is beyond the scope of this work, we do have a brief theoretical explanation for this behavior exhibited in Figure 1 by exploiting Theorem 5 and equation (11). Detailed discussion is presented in the Appendix K.1.

## 5 CONVERGENCE UNDER RANDOM GAUSSIAN INITIALIZATION

This section presents the proof sketch for Theorem 1, extending our analytical framework in the previous section to accommodate random Gaussian initialization.

We divide the training dynamics into three stages: alignment stage $t \in [0, T_1)$, saddle-avoidance stage $t \in [T_1, T_1 + T_2)$, and local convergence stage $t \in [T_2, +\infty)$. Here $T_1 = \frac{1}{\eta \sigma_1(\Sigma)\epsilon^2} \cdot \text{poly}^{-1}(1/\delta, d)$, $T_2 = \frac{1}{\eta \sigma_1(\Sigma)\epsilon^2} \cdot \text{poly}(1/\delta, d)$ ($\delta$ is failure probability in Theorem 1), refer to Theorem 48 and 52 respectively. Following the method in Section 4, we then characterize the skew-Hermitian error term and Hermitian main term by $\|W_1 - W_2^{-1}W_3^H W_4^H\|_F^2$ and $\lambda_{\min}\left(\left(W_1 + W_2^{-1}W_3^H W_4^H\right)^H \left(W_1 + W_2^{-1}W_3^H W_4^H\right)\right)$ respectively.

### 5.1 RANDOM GAUSSIAN INITIALIZATION

We consider the canonical setting of random Gaussian initialization near origin:

$$(W_{1,2,\cdots,N})_{ij} \overset{\text{i.i.d.}}{\sim} \epsilon \cdot \mathcal{N}(0,1)_{\mathbb{F}}. \tag{13}$$

Specifically, we apply Gaussian distribution to generate $W_{1,2,\cdots,N} \in \mathbb{F}^{d \times d}$, $\mathbb{F} = \mathbb{R}$ or $\mathbb{C}$ elementwisely and independently. Then the initialization is scaled by a small positive constant $\epsilon \in \mathbb{R}^+$. The scale of $\epsilon$ is determined in the main convergence Theorem 1.

**Theorem 6.** *For $\epsilon$-scaled random Gaussian initialization on $W_{k,k\in\{1,2,\cdots,N\}}$ over $\mathbb{F} = \mathbb{R}$ or $\mathbb{C}$, $N \in \mathbb{N}^*$,*

*1. If $\mathbb{F} = \mathbb{C}$, at the initialization the following inequalities hold with probability at least $1 - \delta$:*

$$\max_{j,k} \sigma_k(W_j) \leq f_1(\delta, N)\sqrt{d}\epsilon, \ \min_{j,k} \sigma_k(W_j) \leq \frac{\epsilon}{f_1(\delta, N)\sqrt{d}}$$
$$\sigma_{\min}\left(W + \left(WW^H\right)^{1/2}\right) \geq f_2(\delta, N)^{-1} \cdot d^{-(N/2+1)}\epsilon^N. \tag{14}$$

*2. If $\mathbb{F} = \mathbb{R}$, at the initialization we have $\Pr(\det(W) > 0) = \Pr(\det(W) < 0) = \frac{1}{2}$. If the initialization satisfies $\det(W) < 0$, then $\sigma_{\min}\left(W + \left(WW^\top\right)^{1/2}\right) = 0$; otherwise $\det(W) > 0$, then the following inequalities hold with probability at least $1 - \delta$ (given $\det(W) > 0$):*

$$\max_{j,k} \sigma_k(W_j) \leq f_1(\delta, N)\sqrt{d}\epsilon, \ \min_{j,k} \sigma_k(W_j) \leq \frac{\epsilon}{f_1(\delta, N)\sqrt{d}}$$
$$\sigma_{\min}\left(W + \left(WW^\top\right)^{1/2}\right) \geq f_2(\delta, N)^{-1} \cdot d^{-(N/2+1)}\epsilon^N, \tag{15}$$

*where $f_1(\delta, N) = O\left(\frac{N}{\delta}\right)$, $f_2(\delta, N) = O\left(\frac{N^N}{\delta^{N+1}}\right)$.*

Proof is provided in Appendix C.2. For $N = 4$, $f_1 = O\left(\frac{1}{\delta}\right)$, $f_2 = O\left(\frac{1}{\delta^5}\right)$. The term $\sigma_{\min}(W + (WW^H)^{1/2})$ is introduced in Section 5.2.2 for the purpose of analyzing the Hermitian main term.

In the convergence proof below, we consider the initialization where (14) and (15) holds.

### 5.2 STAGE 1: ALIGNMENT STAGE

During alignment stage, the weight matrices align with each other under the convergence of the regularization term, while the Hermitian main term stays away from origin at the end of this stage.

### 5.2.1 Convergence of Regularization term:

The convergence rate of regularization term is lower bounded through the following Theorem:

**Theorem 7.** *(Informal) Convergence rate of the regularization term. For four-layer matrix factorization, suppose the maximum and minimum singular values of the weight matrices are upper and lower bounded by $\mu_{\max}$ and $\mu_{\min}$ respectively, then the regularization term decays by*

$$\mathcal{L}_{\text{reg}}(t+1) \leq \left(1 - \Omega\left(\eta a \mu_{\min}^4 \mu_{\max}^{-2}\right)\right) \cdot \mathcal{L}_{\text{reg}}(t) + O(\eta^2 a^2). \tag{16}$$

The formal version can be found in Theorem 29. A $N$-layer version of this Theorem under gradient flow is provided in Theorem 27.

We can observe that the convergence rate of the regularization term is related to the extreme singular values of weight matrices, which motivates the following Theorem:

**Theorem 8.** *(Informal) Under a small learning rate, the changes in the maximum and minimum singular values are approximately independent of the regularization term:*

$$\max_{j,k} \sigma_k^2(W_j(t+1)) - \max_{j,k} \sigma_k^2(W_j(t)) \leq 2\eta \max_{j,k} \sigma_k(W_j(t)) \max_j \left\|\nabla_{W_j}\mathcal{L}_{\text{ori}}(t)\right\|_{op} + O(\eta^2 a^2)$$

$$\min_{j,k} \sigma_k^2(W_j(t+1)) - \min_{j,k} \sigma_k^2(W_j(t)) \geq -2\eta \min_{j,k} \sigma_k(W_j(t)) \max_j \left\|\nabla_{W_j}\mathcal{L}_{\text{ori}}(t)\right\|_{op} + O(\eta^2 a^2). \tag{17}$$

The complete formal statement can be found in Theorem 30 (and Theorem 28 for the continuous-time case) in the Appendix.

**Remark 4.** *This Theorem ensures the smooth change of the extreme singular values over short time intervals. Although the regularization term can induce significant fluctuations in individual singular values due to its potentially large coefficient, the largest and smallest singular values remain stable. This theoretical conclusion is corroborated by numerical simulations, as shown in Figure 5.*

### 5.2.2 The Behavior of the Hermitian main term at the end of Alignment stage

Typically, the dynamics of the smallest singular value of the Hermitian main term $W_1 + W_2^{-1}W_3^H W_4^H$ is involved and does not obtain a non-trivial lower bound during this stage. However its behavior at the end of alignment stage can be characterized by $W(0) + (W(0)W(0)^H)^{1/2}$:

The Hermitian main term can be written by $(W_1 + W_2^{-1}W_3^H W_4^H)\big|_{t=T_1} = \left(W_2^{-1}W_3^{-1}W_4^{-1}\right)\big|_{t=T_1} \cdot (W + W_4W_3W_3^H W_4^H)\big|_{t=T_1}$. At $t = T_1$, $W_4W_3W_3^H W_4^H \approx (WW^H)^{1/2}$ due to the approximate balancedness. During the alignment stage, the product remains approximately unchanged: $W(t = T_1) \approx W(t = 0)$. For the singular values of $W_{2,3,4}^{-1}$, at $t = 0$ they are bounded through Theorem 6, then Theorem 8 ensures the changes during the alignment stage are small. Together we obtain a lower bound for $\sigma_{\min}\left(W_1 + W_2^{-1}W_3^H W_4^H\right)\big|_{t=T_1}$. Detailed analysis is presented in Corollary 51.

**Remark 5.** *Note that $\sigma_{\min}\left(W_1 + W_2^{-1}W_3^H W_4^H\right)$ is not necessarily lower-bounded by the above expression minus some error terms during the alignment stage. Instead, it may exhibit oscillations or a transient decrease, achieving stability only upon convergence of the regularization term. This behavior is illustrated in Figure 6 in the Appendix.*

### 5.3 Stage 2: Saddle Avoidance stage

After alignment stage, the Hermitian main term is guaranteed to be away from zero while the skew-Hermitian error is upper bounded. During the saddle avoidance stage $t \in [T_1, T_1 + T_2)$, the Hermitian main term $\sigma_{\min}\left(W_1 + W_2^{-1}W_3^H W_4^H\right)$ increases to at least $2^{3/4}\sigma_1^{1/4}(\Sigma)$, while the skew-Hermitian error is upper bounded by $O(1) \cdot \left\|W_1 - W_2^{-1}W_3^H W_4^H\right\|_F (t = T_1)$. Former statements are presented in Lemma 57 and 56 respectively.

Intuitively, these results generalize Theorem 5 and 4 into imbalanced case respectively by bounding the error terms introduced by imbalancedness. To adapt these results into discrete time, new perturbation bound for eigenvalues is discussed in Lemma 19. Another technical challenge is to bound

the operator norm of the inverse of $W_2$ below infinity. Under small balance difference (equivalently small regularization term) which is guaranteed by the previous stage, this is rigorously proved in Lemma 55.

### 5.4 Stage 3: Local Convergence Stage

In the local convergence stage, both the balanced error and skew-Hermitian error remain small, the minimal singular values of the weight matrices, after growing to the scale of the target matrix's, are prevented from decaying. This guarantees the local convergence.

**Theorem 9.** *(Informal) Local convergence. After the second stage ($t \geq T_1 + T_2$),*

$$\mathcal{L}(t) \leq \mathcal{L}_{\text{ori}}(T_1 + T_2) \exp\left(-\eta\sigma_1^{3/2}(\Sigma)(t - T_1 - T_2)\right). \tag{18}$$

Proof is presented in I.3 in the Appendix.

## 6 Conclusions, Limitations and Future work

In this work, we establish a polynomial-time global convergence guarantee for gradient descent applied to four-layer matrix decomposition, under the setting of a target matrix with identical singular values and small random Gaussian initialization beyond the NTK regime. For complex random Gaussian initialization, global convergence is ensured with high probability, whereas for real random Gaussian initialization, it is guaranteed with a probability close to $\frac{1}{2}$.

The analysis developed in this work reveals intrinsic properties of the training dynamics, such as the effective behavior of the regularization term, the monotonically increasing lower bound for the minimum singular value, and the non-increasing nature of the skew-Hermitian error. These findings might provide deeper insight into the training process of Deep Linear Networks. Some of our results are directly generalizable to arbitrary depth $N \geq 2$, see Table 1. We anticipate that this work will stimulate further research on global convergence proofs under general random initialization for matrix factorization with arbitrary depth and arbitrary - possibly low-rank - target matrices.

The observed divergence in convergence behavior between real and complex initializations also reveals a subtle disparity, suggesting that complex initializations may circumvent certain saddle points introduced by exact balancedness that real initializations are not capable of. Previous work have addressed the drawback of exact balancedness on real domain (Xiong et al., 2023). This might motivate more detailed analysis of the performance gap between complex and real neural networks.

### Reproducibility Statement

All theoretical results stated in this paper are proved in full detail in the Appendix, from Section B to I, including the proofs of all main-text theorems as well as intermediate lemmas and derivations, so that a reader can verify each step independently. The numerical illustration in Appendix K, where we specify the hyper-parameters in that section. Because the experiments are straightforward, we have not released an implementation.

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

## A  ORGANIZATION OF THE APPENDIX

This section outlines the organization of the Appendix to facilitate navigation. The core technical journey, comprising the main convergence proofs, spans from Appendix B to I. Following this, Appendix J provides insights into the global convergence rate, and Appendix K presents supporting numerical simulations.

Appendix B completes the proof of Reduction To Diagonal (Identical) Target discussed in Section 3 so that we can assume target matrix to be diagonal (some cases identical). While subsection B.1 proves that the form of dynamics remains the same, B.2 claims that the initializations we considers throughout this paper are invariant under the reduction.

Appendix C proves the properties of balanced Gaussian initialization (6) and random Gaussian initialization (13) stated in Theorem 3 and 6, respectively. C.1 states and proves some lemmas on Circular Ensembles, leading to the proof of Theorem 6 in C.2 and the proof of Theorem 3 in subsection C.3. Then, C.4 establishes a general property for any balanced initialization.

Appendix D presents fundamental lemmas utilized in subsequent sections:

- D.1 collects standard results from classical matrix analysis, including spectral properties and perturbation bounds.

- D.2 provides two specific perturbation bounds, which serve as preliminaries for bounding eigenvalue changes in discrete time.

- D.3 establishes the existence of analytic singular value decomposition for the general $N$-layer matrix factorization under gradient flow. It also derives the time derivatives of the decomposed matrices, thereby laying the groundwork for the proof of Theorem 2 in E.

- D.4 analyzes the dynamics with a regularization term under gradient flow. Specifically, it investigates: 1. the convergence behavior of the regularization term; 2. Upper and lower bounds for the maximum and minimum singular values of the weight matrices.

  The results for gradient flow are then adapted in D.5 to prove the corresponding theorems for gradient descent: Theorem 7 and Theorem 8.

Appendix E analyzes dynamics under gradient flow with balanced Gaussian initialization. E.1 proves Theorem 4 for arbitrary depth $N$, while E.2 proves Theorem 5 for $N = 4$ and target matrix $\Sigma = \sigma_1(\Sigma)I$. By combining these results, E.3 formally states and proves Theorem 2, completing the global convergence proof for balanced Gaussian initialization.

To prepare for generalization of this method on random Gaussian initialization, Appendix F further defines some notations and inequalities, Appendix G adapts the terms studied in Theorem 4 and 5 into imbalanced setting.

Appendix H completes the proof of global convergence under $N = 4$, $\Sigma = \sigma_1(\Sigma)I$ by dividing the training dynamics into three stages analyzed in H.1, H.2 and H.3.

Appendix I then adapts the proof intuition into gradient descent, completing the proof of Theorem 1.

Appendix J provides a discussion of the convergence rate in Theorem 1. J.1 details the calculation of the example after Theorem 1, verifying the near-tightness of the upper bound. J.2 analyzes the exponent of $\sigma_1(\Sigma)$ in the initialization scale and the convergence rate, from both scaling and dimensional analysis perspectives.

Appendix K conducts three simulation experiments. K.1 illustrates the saddle avoidance behavior of both identity and non-identity targets, under complex and real balanced Gaussian initialization. K.2 compares the convergence behavior for different depths under complex balanced Gaussian initialization. K.3 illustrates Theorem 8 and Remark 5 through the simulation with only the balance regularization term.

# B    REDUCTION TO DIAGONAL (IDENTICAL) TARGET

For arbitrary ground truth $\Sigma \in \mathbb{F}^{d \times d}$, $\mathbb{F} = \mathbb{C}$ or $\mathbb{R}$, suppose its singular value decomposition is $\Sigma = U_\Sigma \Sigma' V_\Sigma^H$ (replace $\cdot^H$ by $\cdot^\top$ for the real case, same for the rest of the analysis), we apply the following transformation:

$$\begin{cases} W_1' & = W_1 V_\Sigma \\ W_j' & = W_j, \ j \in \{2, 3, \cdots, N-1\} \\ W_N' & = U_\Sigma^H W_N \end{cases} . \tag{19}$$

Then the balance difference can be rewritten as

$$\Delta_{j,j+1} = \begin{cases} W_j' W_j'^H - W_{j+1}'^H W_{j+1}' & , j \in \{1, 2, \cdots, N-1\} \\ O^{d \times d} & , j \in \{0, N\} \end{cases} . \tag{20}$$

## B.1    TRAINING DYNAMICS

For gradient flow, the dynamics becomes

$$\frac{\mathrm{d}W_j'}{\mathrm{d}t} = \left( \prod_{k=j+1}^{N} W_k'^H \right) \left( \Sigma' - \prod_{k=N}^{1} W_k' \right) \left( \prod_{k=1}^{j-1} W_k'^H \right) + aW_j'\Delta_{j-1,j} - a\Delta_{j,j+1}W_j'. \tag{21}$$

For gradient descent,

$$W_j'(t+1) = W_j'(t) + \eta \left( \prod_{k=j+1}^{N} W_k'(t)^H \right) \left( \Sigma' - \prod_{k=N}^{1} W_k'(t) \right) \left( \prod_{k=1}^{j-1} W_k'(t)^H \right)$$
$$+ \eta a W_j'(t)\Delta_{j-1,j}(t) - \eta a \Delta_{j,j+1}(t)W_j'(t). \tag{22}$$

Both share the same form as the original one (by replacing $\Sigma$ with $\Sigma'$).

## B.2    INITIALIZATION

However, the distributions of $W_1$ and $W_N$ at initialization change correspondingly. To address this issue, we introduce the following definition:

**Definition 1.** *Input-Output Unitary(Orthogonal)-Invariant initialization.*

*For a $N$-layer complex (real) matrix factorization $W = \prod_{j=N}^{1} W_j$, an initialization is input-output unitary-invariant (in the complex case) or orthogonal-invariant (in the real case) if the distribution of $W_N$ is left unitarily (or orthogonally) invariant and the distribution of $W_1$ is right unitarily (or orthogonally) invariant. That is, for all $U, V \in U(d, \mathbb{C})$ (or $O(d, \mathbb{R})$ in the real case),*

$$W_N \overset{d}{=} UW_N, \ W_1 \overset{d}{=} W_1V. \tag{23}$$

**Remark 6.** *The distribution of $W_{j, j \in \{1, 2, \cdots, N\}}$ does not change under transformation 19 if the initialization is Input-Output Unitary(Orthogonal)-Invariant.*

Throughout this work, the initialization schemes discussed (including random Gaussian initialization and balanced Gaussian initialization) are Input-Output Unitary(Orthogonal)-Invariant. This is from the left and right invariance under multiplication of unitary/orthogonal matrices.

Thus without loss of generality, the target matrix can be reduced to positive semi-definite diagonal matrix. Under Input-Output Unitary(Orthogonal)-Invariant initialization discussed in Definition 1, the initialization on $W_1$ and $W_N$ is not affected by this reduction.

Moreover, if all singular values of $\Sigma$ are the same (to rephrase, a unitary/orthogonal matrix scaled by a constant), the convergence analysis can be reduced to $\Sigma' = \sigma_1(\Sigma)I$.

# C INITIALIZATION

First and foremost, we introduce the concept of Circular ensembles (Dyson, 1962) along with some properties.

## C.1 LEMMAS FOR GAUSSIAN RANDOM MATRIX ENSEMBLE AND HAAR MEASURE ON $U(d, \mathbb{C})$ AND $O(d, \mathbb{R})$

In the following derivations, we denote $O(d, \mathbb{R})$ as the $d$-dimensional orthogonal group on real number, and $U(d, \mathbb{C})$ as the $d$-dimensional unitary group on complex number.

We list the classical conclusions in Linear Algebra without proof:

**Lemma 10.** *The eigenvalues of Orthogonal/Unitary Matrices.*

*1. Unitary matrices. $\forall U \in U(d, \mathbb{C})$, $d \in \mathbb{N}^*$, the eigenvalues of $U$ are $e^{i\theta_{1,2,\cdots,d}}$, where $\theta_i \in [0, 2\pi)$.*

*2. Orthogonal matrices. $\forall O \in O(d, \mathbb{R})$, $d \in \mathbb{N}^*$, the eigenvalues of $O$ are:*

$$
\begin{cases}
1, e^{\pm i\theta_{1,2,\cdots,m}} & , d = 2m+1, \det(O) = 1 \\
-1, e^{\pm i\theta_{1,2,\cdots,m}} & , d = 2m+1, \det(O) = -1 \\
e^{\pm i\theta_{1,2,\cdots,m}} & , d = 2m, \det(O) = 1 \\
1, -1, e^{\pm i\theta_{1,2,\cdots,m-1}} & , d = 2m, \det(O) = -1
\end{cases}
\tag{24}
$$

Following the conventions, we call the argument of the eigenvalues as eigenangles.

**Definition 2.** *Circular ensembles. (refer to Dyson (1962), Forrester (2010))*

*The circular ensembles are measures on spaces of unitary(or orthogonal, when generalizing from complex number to real number) matrices.*

*1. Unitary circular ensemble. The distribution of the unitary circular ensemble (CUE) is the Haar measure on $d$-dimensional (complex) unitary group $U(d, \mathbb{C})$.*

*2. Circular real ensemble. The distribution of the circular real ensemble (CRE) is the Haar measure on $d$-dimensional real orthogonal group $O(d, \mathbb{R})$.*

**Lemma 11.** *1-point correlation function of $\mathrm{CUE}(d)$ and $\mathrm{CRE}(d)$.*

*1. CUE. The 1-point correlation function of $\mathrm{CUE}(d)$ is*

$$
\rho_{(1),\mathrm{CUE}}(\theta) = \frac{d}{2\pi}.
\tag{25}
$$

*2. CRE, determinant $1$. The 1-point correlation function of $\mathrm{CRE}(d)$ under determinant $1$ is*

$$
\rho_{(1),\mathrm{CRE},\det=1}(\theta) = \frac{1}{2\pi}\left(d - 1 + (-1)^d \frac{\sin(d-1)|\theta|}{\sin|\theta|}\right), \theta \in (-\pi, \pi].
\tag{26}
$$

**Remark 7.** *1-point correlation function $\rho_{(1)}(\theta)$ can be interpreted as the density of eigenangles at $\theta$ (despite probably existed fixed eigenangles, e.g. $0$, $\pi$).*

*Proof.* Part 1. CUE.

From (146) of Dyson (1962) and Forrester (2010), the joint probability density function of eigenangles is

$$
p_{\mathrm{CUE}}(\theta_{k,k\in\{1,2,\cdots,d\}}) \propto \prod_{1 \le k < j \le d} \left|e^{i\theta_j} - e^{i\theta_k}\right|^2 = \prod_{1 \le k < j \le d} \left|e^{i(\theta_j - \theta_k)} - 1\right|^2.
\tag{27}
$$

Notice that it is rotation invariant, that is $\forall \Delta\theta \in [0, 2\pi]$, $p_{\text{CUE}}(\theta_{k, k\in\{1,2,\cdots,d\}}) = p_{\text{CUE}}((\theta_k + \Delta\theta)_{k\in\{1,2,\cdots,d\}})$. Thus the 1-point correlation function (density of eigenangles at $\theta$) is uniform, which is $\frac{d}{2\pi}$.

Part 2. CRE.

Below we define $x_i = \cos\theta_i$, then $\rho_{(1)}(\theta) = \sin\theta \cdot \rho_{(1)}(x)$, $p(x_{k,k\in\{1,2,\cdots,K\}}) = \left(\prod_{k=1}^{K} \frac{1}{\sqrt{1-x^2}}\right) p(\theta_{k,k\in\{1,2,\cdots,K\}})$.

By combining Proposition 5.1.1 and 5.1.2 in Forrester (2010) together, suppose with $p_k(x)$ a polynomial of degree $k$ which is further more monic (i.e. the coefficient of $x^k$ is unity), $\{p_k(x)\}_{k\in\mathbb{N}}$ is the orthogonal polynomials associated with the weight function $w_2(x)$,

$$\int_{-\infty}^{+\infty} p_j(x)p_k(x)w_2(x)\mathrm{d}x =: \langle p_j, p_k\rangle_2 = \langle p_j, p_j\rangle_2 \delta_{j,k}. \tag{28}$$

Here $\delta_{j,k} = \mathbf{1}\{j = k\}$ is the Kronecker delta function. And the joint probability density function satisfies

$$p(x_{k,k\in\{1,2,\cdots,K\}}) \propto \prod_{1\leq k<j\leq K} (x_j - x_k)^2 \prod_{l=1}^{K} w_2(x). \tag{29}$$

The 1-point correlation function is

$$\rho_{(1)}(x) = w_2(x) \sum_{\nu=0}^{K-1} \frac{p_\nu^2(x)}{\langle p_\nu, p_\nu\rangle_2}. \tag{30}$$

Note that the restriction of monic can be omitted since there is a normalization coefficient on the denominator.

2.1. CRE, determinant 1, $d = 2K$. From (135) of Dyson (1962), Section 2.9 of Forrester (2010) and Girko (1985),

$$p_{\text{CRE,even,det=1}}(\theta_{k,k\in\{1,2,\cdots,K\}}) \propto \prod_{1\leq k<j\leq K} |\cos\theta_j - \cos\theta_k|^2, \quad \theta_{k,k\in\{1,2,\cdots,K\}} \in [0, \pi]. \tag{31}$$

By the change of variables,

$$p_{\text{CRE,even,det=1}}(x_{k,k\in\{1,2,\cdots,K\}}) \propto \prod_{1\leq k<j\leq K} (x_j - x_k)^2 \prod_{l=1}^{K} \frac{1}{\sqrt{1-x_l^2}}. \tag{32}$$

Here $w_2(x) = \frac{1}{\sqrt{1-x^2}}$. From knowledge of orthogonal polynomials ((1.12.3), (4.1.7), Szegő (1939)), Chebyshev polynomials of the first kind $T_n(x) = \cos(n\arccos x)$ associates with $w_2(x) = \frac{1}{\sqrt{1-x^2}}$:

$$\int_{-1}^{1} T_j(x)T_k(x)w_2(x)\mathrm{d}x = \begin{cases} \pi, & j = k = 0 \\ \frac{\pi}{2}, & j = k \geq 1 \\ 0, & j \neq k \end{cases}. \tag{33}$$

By (30),

$$\rho_{(1),\text{CRE,even,det}=1}(x) = \frac{1}{\sqrt{1-x^2}} \cdot \left( \frac{1}{\pi} + \frac{2}{\pi} \sum_{\nu=1}^{K-1} \cos^2 \nu\theta \right)$$
$$= \frac{1}{2\pi \sin\theta} \left[ 2K - 1 + \frac{\sin(2K-1)\theta}{\sin\theta} \right]. \tag{34}$$

$$\rho_{(1),\text{CRE,even,det}=1}(\theta) = \frac{1}{2\pi} \left[ d - 1 + \frac{\sin(d-1)\theta}{\sin\theta} \right], \, \theta \in [0, \pi]. \tag{35}$$

From symmetry, $\rho_{(1),\text{CRE,even,det}=1}(-\theta) = \rho_{(1),\text{CRE,even,det}=1}(\theta)$.

2.2. CRE, determinant 1, $d = 2K + 1$. From (137) of Dyson (1962), Section 2.9 of Forrester (2010) and Girko (1985),

$$p_{\text{CRE,odd,det}=1}(\theta_{k,k\in\{1,2,\cdots,K\}}) \propto \prod_{1 \le k < j \le K} |\cos\theta_j - \cos\theta_k|^2 \prod_{l=1}^{K} (1 - \cos\theta_l), \, \theta_{k,k\in\{1,2,\cdots,K\}} \in [0, \pi]. \tag{36}$$

By the change of variables,

$$p_{\text{CRE,odd,det}=1}(x_{k,k\in\{1,2,\cdots,K\}}) \propto \prod_{1 \le k < j \le K} (x_j - x_k)^2 \prod_{l=1}^{K} \sqrt{\frac{1-x_l}{1+x_l}}. \tag{37}$$

Here $w_2(x) = \sqrt{\frac{1-x}{1+x}}$. From knowledge of orthogonal polynomials ((1.12.3), (4.1.7), Szegő (1939)), Chebyshev polynomials of the fourth kind $W_n(x) = \frac{\sin\left(\left(n+\frac{1}{2}\right)\theta\right)}{\sin\left(\frac{\theta}{2}\right)}$, $\theta = \arccos x$ associates with $w_2(x) = \sqrt{\frac{1-x}{1+x}}$:

$$\int_{-1}^{1} W_j(x) W_k(x) w_2(x) \mathrm{d}x = \begin{cases} \pi, & j = k \ge 0 \\ 0, & j \ne k \end{cases}. \tag{38}$$

By (30),

$$\rho_{(1),\text{CRE,odd,det}=1}(x) = \sqrt{\frac{1-x}{1+x}} \cdot \left( \frac{1}{\pi} \sum_{\nu=0}^{K-1} \left( \frac{\sin\left(\left(n+\frac{1}{2}\right)\theta\right)}{\sin\left(\frac{\theta}{2}\right)} \right)^2 \right)$$
$$= \frac{1}{2\pi \sin(\theta)} \left[ 2K - \frac{\sin(2K\theta)}{\sin\theta} \right]. \tag{39}$$

$$\rho_{(1),\text{CRE,odd,det}=1}(\theta) = \frac{1}{2\pi} \left[ d - 1 - \frac{\sin(d-1)\theta}{\sin\theta} \right], \, \theta \in [0, \pi]. \tag{40}$$

From symmetry, $\rho_{(1),\text{CRE,odd,det}=1}(-\theta) = \rho_{(1),\text{CRE,odd,det}=1}(\theta)$.

This completes the proof.

$\square$

**Theorem 12.** *For $Q$ sampled from Haar measure on $U(d, \mathbb{C})$ (or $O(d, \mathbb{R})$ if $\mathbb{F} = \mathbb{R}$),*

*1. $\mathbb{F} = \mathbb{C}$. $\Pr(\sigma_{\min}(I + Q) \ge \pi\delta d^{-1}) \ge 1 - \delta$.*

*2. $\mathbb{F} = \mathbb{R}$. If $d \ge 2$, $\Pr\left(\sigma_{\min}(I + Q) \ge \frac{\pi\delta}{2}(d-1)^{-1} \middle| \det(Q) = 1\right) \ge 1 - \delta$.*

**Remark 8.** *For $\mathbb{F} = \mathbb{R}$, $d = 1$, the eigenvalue of $Q$ is $\det(Q)$, and thus $\Pr(\sigma_{\min}(I + Q) \geq 2 - \Delta| \det(Q) = 1) = 1$, $\forall \Delta \in (0, 2)$.*

**Remark 9.** *For $\mathbb{F} = \mathbb{R}$, $\Pr(\det(Q) = 1) = \Pr(\det(Q) = -1) = \frac{1}{2}$. If $\det(Q) = -1$, $Q$ has an eigenvalue of $-1$, causing $\Pr(\sigma_{\min}(I + Q)) = 0$.*

*Proof.* Consider $\theta_k \in (-\pi, \pi]$,

$$\sigma_k(I + Q) = \sqrt{\lambda_k(2I + Q + Q^H)} = \sqrt{2 + e^{i\theta_k} + 1/e^{i\theta_k}} = 2\cos\left(\frac{\theta_k}{2}\right)$$

$$\sigma_{\min}(I + Q) = \min_k \cos\left(\frac{\theta_k}{2}\right). \tag{41}$$

The second step is from the fact that $Q^H = Q^{-1}$ shares the same eigenvectors with $Q$, and corresponding eigenvalues are the reciprocal of the original eigenvalues.

Denote $N(\delta\theta)$ to be number of eigenvectors in $(-\pi, -\pi + \delta\theta] \cup [\pi - \delta\theta, \pi]$, $\delta\theta \in (0, \pi)$. From Markov inequality,

$$\Pr\left(\sigma_{\min}(I + Q) \geq \delta\theta\right) \geq \Pr\left(\sigma_{\min}(I + Q) \geq 2\sin\frac{\delta\theta}{2}\right)$$

$$= 1 - \Pr(N(\delta\theta) \geq 1) \tag{42}$$

$$\geq 1 - \mathbb{E}(N(\delta\theta)) = 1 - \int_{\theta \in (-\pi, -\pi+\delta\theta] \cup [\pi-\delta\theta, \pi]} \rho_{(1)}(\theta)\mathrm{d}\theta.$$

By invoking Lemma 11,

1. For $\mathbb{F} = \mathbb{C}$,

$$\mathbb{E}(N(\delta\theta)) = \frac{d}{2\pi} \cdot 2\delta\theta. \tag{43}$$

By setting $\delta\theta = \pi\delta d^{-1}$, $\Pr\left(\sigma_{\min}(I + Q) \geq \delta\theta\right) \geq 1 - \delta$.

2. For $\mathbb{F} = \mathbb{R}$ under determinant 1, for $\theta' \in [0, \pi]$, $\rho_{(1)}(\pi - \theta') = \frac{1}{2\pi}\left(d - 1 + \frac{\sin(d-1)\theta'}{\sin\theta'}\right)$.

If $d = 1$, $\rho_{(1)}(\theta) \equiv 0$ and thus $\mathbb{E}(N(\delta\theta)) = 0$. For $d \geq 2$:

From $\frac{\sin(d-1)\theta}{\sin\theta} \leq d - 1$,

$$\mathbb{E}(N(\delta\theta)) = 2\int_0^{\delta\theta} \rho_{(1)}(\pi - \theta')\mathrm{d}\theta' \leq 2\int_0^{\delta\theta} \frac{1}{2\pi} \cdot 2(d-1)\mathrm{d}\theta' = \frac{2(d-1)}{\pi}\delta\theta. \tag{44}$$

By setting $\delta\theta = \frac{\pi\delta}{2}(d - 1)^{-1}$, $\Pr\left(\sigma_{\min}(I + Q) \geq \delta\theta| \det(Q) = 1\right) \geq 1 - \delta$.

This completes the proof.

$\square$

## C.2 RANDOM GAUSSIAN INITIALIZATION

In the following, we present the proof for Theorem 6.

For a real/complex Gaussian random matrix of dimension $d \times d$, with probability at least $\delta$, the largest singular value is upper bounded by $O\left(\left(1 + \sqrt{\frac{\ln\left(\frac{1}{\delta}\right)}{d}}\right)\sqrt{d}\right)$ (Theorem 4.4.5, Vershynin (2018)), while the smallest is lower bounded by $\Omega\left(\frac{\delta}{\sqrt{d}}\right)$ (Theorem 1.1, Tao & Vu (2009)). (also refer to Corollary 2.3.5 and Theorem 2.7.5 of Tao )

*Proof.* The upper and lower bound for singular values of $W_k$ follows immediately. The main challenge is the minimum singular value of $W + (WW^H)^{1/2}$.

At the beginning, we define a modification of Gaussian random matrix ensemble for simplification:

$W$ is sampled from (complex or real) Gaussian random matrix ensemble, and if $\text{rank}(W)$ is not full, sample $W$ from Gaussian random matrix ensemble again until it is full rank.

Since the set of $\text{rank}(W)$ not being full is zero measure, the distribution of $W$ shares the same with the one before modification almost surely, and thus changing Gaussian random matrix ensemble to modified version *does not affect* the analysis below essentially.

This modification is for better expression on definition of left and right unitary (orthogonal) matrix of SVD. For full rank square matrix $W = U\Sigma V^H$, $U$ and $V$ are not unique, but $VU^H$ is (even if the singular values are non-distinct, or changing the order of diagonal elements of $\Sigma$. This is due to the uniqueness of polar decomposition $W = SQ$ under full rank, where $Q = UV^H$, $S = (WW^H)^{1/2}$. ) and thus well-defined.

Without changing the result, we analysis the initialization scheme of modified Gaussian random matrix ensemble instead. Then $W$ is full rank and thus polar decomposition is unique.

Generally, suppose the right polar decomposition of $W$ is $W = \left(WW^H\right)^{1/2} Q$, then

$$W + \left(WW^H\right)^{1/2} = \left(WW^H\right)^{1/2} (I + Q). \tag{45}$$

If $\mathbb{F} = \mathbb{R}$, $\Pr(\det(W) > 0) = \Pr(\det(W) < 0) = \frac{1}{2}$ due to the symmetry of Gaussian random matrix ensemble. If $\det(W) = \det\left(\left(WW^H\right)^{1/2}\right)\det(Q) < 0$, $\det(Q) = -1$, then $\sigma_{\min}(I + Q) = 0$ and further $\sigma_{\min}\left(W + \left(WW^\top\right)^{1/2}\right) = 0$.

Consider both $\mathbb{F} = \mathbb{C}$ and $\mathbb{F} = \mathbb{R}$, $\det(W) > 0$ (which indicates $\det(Q) = 1$):

$$
\begin{aligned}
\sigma_{\min}\left(W + \left(WW^H\right)^{1/2}\right) &\geq \sigma_{\min}\left(\left(WW^H\right)^{1/2}\right)\sigma_{\min}(I + Q) \\
&= \sigma_{\min}(W)\sigma_{\min}(I + Q) \\
&\geq \left[\prod_{k=1}^{N}\sigma_{\min}(W_k)\right]\sigma_{\min}(I + Q).
\end{aligned}
\tag{46}
$$

From Theorem 1.1 of Tao & Vu (2009), by applying union bound, $\sigma_{\min}(W_{k,k\in\{1,2,\cdots,N\}}) > f_1^{-1}(\delta, N)d^{-1/2}\epsilon$ with high probability $1 - \delta/2$, where $f_1(\delta, N) = O\left(\frac{N}{\delta}\right)$. Then $\left[\prod_{k=1}^{N}\sigma_{\min}(W_k)\right] \geq \left(f_1^{-1}(\delta, N)d^{-1/2}\epsilon\right)^N$, and it remains to find lower bound for $\sigma_{\min}(I + Q)$.

To apply results in Theorem 12, it is sufficient to show that $Q$ follows Haar measure on $U(d, \mathbb{C})$ (or $O(d, \mathbb{R})$).

Due to the property of invariance under left and right multiplication of unitary (orthogonal) matrix for Gaussian random matrix ensemble (Section 2.6.2, (2.131), Tao), $\forall$ fixed $Q_0 \in U(d, \mathbb{C})$ (or $O(d, \mathbb{R})$ if $\mathbb{F} = \mathbb{R}$), $W_1 Q_0^H$ follows the same distribution as $W_1$ while still independent of $W_{k,k\in\{2,3,\cdots,N\}}$, resulting that $WQ_0^H$ follows the same distribution as $W$. Since the right polar decomposition of $WQ_0^H$ is $WQ_0^H = \left(WQ_0^H Q_0 W^H\right)^{1/2} QQ_0^H = \left(WW^H\right)^{1/2}\left(QQ_0^H\right)$, we have

$$Q_0 Q \overset{d}{=} Q, \; \forall \text{ fixed } Q_0 \in U(d, \mathbb{C}) \text{ (or } O(d, \mathbb{R}) \text{ if } \mathbb{F} = \mathbb{R}). \tag{47}$$

Likewise

$$QQ_0 \overset{d}{=} Q, \; \forall \text{ fixed } Q_0 \in U(d, \mathbb{C}) \text{ (or } O(d, \mathbb{R}) \text{ if } \mathbb{F} = \mathbb{R}). \tag{48}$$

From the fact that the only measure invariant under left (or right) multiplication of arbitrary element of a compact lie group is Haar measure, $Q$ follows Haar measure on $U(d, \mathbb{C})$ (or $O(d, \mathbb{R})$), and the proof is completed.

$\square$

By Theorem 6, for depth $N = 4$, if $\mathbb{F} = \mathbb{C}$ then with high probability $1 - \delta$ (if $\mathbb{F} = \mathbb{R}$ then with probability $1/2$, $\sigma_{\min}\left(W(0) + \left(W(0)W(0)^\top\right)^{1/2}\right) = 0$, and with probability $(1 - \delta)/2$ the following holds), $\exists f_1(\delta) = O\left(\frac{1}{\delta}\right), f_2(\delta) = O(\frac{1}{\delta^5})$ such that

$$\max_{j,k} \sigma_k(W_j(0)) \leq f_1(\delta)\sqrt{d}\epsilon$$

$$\min_{j,k} \sigma_k(W_j(0)) \leq \frac{1}{f_1(\delta)\sqrt{d}} \cdot \epsilon \tag{49}$$

$$\sigma_{\min}\left(W(0) + \left(W(0)W(0)^H\right)^{1/2}\right) \geq \frac{1}{f_2(\delta)d^3} \cdot \epsilon^4.$$

Consequently,

$$e_\Delta(0) := \left.\sqrt{\sum_{i=1}^{3} \|\Delta_{i,i+1}\|_F^2}\right|_{t=0} \leq \sqrt{3} \cdot 2\sqrt{d} \cdot \max_{j,k} \sigma_k^2(W_j(0)) = 2\sqrt{3}f_1^2(\delta)d^{3/2}\epsilon^2. \tag{50}$$

### C.3 BALANCED GAUSSIAN INITIALIZATION

This section analyzes the balanced Gaussian initialization scheme.

**Corollary 13.** *Under balanced Gaussian initialization scheme (6), each matrix $W_{k,k\in\{1,2,\cdots,N\}}$ is a Gaussian random matrix ensemble scaled by $\epsilon$.*

*Proof.* This is immediately from the property of invariance under left and right multiplication of unitary (orthogonal) matrix for Gaussian random matrix ensemble (Section 2.6.2, (2.131), Tao).

$\square$

Due to Corollary 24, the product matrix can be expressed as $U\Sigma_w^N V^H$. Then we present the proof of Theorem 3.

*Proof.* We first consider $2 \mid N$. From (6), $W(t = 0) = s\epsilon^N Q_{N,N+1}(G^H G)^{N/2} Q_{01}^H$.

Naturally $\|\Sigma_w\|_{op} = \epsilon \left\|(G^H G)^{1/2}\right\|_{op} = \epsilon \|G\|_{op} = O\left(1 + \sqrt{\frac{\ln\left(\frac{1}{\delta}\right)}{d}}\right)\sqrt{d}\epsilon$. Last step is from Theorem 4.4.5 of Vershynin (2018) directly.

For the other two terms,

$$\sigma_{\min}\left((U + V)\Sigma_w\right)|_{t=0}$$

$$= \left.\sqrt{\lambda_{\min}\left((U + V)\Sigma_w^2(U + V)^H\right)}\right|_{t=0}$$

$$= \left.\sqrt{\lambda_{\min}\left((WW^H)^{\frac{1}{N}} + (W^H W)^{\frac{1}{N}} + (WW^H)^{-\frac{N-2}{2N}} W + (W^H W)^{-\frac{N-2}{2N}} W^H\right)}\right|_{t=0} \tag{51}$$

$$= \epsilon\sqrt{\lambda_{\min}\left((Q_{01} + sQ_{N,N+1})(G^H G)(Q_{01} + sQ_{N,N+1})^H\right)}$$

$$\in \left[\epsilon\sigma_{\min}(I + sQ_{01}^H Q_{N,N+1})\sigma_{\min}(G), \epsilon\sigma_{\min}(I + sQ_{01}^H Q_{N,N+1})\sigma_{\max}(G)\right].$$

And

$$\|(U - V)\Sigma_w\|_F|_{t=0} \leq 2\sqrt{d}\epsilon\|G\|_{op}. \tag{52}$$

Since $Q_{N,N+1}$ and $Q_{01}$ are independent and both sampled from Haar measure, then $Q_{01}^H Q_{N,N+1} \sim$ Haar on $U(d, \mathbb{C})$ (or $O(d, \mathbb{R})$ if $\mathbb{F} = \mathbb{R}$) as well.

For $\mathbb{F} = \mathbb{R}$, since $s$ is independent of $Q_{j,j \in \{0,1,\cdots,N\}}$, $\Pr(s \det(Q_{N,N+1}) \det(Q_{01}) = 1) = \Pr(s \det(Q_{N,N+1}) \det(Q_{01}) = -1) = \frac{1}{2}$ is directly from symmetry of Haar measure.

Then by combining Theorem 12 and Theorem 4.4.5 of Vershynin (2018), Theorem 1.1 of Tao & Vu (2009) (with high probability $1 - \delta'$, $\max(\|G\|_{op}, \|G^{-1}\|_{op}) \leq f_1(\delta')\sqrt{d}$, $f_1(\delta') = O(\frac{1}{\delta'})$), the proof for $2 \mid N$ is completed.

For $2 \nmid N$, suppose the SVD of $G$ is $G = U_G \Sigma_G V_G^H$, then $W(t = 0) = s\epsilon^N(Q_{N,N+1}U_G V_G^H)(G^H G)^{N/2}Q_{01}^H$. Note that since $Q_{N,N+1}$ and $G$ are independent, then $Q_{N,N+1}U_G V_G^H \sim$ Haar, $Q_{N,N+1}U_G V_G^H$ and $Q_{01}$ are independent. Then the proof for $2 \nmid N$ is completed by replacing the $Q_{N,N+1}$ with $Q_{N,N+1}U_G V_G^H$ in the derivations.

$\square$

### C.4 GENERAL BALANCED INITIALIZATION

This section introduces a property for general balanced and input-output orthogonal-invariant initialization (refer to Definition 1) under real field.

**Theorem 14.** *For any real matrix factorization, if the initialization is balanced and input-output orthogonal-invariant, then the minimum singular value of $W + (WW^\top)^{1/2}$ at $t = 0$ is exactly $0$ with at least probability $1/2$:*

$$\Pr\left(\sigma_{\min}\left(W + (WW^\top)^{1/2}\right) = 0\right) \geq 1/2. \tag{53}$$

*Proof.* As a direct consequence of Definition 1, $W$ is left and right orthogonal invariant:

$$W \stackrel{d}{=} U'WV', \forall U', V' \in O(d, \mathbb{R}). \tag{54}$$

Suppose the right polar decomposition of $W$ is $W = WW^\top Q$, following the same arguments in the proof (C.2) of Theorem 6,

$$W + (WW^\top)^{1/2} = (WW^\top)^{1/2}(I + Q), Q \sim \text{Haar}. \tag{55}$$

From Theorem 12, $\Pr(\sigma_{\min}(I + Q) = 0) = \frac{1}{2}$, resulting

$$\Pr\left(\sigma_{\min}\left(W + (WW^\top)^{1/2}\right) = 0\right) \geq \Pr(\sigma_{\min}(I + Q) = 0) = \frac{1}{2}. \tag{56}$$

This completes the proof.

$\square$

## D BASIC LEMMAS

### D.1 CLASSIC MATRIX ANALYSIS CONCLUSIONS

**Lemma 15.** *Let $R \in \mathbb{F}^{d \times d}$, where $\mathbb{F} = \mathbb{C}$ or $\mathbb{R}$. Then:*

*1. $I - RR^H$ and $I - R^H R$ (or $I - RR^\top$ and $I - R^\top R$ if $\mathbb{F} = \mathbb{R}$) share the same set of eigenvalues.*

*2. These eigenvalues are real-valued.*

*Proof.* We prove the complex case, and the real case follows. Suppose the singular value decomposition of $R$ is $U_R \Sigma_R V_R^H$, then

$$
\begin{aligned}
I - RR^H &= I - U_R \Sigma_R^2 U_R^H = U_R \left( I - \Sigma_R^2 \right) U_R^H \\
I - R^H R &= I - V_R \Sigma_R^2 V_R^H = V_R \left( I - \Sigma_R^2 \right) V_R^H.
\end{aligned}
\tag{57}
$$

Thus both $I - RR^H$ and $I - R^H R$ are unitarily similar to $I - \Sigma_R^2$, which completes the proof. $\square$

**Lemma 16.** *Given symmetric matrices $X, \Delta \in \mathbb{F}^{d \times d}$, where $\mathbb{F} = \mathbb{C}$ or $\mathbb{R}$, suppose $X \succ \|\Delta\|_{op} I \succ O$, then*

$$
\left\| X^{1/2} - (X + \Delta)^{1/2} \right\|_{op} \leq \frac{\|\Delta\|_{op}}{2(\lambda_{\min}(X) - \|\Delta\|_{op})^{1/2}}.
\tag{58}
$$

*Proof.* Directly by Theorem X.3.8 and inequality (X.46) in Bhatia (1996).

$\square$

**Lemma 17.** $\forall X, \Delta \in \mathbb{F}^{d \times d}$, *where $\mathbb{F} = \mathbb{C}$ or $\mathbb{R}$, if $X$ and $X + \Delta$ are both invertible, then*

$$
(X + \Delta)^{-1} - \left( X^{-1} - X^{-1} \Delta X^{-1} \right) = X^{-1} \Delta X^{-1} \Delta (X + \Delta)^{-1}.
\tag{59}
$$

*Proof.*

$$
\begin{aligned}
(X + \Delta)^{-1} - \left( X^{-1} - X^{-1} \Delta X^{-1} \right) &= X^{-1} \left[ X - (X - \Delta) X^{-1} (X + \Delta) \right] (X + \Delta)^{-1} \\
&= X^{-1} \Delta X^{-1} \Delta (X + \Delta)^{-1}.
\end{aligned}
\tag{60}
$$

$\square$

**Lemma 18.** *Bound of eigenvalues under perturbation.*

*For unitary (or orthogonal, for real field) $d$-dimensional matrices $U$, $V$, positive semi-definite matrix $S$, denote $P := \left( \frac{U+V}{2} \right) S \left( \frac{U+V}{2} \right)^H$, then the eigenvalues of $S$ are bounded by*

$$
\lambda_k (P) \leq \lambda_k(S) \leq
\begin{cases}
2 \left[ \lambda_k (P) + \left\| \left( \frac{U-V}{2} \right) S \left( \frac{U-V}{2} \right)^H \right\|_{op} \right] & , 1 \leq k \leq d - 1 \\
\lambda_k (P) + \left\| \left( \frac{U-V}{2} \right) S \left( \frac{U-V}{2} \right)^H \right\|_{op} & , k = d
\end{cases}.
\tag{61}
$$

*Proof.* Let $Q = U^H V$.

Due to Courant-Fischer min-max Theorem, $A \succeq B$ indicates $\lambda_k(A) \geq \lambda_k(B)$. Then the lower bound is straight forward:

$$
\begin{aligned}
\lambda_k \left( \left( \frac{U+V}{2} \right) S \left( \frac{U+V}{2} \right)^H \right) &= \lambda_k \left( S^{1/2} \left( \frac{U+V}{2} \right) \left( \frac{U+V}{2} \right)^H S^{1/2} \right) \\
&\leq \lambda_k \left( S^{1/2} \left( \left\| \frac{U+V}{2} \right\|_{op}^2 I \right) S^{1/2} \right) \\
&\leq \lambda_k \left( S^{1/2} \left( \left( \frac{\|U\|_{op} + \|V\|_{op}}{2} \right)^2 I \right) S^{1/2} \right) = \lambda_k (S).
\end{aligned}
\tag{62}
$$

For upper bound, by applying Wely inequality,

$$\lambda_k \left( \left( \frac{U+V}{2} \right) S \left( \frac{U+V}{2} \right)^H \right) = \lambda_k \left( \left( \frac{I+Q}{2} \right) S \left( \frac{I+Q^H}{2} \right) \right)$$

$$\geq \lambda_k \left( \left( \frac{I+Q}{2} \right) S \left( \frac{I+Q^H}{2} \right) + \left( \frac{I-Q}{2} \right) S \left( \frac{I-Q^H}{2} \right) \right) - \left\| \left( \frac{I-Q}{2} \right) S \left( \frac{I-Q^H}{2} \right) \right\|_{op}$$

$$= \frac{1}{2} \lambda_k \left( S + QSQ^H \right) - \left\| \left( \frac{U-V}{2} \right) S \left( \frac{U-V}{2} \right)^H \right\|_{op}.$$

$$(63)$$

For arbitrary $k$, $\lambda_k \left( S + QSQ^H \right) \geq \lambda_k \left( S \right)$; for $k = d$, $\lambda_d \left( S + QSQ^H \right) \geq 2\lambda_d \left( S \right)$. This completes the proof.

$\square$

### D.2 LEMMAS ON EIGENVALUE CHANGE UNDER DISCRETE TIME

**Lemma 19.** *Suppose* $\Sigma, S \in \mathbb{F}^{d \times d}$ *are positive semi-definite matrices,* $0 \leq \alpha \leq \frac{1}{6} \|S\|_{op}^{-1}$, $\mathbb{F} = \mathbb{C}$ *or* $\mathbb{R}$. *Consider* $S' = (I + \alpha(\Sigma - S))S(I + \alpha(\Sigma - S))$,

$$\lambda_{\min}(S') \geq \lambda_{\min}(S)(1 + \alpha(\lambda_{\min}(\Sigma) - \lambda_{\min}(S)))^2 + O\left(\alpha^2 \left(\|\Sigma\|_{op}^2 + \|S\|_{op}^2\right) \|S\|_{op}\right)$$

$$\lambda_{\max}(S') \leq \lambda_{\max}(S)(1 + \alpha(\lambda_{\max}(\Sigma) - \lambda_{\max}(S)))^2.$$

$$(64)$$

This generalizes Lemma 3.2 in Ye & Du (2021).

*Proof.* Following the derivations in Ye & Du (2021), $\forall \beta \in (0, 1)$, rewrite the terms by the following:

$$S' = \beta \left( I - \frac{\alpha}{\beta} S \right) S \left( I - \frac{\alpha}{\beta} S \right) + (1 - \beta) \left( I + \frac{\alpha}{1 - \beta} \Sigma \right) S \left( I + \frac{\alpha}{1 - \beta} \Sigma \right)$$

$$- \frac{\alpha^2}{\beta(1 - \beta)} \left[ (1 - \beta)S + \beta\Sigma \right] S \left[ (1 - \beta)S + \beta\Sigma \right].$$

$$(65)$$

The first term has eigenvalues $\lambda_{i'}(S') = \beta \left( 1 - \frac{\alpha}{\beta} \lambda_i(S) \right)^2 \lambda_i(S)$ (note that $f(x) = (1 - x)^2 x$ is non-decreasing in $\left[ 0, \frac{1}{3} \right]$, so $\lambda_{i'}(S')$ is exactly the $i^{th}$ eigenvalue of the first term when $\beta \geq \frac{1}{2}$), while the second term is bounded by

$$(1 - \beta) \left( I + \frac{\alpha}{1 - \beta} \lambda_{\min}(\Sigma) \right)^2 \lambda_{\min}(S) \preceq \text{term2} \preceq (1 - \beta) \left( I + \frac{\alpha}{1 - \beta} \lambda_{\max}(\Sigma) \right)^2 \lambda_{\max}(S).$$

$$(66)$$

By treating the third term as error term and taking $\beta = \frac{1}{2}$, the proof is completed.

$\square$

**Lemma 20.** *Suppose* $D, S \in \mathbb{F}^{d \times d}$ *are positive semi-definite matrices,* $E \in \mathbb{F}^{d \times d}$, $\mathbb{F} = \mathbb{C}$ *or* $\mathbb{R}$. *Denote* $M = S + D$. *Consider* $S' = \left( I + \eta \left( aM - M^3 + E \right) \right) S \left( I + \eta \left( aM - M^3 + E \right) \right)$, *under* $\eta < \frac{1}{16 \left( \|M\|_{op}^3 + \|E\|_{op} \right)}$,

$$\lambda_{\min}(S') \geq \lambda_{\min}(S) + 2\eta \left( a - 2\|D\|_{op}\|M\|_{op} - \|M\|_{op}\lambda_{\min}(S) \right) \lambda_{\min}^2(S)$$

$$- 2\eta \left( \|E\|_{op} + \|D\|_{op}^2\|M\|_{op} \right) \lambda_{\min}(S)$$

$$+ O\left( \left( a^2\|M\|_{op}^2 + \|M\|_{op}^6 + \|E\|_{op}^2 \right) \|S\|_{op} \right).$$

$$(67)$$

*Proof.* Expand the expression of $S'$:

$$
\begin{aligned}
S' &= S + \eta\left(aM + E - DMD\right)S + \eta S\left(aM + E - DMD\right) \\
&\quad - \eta(DMS^2 + S^2 MD) - \eta S(MD + DM)S - \eta(SMS^2 + S^2 MS) + \eta^2 M'_{\text{error}} \\
&= \frac{1}{4}\left(I + 4\eta\left(aM + E - DMD\right)\right)S\left(I + 4\eta\left(aM + E - DMD\right)\right) \\
&\quad + \frac{1}{4s}\left(I - 4\eta s DM\right)S^2\left(I - 4\eta s MD\right) + \frac{1}{4s}S\left(I - 4\eta s\left(MD + DM\right)\right)S \\
&\quad + \frac{1}{4s^2}S\left(I - 4\eta s^2 M\right)S\left(I - 4\eta s^2 M\right)S + \left(\frac{3}{4}S - \frac{1}{2s}S^2 - \frac{1}{4s^2}S^3\right) + \eta^2 M'_{\text{error}}.
\end{aligned}
\tag{68}
$$

where $\|M'_{\text{error}}\|_{op} = O\left(\left(a^2\|M\|_{op}^2 + \|M\|_{op}^6 + \|E\|_{op}^2\right)\|S\|_{op}\right)$.

Notice that $\frac{3}{4}S - \frac{1}{2s}S^2 - \frac{1}{4s^2}S^3$ has eigenvalues $\lambda_{i'}(S') = \frac{3}{4}\lambda_i(S) - \frac{1}{2s}\lambda_i^2(S) - \frac{1}{4s^2}\lambda_i^3(S)$, so by taking $s = 2\|S\|_{op}$, $\lambda_{i'}(S')$ is exactly the $i^{th}$ eigenvalue of $S'$.

This further gives

$$
\begin{aligned}
\lambda_{\min}(S') &\geq \frac{1}{4}\left(1 + 4\eta\left(a\lambda_{\min}(M) - \|E\|_{op} - \|D\|_{op}^2\|M\|_{op}\right)\right)^2 \lambda_{\min}(S) \\
&\quad + \frac{1}{4s}\left(1 - 4\eta s\|D\|_{op}\|M\|_{op}\right)^2 \lambda_{\min}^2(S) + \frac{1}{4s}\left(1 - 8\eta s\|M\|_{op}\|D\|_{op}\right)\lambda_{\min}^2(S) \\
&\quad + \frac{1}{4s^2}\left(1 - 4\eta s^2\|M\|_{op}\right)^2 \lambda_{\min}^3(S) + \left(\frac{3}{4}\lambda_{\min}(S) - \frac{1}{2s}\lambda_{\min}^2(S) - \frac{1}{4s^2}\lambda_{\min}^3(S)\right) \\
&\quad + \eta^2\|M'_{\text{error}}\|_{op} \\
&\geq \lambda_{\min}(S) + 2\eta\left(a\lambda_{\min}(M) - 2\|D\|_{op}\|M\|_{op}\lambda_{\min}(S) - \|M\|_{op}\lambda_{\min}^2(S)\right)\lambda_{\min}(S) \\
&\quad - 2\eta\left(\|E\|_{op} + \|D\|_{op}^2\|M\|_{op}\right)\lambda_{\min}(S) + \eta^2\|M'_{\text{error}}\|_{op}.
\end{aligned}
\tag{69}
$$

From $\lambda_{\min}(M) \geq \lambda_{\min}(S)$, the proof is completed.

$\square$

### D.3 LEMMAS ON ANALYTIC SINGULAR VALUE DECOMPOSITION OF PRODUCT MATRIX UNDER BALANCED INITIALIZATION AND GRADIENT FLOW

**Lemma 21.** *Existence of analytic singular value decomposition (ASVD).*

*Under Section 3 with gradient flow and balanced initialization, for $t \in \mathbb{R}^+ \cup \{0\}$, there exists analytical singular value decompositions for $W_{j,j\in\{1,2,\cdots,N\}}(t)$ and $W(t)$.*

*Proof.* For $\mathbb{F} = \mathbb{R}$, the proof is exactly the same as Lemma 1 in Arora et al. (2019b): real analytic matrices have ASVD (Theorem 1 in Bunse-Gerstner et al. (1991/92)), and $W_j(t)$ are analytic then so does $W(t)$. For complex case, Theorem 1 and 3 in De Moor & Boyd (1989) gives that complex analytic matrices (of a real parameter) have ASVD, then the rest of proof follows.

$\square$

**Remark 10.** *For complex field here, the "analytic" here has **no relation with the standard definition of "complex analytic function"**, who has complex parameters and consequently more restrictions on definition of derivatives.*

*Throughout the proof for gradient flow (continuous time), we only deal with real-valued parameter $t \in \mathbb{R}^+ \cup \{0\}$, so any "analytic" means real-analytic (for $\mathbb{F} = \mathbb{C}$, it means the real and imaginary part are both real-analytic), not complex-analytic.*

**Lemma 22.** *Suppose the analytic singular value decomposition of $M(t)$ exists and is $U(t)\Sigma_M(t)V^H(t)$, $M(t) \in \mathbb{F}^{d\times d}$, where $\mathbb{F} = \mathbb{C}$ or $\mathbb{R}$, then the derivative of the $k^{th}$ singular value is*

$$\frac{\mathrm{d}\sigma_k(M)}{\mathrm{d}t} = \Re\left(u_k^H \frac{\mathrm{d}M}{\mathrm{d}t} v_k\right), \tag{70}$$

*where $u_k$, $v_k$ are the $k^{th}$ column vectors of left and right unitary (or orthogonal if $\mathbb{F} = \mathbb{R}$) matrices respectively.*

*Proof.* We prove the case when $\mathbb{F} = \mathbb{C}$. For $\mathbb{F} = \mathbb{R}$, replace $\cdot^H$ by $\cdot^\top$.

$$\frac{\mathrm{d}M}{\mathrm{d}t} = \frac{\mathrm{d}U}{\mathrm{d}t}\Sigma_M V^H + U\frac{\mathrm{d}\Sigma_M}{\mathrm{d}t}V^H + U\Sigma_M \frac{\mathrm{d}V}{\mathrm{d}t}^H. \tag{71}$$

Then

$$\begin{aligned}
\Re\left(u_k^H \frac{\mathrm{d}M}{\mathrm{d}t} v_k\right) &= \Re\left(u_k^H \frac{\mathrm{d}U}{\mathrm{d}t}\Sigma_M V^H v_k + u_k^H U\frac{\mathrm{d}\Sigma_M}{\mathrm{d}t}V^H v_k + u_k^H U\Sigma_M \frac{\mathrm{d}V}{\mathrm{d}t}^H v_k\right) \\
&= \frac{\mathrm{d}\sigma_k(M)}{\mathrm{d}t} + \sigma_k(M)\left(\Re\left(u_k^H \frac{\mathrm{d}u_k}{\mathrm{d}t}\right) + \Re\left(\frac{\mathrm{d}v_k^H}{\mathrm{d}t} v_k\right)\right).
\end{aligned} \tag{72}$$

From $\Re\left(u_k^H \frac{\mathrm{d}u_k}{\mathrm{d}t}\right) = \frac{\mathrm{d}}{\mathrm{d}t}\left(\frac{1}{2}\|u_k\|^2\right) = 0$, $\Re\left(\frac{\mathrm{d}v_k^H}{\mathrm{d}t} v_k\right) = \frac{\mathrm{d}}{\mathrm{d}t}\left(\frac{1}{2}\|v_k\|^2\right) = 0$, the proof is done.

$\square$

**Remark 11.** *If $M$ is Hermitian, then the $\Re$ can be omitted.*

**Remark 12.** *This generalizes Lemma 2 in Arora et al. (2019b) from real field into complex field by adding a $\Re$ on the right side:*

$$\frac{\mathrm{d}\sigma_r(S)}{\mathrm{d}t} = -N(\sigma_r^2(S))^{1-1/N} \cdot \Re\left(\left\langle \nabla_W \mathcal{L}(W), u_r v_r^H \right\rangle\right). \tag{73}$$

**Lemma 23.** *Under Section 3 with gradient flow, $\mathcal{L}_{\mathrm{ori}}$ is non-increasing.*

*For $t \in [0, +\infty)$,*

$$\frac{\mathrm{d}}{\mathrm{d}t}\mathcal{L}_{\mathrm{ori}} \leq -2N \min_{j,k}|\sigma_k(W_j)|^{2(N-1)}\mathcal{L}_{\mathrm{ori}}. \tag{74}$$

*Proof.* Naturally we have the derivative of product matrix $W(t)$:

$$
\begin{aligned}
\frac{\mathrm{d}W}{\mathrm{d}t} &= \sum_{j=1}^{N} W_{\Pi_L,j+1} \left[ W_{\Pi_L,j+1}^{H} \left( \Sigma - W \right) W_{\Pi_R,j-1}^{H} + a \left( W_j \Delta_{j-1,j} - \Delta_{j,j+1} W_j \right) \right] W_{\Pi_R,j-1} \\
&= \sum_{j=1}^{N} W_{\Pi_L,j+1} W_{\Pi_L,j+1}^{H} \left( \Sigma - W \right) W_{\Pi_R,j-1}^{H} W_{\Pi_R,j-1} \\
&\quad + a \sum_{j=1}^{N} W_{\Pi_L,j} \Delta_{j-1,j} W_{\Pi_R,j-1} - a \sum_{j=1}^{N} W_{\Pi_L,j+1} \Delta_{j,j+1} W_{\Pi_R,j} \\
&= \sum_{j=1}^{N} W_{\Pi_L,j+1} W_{\Pi_L,j+1}^{H} \left( \Sigma - W \right) W_{\Pi_R,j-1}^{H} W_{\Pi_R,j-1} + a \left( W \Delta_{0,1} - \Delta_{N,N+1} W \right) \\
&= \sum_{j=1}^{N} W_{\Pi_L,j+1} W_{\Pi_L,j+1}^{H} \left( \Sigma - W \right) W_{\Pi_R,j-1}^{H} W_{\Pi_R,j-1}.
\end{aligned}
\tag{75}
$$

Then

$$
\begin{aligned}
\frac{\mathrm{d}}{\mathrm{d}t} \mathcal{L}_{\mathrm{ori}} &= -\Re \left( \left\langle \Sigma - W, \frac{\mathrm{d}W}{\mathrm{d}t} \right\rangle \right) \\
&= -\Re \left( \left\langle \Sigma - W, \sum_{j=1}^{N} W_{\Pi_L,j+1} W_{\Pi_L,j+1}^{H} \left( \Sigma - W \right) W_{\Pi_R,j-1}^{H} W_{\Pi_R,j-1} \right\rangle \right) \\
&= -\sum_{j=1}^{N} \Re \left( \left\langle \Sigma - W, W_{\Pi_L,j+1} W_{\Pi_L,j+1}^{H} \left( \Sigma - W \right) W_{\Pi_R,j-1}^{H} W_{\Pi_R,j-1} \right\rangle \right) \\
&= -\sum_{j=1}^{N} \Re \left( \left\langle W_{\Pi_L,j+1}^{H} \left( \Sigma - W \right) W_{\Pi_R,j-1}^{H}, W_{\Pi_L,j+1}^{H} \left( \Sigma - W \right) W_{\Pi_R,j-1}^{H} \right\rangle \right) \\
&= -\sum_{j=1}^{N} \left\| W_{\Pi_L,j+1}^{H} \left( \Sigma - W \right) W_{\Pi_R,j-1}^{H} \right\|_F^2.
\end{aligned}
\tag{76}
$$

From $\|LXR\|_F \geq \sigma_{\min}(L)\sigma_{\min}(R)\|X\|_F$, $\sigma_{\min}\left( W_{\Pi_L,j+1}^{H} \right) \geq \min_{j,k} |\sigma_k(W_j)|^{N-j}$ and $\sigma_{\min}\left( W_{\Pi_R,j-1}^{H} \right) \geq \min_{j,k} |\sigma_k(W_j)|^{j-1}$, the proof is completed.

$\square$

**Lemma 24.** *Analytic singular value decomposition of product matrix with positive semi-definite diagonal matrix.*

*Under Section 3 with gradient flow and any bounded (i.e. $W_{j,j\in\{1,2,\cdots,N\}}(t=0)$ is bounded) balanced initialization, $\forall$ positive integer $N \geq 2$, the product matrix $W(t)$ can be expressed as:*

$$
W(t) = U(t)S(t)V(t)^{H},
\tag{77}
$$

*where: $U(t) \in \mathbb{F}^{d\times d}$, $S(t) \in \mathbb{R}^{d\times d}$ and $V(t) \in \mathbb{F}^{d\times d}$ are analytic functions of $t$, $U(t)$ and $V(t)$ are orthogonal matrices, $S(t)$ is diagonal and positive semi-definite (elements on its diagonal may appear in any order), $\Sigma_w(t) := S(t)^{1/N}$ is well-defined (meaning the real-valued operation $S_{ii} \mapsto (S_{ii})^{1/N}$ is applied to each diagonal element of $S(t)$, resulting in another semi-positive diagonal matrix) and analytic.*

*Moreover, if the singular values of product matrix $W$ are non-zero, then throughout the optimization $W$ remains full rank in finite time.*

*Proof.* From Lemma 21, it is left to construct a new ASVD (analytic singular value decomposition) of $W(t)$ using existed ASVD $W(t) = U(t)S(t)V(t)^H$ ($S(t)$ is not guaranteed to be positive semi-definite).

By Lemma 23, $\|\Sigma - W\|_F \leq \|\Sigma - W(t=0)\|_F$. Then the following term is bounded by a constant for all $t \in \mathbb{R}^+ \cup \{0\}$:

$$
\begin{aligned}
\left|\langle \nabla l(W(t)), u_r(t)v_r(t)^H \rangle\right| &\leq \|\nabla l(W(t))\|_{op} = \|\Sigma - W\|_{op} \\
&\leq \|\Sigma - W\|_F \leq \|\Sigma - W(t=0)\|_F .
\end{aligned}
\tag{78}
$$

By invoking Theorem 3 in Arora et al. (2019b) (for complex case, add $\Re$), the absolute value of time derivative of $\sigma_r(t)$ is bounded by:

$$
\left|\frac{\mathrm{d}\sigma_r(t)}{\mathrm{d}t}\right| \leq \|\Sigma - W(t=0)\|_F \cdot N \left(\sigma_r^2(t)\right)^{1-1/N} .
\tag{79}
$$

Thus all $\sigma_r(t)$ do not change sign for $t \in \mathbb{R}^+ \cup \{0\}$. Moreover, if $|\sigma_r(t=0)| > 0$, the it never decrease to 0 in finite time.

Then we construct $S_{\mathrm{new}}(t)$ by flipping the sign of negative diagonal terms, and $U_{\mathrm{new}}(t)$ by changing the sign of corresponding columns of $U(t)$. Now $W(t) = U_{\mathrm{new}}(t)S_{\mathrm{new}}(t)V(t)^H$ is also an ASVD of $W(t)$, $U_{\mathrm{new}}(t)$ is analytic and unitary (orthogonal), $S_{\mathrm{new}}(t)$ is analytic, diagonal and positive semi-definite.

Specially, if for some $r$, $\sigma_r(t) = 0$ at time $t$, then it remains zero. Thus, from $S_{\mathrm{new}}(t)$ is analytic, so is $\Sigma_w(t)$. This completes the proof.

$\square$

Finally, we generalize Lemma 2 in Arora et al. (2019b) into complex field. Here we assume all matrices are square matrices of dimension $d \times d$.

**Lemma 25.** *Under balanced initialization, assume the singular values of $W(t) = U(t)S(t)V(t)^H$ ($U$, $V$ are unitary, $S$ is real-valued and diagonal) are distinct and different from zero at initialization, then the derivatives of $U$, $V$ satisfy*

$$
\frac{\mathrm{d}U}{\mathrm{d}t} = U\left(F \odot M_U + D_U\right), \frac{\mathrm{d}V}{\mathrm{d}t} = V\left(F \odot M_V + D_V\right),
\tag{80}
$$

*where $D_U$, $D_V$ are diagonal matrices with pure imaginary entries (and thus skew-Hermitian) satisfying*

$$
(D_U)_{jj} - (D_V)_{jj} = -\frac{N}{2}\left(\sigma_j^2(S)\right)^{1/2-1/N}\left[\left(U^H(\nabla_W \mathcal{L}_{\mathrm{ori}})V\right)_{jj} - \left(V^H(\nabla_W \mathcal{L}_{\mathrm{ori}})^H U\right)_{jj}\right],
\tag{81}
$$

*and*

$$
\begin{aligned}
M_U &= -\left[U^H(\nabla_W \mathcal{L}_{\mathrm{ori}})VS + SV^H(\nabla_W \mathcal{L}_{\mathrm{ori}})^H U\right] \\
M_V &= -\left[V^H(\nabla_W \mathcal{L}_{\mathrm{ori}})^H US + SU^H(\nabla_W \mathcal{L}_{\mathrm{ori}})V\right] .
\end{aligned}
\tag{82}
$$

*Here $\odot$ stands for Hadamard (element-wise) product and $F$ is defined by*

$$
F_{jk} = \begin{cases} 0 & , j = k \\ \frac{1}{\left(\sigma_k^2(S)\right)^{1/N} - \left(\sigma_j^2(S)\right)^{1/N}} & , j \neq k. \end{cases}
\tag{83}
$$

**Remark 13.** *Note that only the difference $D_U - D_V$ is uniquely determined. Adding the same purely imaginary diagonal matrix to both $D_U$ and $D_V$ leaves the dynamics of $W$ unchanged, corresponding to a shared phase rotation of $U$ and $V$.*

*For real matrices, R.H.S. of equation (81) is zero, $D_U = D_V = O$, then this Lemma degenerates into Lemma 2 of Arora et al. (2019b).*

*Proof.* We calculate the time derivative of $U$ and the time derivative of $V$ follows the same way.

Following the derivations in Arora et al. (2019b),

$$U^H \frac{\mathrm{d}W}{\mathrm{d}t} V = U^H \frac{\mathrm{d}U}{\mathrm{d}t} S + \frac{\mathrm{d}S}{\mathrm{d}t} + S \frac{\mathrm{d}V}{\mathrm{d}t}^H V, \tag{84}$$

where $U^H \frac{\mathrm{d}U}{\mathrm{d}t} = -\frac{\mathrm{d}U}{\mathrm{d}t}^H U$ and $V^H \frac{\mathrm{d}V}{\mathrm{d}t} = -\frac{\mathrm{d}V}{\mathrm{d}t}^H V$ are skew-Hermitian matrices, whose diagonal entries are therefore purely imaginary. Since $S$ is real, denote $\bar{I}_d$ to be a matrix holding zeros on its diagonal and ones elsewhere,

$$\Re \left( \bar{I}_d \odot \left( U^H \frac{\mathrm{d}W}{\mathrm{d}t} VS + SV^H \frac{\mathrm{d}W}{\mathrm{d}t}^H U \right) \right) = \Re \left( U^H \frac{\mathrm{d}U}{\mathrm{d}t} S^2 - S^2 U^H \frac{\mathrm{d}U}{\mathrm{d}t} \right)$$

$$\Im \left( U^H \frac{\mathrm{d}W}{\mathrm{d}t} VS + SV^H \frac{\mathrm{d}W}{\mathrm{d}t}^H U \right) = \Im \left( U^H \frac{\mathrm{d}U}{\mathrm{d}t} S^2 - S^2 U^H \frac{\mathrm{d}U}{\mathrm{d}t} \right). \tag{85}$$

Since $U^H \frac{\mathrm{d}W}{\mathrm{d}t} VS + SV^H \frac{\mathrm{d}W}{\mathrm{d}t}^H U$ is Hermitian, its diagonal entries are real, further giving $\Im \left( U^H \frac{\mathrm{d}W}{\mathrm{d}t} VS + SV^H \frac{\mathrm{d}W}{\mathrm{d}t}^H U \right) = \Im \left( \bar{I}_d \odot \left( U^H \frac{\mathrm{d}W}{\mathrm{d}t} VS + SV^H \frac{\mathrm{d}W}{\mathrm{d}t}^H U \right) \right)$. Combining the real and imaginary parts gives

$$\bar{I}_d \odot \left( U^H \frac{\mathrm{d}W}{\mathrm{d}t} VS + SV^H \frac{\mathrm{d}W}{\mathrm{d}t}^H U \right) = U^H \frac{\mathrm{d}U}{\mathrm{d}t} S^2 - S^2 U^H \frac{\mathrm{d}U}{\mathrm{d}t}. \tag{86}$$

Here $U^H \frac{\mathrm{d}W}{\mathrm{d}t} V = -\sum_{j=1}^{N} (S^2)^{\frac{j-1}{N}} U^H (\nabla_W \mathcal{L}_{\mathrm{ori}}) V (S^2)^{\frac{N-j}{N}}$. Then the non-diagonal entries of $U^H \frac{\mathrm{d}U}{\mathrm{d}t}$ follows by the proof of Lemma 2 in Arora et al. (2019b).

For the diagonal entries of $U^H \frac{\mathrm{d}U}{\mathrm{d}t}$, by taking imaginary part of equation (84),

$$\sigma_j(S) \left( \left( U^H \frac{\mathrm{d}U}{\mathrm{d}t} \right)_{jj} - \left( V^H \frac{\mathrm{d}V}{\mathrm{d}t} \right)_{jj} \right) = i\Im \left( \sigma_j(S) \left( \left( U^H \frac{\mathrm{d}U}{\mathrm{d}t} \right)_{jj} - \left( V^H \frac{\mathrm{d}V}{\mathrm{d}t} \right)_{jj} \right) \right)$$

$$= i\Im \left( U^H \frac{\mathrm{d}W}{\mathrm{d}t} V \right)_{jj} = \frac{1}{2} \left( \left( U^H \frac{\mathrm{d}W}{\mathrm{d}t} V \right)_{jj} - \left( V^H \frac{\mathrm{d}W}{\mathrm{d}t}^H U \right)_{jj} \right). \tag{87}$$

The last step uses the fact that $i\Im(z) = \frac{1}{2} (z - \bar{z})$. This deduces that

$$\left( U^H \frac{\mathrm{d}U}{\mathrm{d}t} \right)_{jj} - \left( V^H \frac{\mathrm{d}V}{\mathrm{d}t} \right)_{jj} = -\frac{N}{2} \left( \sigma_j^2(S) \right)^{1/2 - 1/N} \left[ \left( U^H (\nabla_W \mathcal{L}_{\mathrm{ori}}) V \right)_{jj} - \left( V^H (\nabla_W \mathcal{L}_{\mathrm{ori}})^H U \right)_{jj} \right]. \tag{88}$$

This completes the proof.

$\square$

### D.4 LEMMAS ON REGULARIZATION, GRADIENT FLOW

**Lemma 26.** *Consider optimizing a generalized loss function coupled with a generalized regularization term using gradient flow:*

$$\mathcal{L}(W_1, \cdots, W_N) := \mathcal{L}_{\text{ori}}\left(\prod_{j=N}^{1} W_j\right) + \frac{1}{4}\sum_{j=1}^{N-1} a_{j,j+1}\|\Delta_{j,j+1}\|_F^2,\ a_{j,j+1} \in \mathbb{R}^+ \cup \{0\}. \quad (89)$$

*Where $\Delta_{j,j+1}$ is defined in (4). Then the regularization terms decays by:*

$$\frac{\mathrm{d}}{\mathrm{d}t}\left(\sum_{j=1}^{N-1} a_{j,j+1}\|\Delta_{j,j+1}\|_F^2\right) = -4\sum_{j=1}^{N}\|a_{j,j+1}\Delta_{j,j+1}W_j - a_{j-1,j}W_j\Delta_{j-1,j}\|_F^2. \quad (90)$$

*Proof.*

$$\begin{aligned}
\frac{\mathrm{d}}{\mathrm{d}t}W_j W_j^H &= -\Big[\left(\nabla_{W_j}\mathcal{L}_{\text{ori}}\right)W_j^H + W_j\left(\nabla_{W_j}\mathcal{L}_{\text{ori}}\right)^H \\
&\quad - 2a_{j-1,j}W_j\Delta_{j-1,j}W_j^H \\
&\quad + a_{j,j+1}\left(\Delta_{j,j+1}W_j W_j^H + W_j W_j^H \Delta_{j,j+1}\right)\Big] \\
\frac{\mathrm{d}}{\mathrm{d}t}W_{j+1}^H W_{j+1} &= -\Big[\left(\nabla_{W_{j+1}}\mathcal{L}_{\text{ori}}\right)^H W_{j+1} + W_{j+1}^H\left(\nabla_{W_{j+1}}\mathcal{L}_{\text{ori}}\right) \\
&\quad + 2a_{j+1,j+2}W_{j+1}^H\Delta_{j+1,j+2}W_{j+1} \\
&\quad - a_{j,j+1}\left(\Delta_{j,j+1}W_{j+1}^H W_{j+1} + W_{j+1}^H W_{j+1}\Delta_{j,j+1}\right)\Big].
\end{aligned} \quad (91)$$

Denote $W_{\prod_L,j} := \prod_{k=N}^{j} W_k$, $W_{\prod_R,j} := \prod_{k=j}^{1} W_k$, $W := \prod_{k=N}^{1} W_k = W_{\prod_L,1} = W_{\prod_R,N}$. From property of the loss $\mathcal{L}_{\text{ori}}$,

$$\left(\nabla_{W_j}\mathcal{L}_{\text{ori}}\right)W_j^H = W_{\prod_L,j+1}^H\left(\nabla_W \mathcal{L}_{\text{ori}}(W)\right)W_{\prod_R,j} = W_{j+1}^H\left(\nabla_{W_{j+1}}\mathcal{L}_{\text{ori}}\right),\ \forall j \in \{1,2,\cdots,N-1\}. \quad (92)$$

Thus we have

$$\begin{aligned}
\frac{\mathrm{d}}{\mathrm{d}t}\Delta_{j,j+1} &= 2a_{j-1,j}W_j\Delta_{j-1,j}W_j^H + 2a_{j+1,j+2}W_{j+1}^H\Delta_{j+1,j+2}W_{j+1} \\
&\quad - a_{j,j+1}\left(\Delta_{j,j+1}\left(W_j W_j^H + W_{j+1}^H W_{j+1}\right) + \left(W_j W_j^H + W_{j+1}^H W_{j+1}\right)\Delta_{j,j+1}\right),
\end{aligned} \quad (93)$$

$$\begin{aligned}
\frac{\mathrm{d}\|\Delta_{j,j+1}\|_F^2}{\mathrm{d}t} &= 4a_{j-1,j}\mathrm{tr}\left(W_j\Delta_{j-1,j}W_j^H\Delta_{j,j+1}\right) \\
&\quad + 4a_{j+1,j+2}\mathrm{tr}\left(W_{j+1}\Delta_{j,j+1}W_{j+1}^H\Delta_{j+1,j+2}\right) \\
&\quad - 4a_{j,j+1}\mathrm{tr}\left((W_j W_j^H + W_{j+1}^H W_{j+1})\Delta_{j,j+1}^2\right) \\
&= -\frac{2}{a_{j,j+1}}\Big[\|a_{j,j+1}\Delta_{j,j+1}W_j - a_{j-1,j}W_j\Delta_{j-1,j}\|_F^2 \\
&\quad + \|a_{j+1,j+2}\Delta_{j+1,j+2}W_{j+1} - a_{j,j+1}W_{j+1}\Delta_{j,j+1}\|_F^2 \\
&\quad + a_{j,j+1}^2\left(\|\Delta_{j,j+1}W_j\|_F^2 + \|W_{j+1}\Delta_{j,j+1}\|_F^2\right) \\
&\quad - a_{j-1,j}^2\|W_j\Delta_{j-1,j}\|_F^2 - a_{j+1,j+2}^2\|\Delta_{j+1,j+2}W_{j+1}\|_F^2\Big].
\end{aligned} \quad (94)$$

By taking weighted sum,

$$\frac{\mathrm{d}}{\mathrm{d}t}\left(\sum_{j=1}^{N-1} a_{j,j+1}\|\Delta_{j,j+1}\|_F^2\right) = -4\sum_{j=1}^{N}\|a_{j,j+1}\Delta_{j,j+1}W_j - a_{j-1,j}W_j\Delta_{j-1,j}\|_F^2. \tag{95}$$

$\square$

Below we back to $a_{j,j+1} \equiv a \in \mathbb{R}^+ \cup \{0\}$, $\forall j \in \{1, 2, \cdots, N-1\}$. Then 89 becomes

$$\mathcal{L}(W_1, \cdots, W_N) \coloneqq \mathcal{L}_{\mathrm{ori}}\left(\prod_{j=N}^{1} W_j\right) + \frac{1}{4}\sum_{j=1}^{N-1} a\|\Delta_{j,j+1}\|_F^2, \ a \in \mathbb{R}^+ \cup \{0\}. \tag{96}$$

**Theorem 27.** *Suppose for all $j \in \{1, 2, \cdots, N\}$, $\sigma_{\min}(W_j) \geq \mu_{\min} > 0$, $\sigma_{\max}(W_j) \leq \mu_{\max}$. Consider optimizing 96 under gradient flow, then the convergence rate of the regularization term is lower bounded:*

$$\frac{\mathrm{d}}{\mathrm{d}t}\left(\sum_{j=1}^{N-1}\|\Delta_{j,j+1}\|_F^2\right) \leq -4a \cdot \frac{2}{N-1}\frac{\mu_{\max}^2 - \mu_{\min}^2}{\left(\frac{\mu_{\max}}{\mu_{\min}}\right)^{2\lfloor N/2\rfloor}-1} \cdot \left(\sum_{j=1}^{N-1}\|\Delta_{j,j+1}\|_F^2\right). \tag{97}$$

*Proof.* Denote $D_j = \Delta_{j,j+1}W_j - W_j\Delta_{j-1,j}$. Then

$$\Delta_{j,j+1} = (D_j + W_j\Delta_{j-1,j})W_j^{-1}. \tag{98}$$

Deducing

$$\|\Delta_{j,j+1}\|_F \leq \left\|W_j^{-1}\right\|_{op}(\|D_j\|_F + \|\Delta_{j-1,j}\|_F\|W_j\|_{op}) \leq \frac{1}{\mu_{\min}}\|D_j\|_F + \frac{\mu_{\max}}{\mu_{\min}}\|\Delta_{j-1,j}\|_F. \tag{99}$$

From $\Delta_{0,1} = O$, inductively we have

$$\|\Delta_{j,j+1}\|_F^2 \leq \frac{1}{\mu_{\min}^2}\left(\sum_{k=1}^{j}\left(\frac{\mu_{\max}}{\mu_{\min}}\right)^{j-k}\|D_k\|_F\right)^2 \leq \frac{1}{\mu_{\min}^2}\left(\sum_{k=1}^{j}\left(\frac{\mu_{\max}}{\mu_{\min}}\right)^{2(j-k)}\right)\left(\sum_{k=1}^{j}\|D_k\|_F^2\right)$$

$$= \frac{1}{\mu_{\min}^2}\frac{\left(\frac{\mu_{\max}}{\mu_{\min}}\right)^{2j}-1}{\left(\frac{\mu_{\max}}{\mu_{\min}}\right)^2-1}\sum_{k=1}^{j}\|D_k\|_F^2. \tag{100}$$

The last two step use Cauchy-Schwarz inequality.

From $\Delta_{N,N+1} = O$, following the same procedure we have

$$\|\Delta_{N-j,N-j+1}\|_F^2 \leq \frac{1}{\mu_{\min}^2}\frac{\left(\frac{\mu_{\max}}{\mu_{\min}}\right)^{2j}-1}{\left(\frac{\mu_{\max}}{\mu_{\min}}\right)^2-1}\sum_{k=N-j+1}^{N}\|D_k\|_F^2. \tag{101}$$

Summing all terms up, for odd $N$ we have

$$\sum_{j=1}^{N-1} \|\Delta_{j,j+1}\|_F^2 = \sum_{j=1}^{(N-1)/2} \left( \|\Delta_{j,j+1}\|_F^2 + \|\Delta_{N-j,N-j+1}\|_F^2 \right)$$

$$\leq \sum_{j=1}^{(N-1)/2} \left( \frac{1}{\mu_{\min}^2} \frac{\left(\frac{\mu_{\max}}{\mu_{\min}}\right)^{2j} - 1}{\left(\frac{\mu_{\max}}{\mu_{\min}}\right)^2 - 1} \sum_{k=1}^{j} \left( \|D_k\|_F^2 + \|D_{N+1-k}\|_F^2 \right) \right)$$

$$= \sum_{k=1}^{(N-1)/2} \left( \left( \|D_k\|_F^2 + \|D_{N+1-k}\|_F^2 \right) \sum_{j=k}^{(N-1)/2} \left( \frac{1}{\mu_{\min}^2} \frac{\left(\frac{\mu_{\max}}{\mu_{\min}}\right)^{2j} - 1}{\left(\frac{\mu_{\max}}{\mu_{\min}}\right)^2 - 1} \right) \right)$$

$$\leq \frac{N-1}{2} \frac{\left(\frac{\mu_{\max}}{\mu_{\min}}\right)^{N-1} - 1}{\mu_{\max}^2 - \mu_{\min}^2} \left( \sum_{k=1}^{N} \|D_k\|^2 \right). \tag{102}$$

For even $N$,

$$\sum_{j=1}^{N-1} \|\Delta_{j,j+1}\|_F^2 = \sum_{j=1}^{N/2-1} \left( \|\Delta_{j,j+1}\|_F^2 + \|\Delta_{N-j,N-j+1}\|_F^2 \right) + \|\Delta_{N/2,N/2+1}\|_F^2$$

$$\leq \sum_{j=1}^{N/2-1} \left( \frac{1}{\mu_{\min}^2} \frac{\left(\frac{\mu_{\max}}{\mu_{\min}}\right)^{2j} - 1}{\left(\frac{\mu_{\max}}{\mu_{\min}}\right)^2 - 1} \sum_{k=1}^{j} \left( \|D_k\|_F^2 + \|D_{N+1-k}\|_F^2 \right) \right)$$

$$+ \frac{1}{2\mu_{\min}^2} \frac{\left(\frac{\mu_{\max}}{\mu_{\min}}\right)^{N} - 1}{\left(\frac{\mu_{\max}}{\mu_{\min}}\right)^2 - 1} \sum_{k=1}^{N/2} \left( \|D_k\|_F^2 + \|D_{N+1-k}\|_F^2 \right)$$

$$= \sum_{k=1}^{N/2-1} \left( \left( \|D_k\|_F^2 + \|D_{N+1-k}\|_F^2 \right) \sum_{j=k}^{N/2-1} \left( \frac{1}{\mu_{\min}^2} \frac{\left(\frac{\mu_{\max}}{\mu_{\min}}\right)^{2j} - 1}{\left(\frac{\mu_{\max}}{\mu_{\min}}\right)^2 - 1} \right) \right) \tag{103}$$

$$+ \frac{1}{2\mu_{\min}^2} \frac{\left(\frac{\mu_{\max}}{\mu_{\min}}\right)^{N} - 1}{\left(\frac{\mu_{\max}}{\mu_{\min}}\right)^2 - 1} \sum_{k=1}^{N/2} \left( \|D_k\|_F^2 + \|D_{N+1-k}\|_F^2 \right)$$

$$\leq \frac{N-1}{2} \frac{\left(\frac{\mu_{\max}}{\mu_{\min}}\right)^{N} - 1}{\mu_{\max}^2 - \mu_{\min}^2} \left( \sum_{k=1}^{N} \|D_k\|^2 \right).$$

Thus

$$\sum_{j=1}^{N} \|D_j\|^2 \geq \frac{2}{N-1} \frac{\mu_{\max}^2 - \mu_{\min}^2}{\left(\frac{\mu_{\max}}{\mu_{\min}}\right)^{2\lfloor N/2 \rfloor} - 1} \sum_{i=1}^{N-1} \|\Delta_{i,i+1}\|_F^2. \tag{104}$$

Combine with Lemma 26, then the proof is done.

$\square$

**Remark 14.** *For $N = 4$, Theorem 27 reduces to*

$$\frac{\mathrm{d}}{\mathrm{d}t}\left(\sum_{j=1}^{3}\|\Delta_{j,j+1}\|_F^2\right) \leq -\frac{8a}{3}\frac{\mu_{\min}^4}{\mu_{\max}^2+\mu_{\min}^2}\cdot\left(\sum_{j=1}^{3}\|\Delta_{j,j+1}\|_F^2\right). \tag{105}$$

**Theorem 28.** *Under problem settings in section 3 with gradient flow, the change of maximum and minimum singular values of $W_j$s have bounds that are irrelevant to the regularization term:*

$$\frac{\mathrm{d}\max_{j,k}\sigma_k^2(W_j)}{\mathrm{d}t} \leq 2\max_{j,k}|\sigma_k(W_j)|\max_j\left\|\nabla_{W_j}\mathcal{L}_{\mathrm{ori}}\right\|_{op}$$
$$\frac{\mathrm{d}\min_{j,k}\sigma_k^2(W_j)}{\mathrm{d}t} \geq -2\min_{j,k}|\sigma_k(W_j)|\max_j\left\|\nabla_{W_j}\mathcal{L}_{\mathrm{ori}}\right\|_{op}. \tag{106}$$

**Remark 15.** *If $\arg\max_{(j,k)}|\sigma_k(W_j)|$, $\arg\min_{(j,k)}|\sigma_k(W_j)|$ are not unique, the derivatives are not well-defined. In these cases, the inequalities become:*

$$\frac{\mathrm{d}\sigma_{k'}^2(W_{j'})}{\mathrm{d}t} \leq 2\max_{j,k}|\sigma_k(W_j)|\max_j\left\|\nabla_{W_j}\mathcal{L}_{\mathrm{ori}}\right\|_{op}, (j',k')\in\arg\max_{(j,k)}|\sigma_k(W_j)|$$
$$\frac{\mathrm{d}\sigma_{k'}^2(W_{j'})}{\mathrm{d}t} \geq -2\min_{j,k}|\sigma_k(W_j)|\max_j\left\|\nabla_{W_j}\mathcal{L}_{\mathrm{ori}}\right\|_{op}, (j',k')\in\arg\min_{(j,k)}|\sigma_k(W_j)|. \tag{107}$$

*Proof.* For simplicity, set $W_0\equiv W_1$, $W_5\equiv W_4$.

Denote the analytic singular value decomposition of $W_j(t)$ to be $U^{(j)}\Sigma_w^{(j)}V^{(j)H}$, then from Lemma 22, we have

$$\begin{aligned}\frac{\mathrm{d}\sigma_k(W_j)}{\mathrm{d}t} &= \Re\left(u_k^{(j)H}\left(-\nabla_{W_j}\mathcal{L}_{\mathrm{ori}}+aW_j\Delta_{j-1,j}-a\Delta_{j,j+1}W_j\right)v_k^{(j)}\right)\\ &= \Re\left(u_k^{(j)H}\left(-\nabla_{W_j}\mathcal{L}_{\mathrm{ori}}\right)v_k^{(j)}\right)\\ &\quad+au_k^{(j)H}\left(W_jW_{j-1}W_{j-1}^H+W_{j+1}^HW_{j+1}W_j-2W_jW_j^HW_j\right)v_k^{(j)}\\ &= \Re\left(u_k^{(j)H}\left(-\nabla_{W_j}\mathcal{L}_{\mathrm{ori}}\right)v_k^{(j)}\right)\\ &\quad+a\left[\left(u_k^{(j)H}W_{j+1}^HW_{j+1}u_k^{(j)}+v_k^{(j)H}W_{j-1}W_{j-1}^Hv_k^{(j)}\right)\sigma_k(W_j)-2\sigma_k(W_j)^3\right].\end{aligned} \tag{108}$$

From $u_k^{(j)H}W_{j+1}^HW_{j+1}u_k^{(j)}$, $v_k^{(j)H}W_{j-1}W_{j-1}^Hv_k^{(j)}\in[\min_{j,k}\sigma_k^2(W_j),\max_{j,k}\sigma_k^2(W_j)]$, the proof is completed. $\qquad\square$

Note:

$$\max_j\left\|\nabla_{W_j}\mathcal{L}_{\mathrm{ori}}\right\|_{op} \leq \max_{j,k}|\sigma_k(W_j)|^{N-1}\left(\sigma_1(\Sigma)+\max_{j,k}|\sigma_k(W_j)|^N\right). \tag{109}$$

## D.5 LEMMAS ON REGULARIZATION, GRADIENT DESCENT

**Theorem 29.** *Suppose for all $j\in\{1,2,3,4\}$, $\sigma_{\min}(W_j(t))\geq\mu_{\min}>0$, $\sigma_{\max}(W_j(t))\leq\mu_{\max}$, then the convergence rate of the regularization term is lower bounded by:*

$$\mathcal{L}_{\text{reg}}(t+1) \leq \left(1 - \frac{8}{3}\frac{\eta a \mu_{\min}^4}{\mu_{\max}^2 + \mu_{\min}^2}\right) \cdot \mathcal{L}_{\text{reg}}(t)$$
$$+ \eta^2 O\left(a^2 \mu_{\max}^4 \mathcal{L}_{\text{reg}}(t) + \sqrt{a\mathcal{L}_{\text{reg}}(t)}\mu_{\max}^6 \mathcal{L}_{\text{ori}}(t)\right) \quad (110)$$
$$+ \eta^4 O\left(a\mu_{\max}^{12}\mathcal{L}_{\text{ori}}(t)^2 + a^3 \mu_{\max}^4 \mathcal{L}_{\text{reg}}(t)^2\right).$$

*Proof.*
$$\Delta_{j,j+1}(t+1) - \Delta_{j,j+1}(t) = 2\eta a W_j(t)\Delta_{j-1,j}(t)W_j(t)^H$$
$$+ 2\eta a W_{j+1}(t)^H \Delta_{j+1,j+2}(t)W_{j+1}(t)$$
$$- \eta a \Delta_{j,j+1}(t)\left(W_j(t)W_j(t)^H + W_{j+1}(t)^H W_{j+1}(t)\right) \quad (111)$$
$$- \eta a \left(W_j(t)W_j(t)^H + W_{j+1}(t)^H W_{j+1}(t)\right)\Delta_{j,j+1}(t)$$
$$+ \eta^2 \left[\nabla_{W_j}\mathcal{L}(t)\nabla_{W_j}\mathcal{L}(t)^H - \nabla_{W_{j+1}}\mathcal{L}(t)^H \nabla_{W_{j+1}}\mathcal{L}(t)\right].$$

From

$$\left\|\nabla_{W_j}\mathcal{L}(t)\right\|_F \leq \left\|\nabla_{W_j}\mathcal{L}_{\text{ori}}(t)\right\|_F + \left\|\nabla_{W_j}\mathcal{L}_{\text{reg}}(t)\right\|_F$$
$$= O\left(\mu_{\max}^3 \sqrt{\mathcal{L}_{\text{ori}}(t)} + \mu_{\max}\sqrt{a\mathcal{L}_{\text{reg}}(t)}\right)$$
$$\left\|\Delta_{j,j+1}(t+1) - \Delta_{j,j+1}(t)\right\|_F = O\left(\eta\mu_{\max}^2 \sqrt{a\mathcal{L}_{\text{reg}}(t)} + \eta^2 \left\|\nabla_{W_j}\mathcal{L}(t)\right\|_F^2\right)$$
$$= O\left(\eta\mu_{\max}^2 \sqrt{a\mathcal{L}_{\text{reg}}(t)} + \eta^2 \mu_{\max}^6 \mathcal{L}_{\text{ori}}(t) + \eta^2 a\mu_{\max}^2 \mathcal{L}_{\text{reg}}(t)\right).$$
$$(112)$$

We have

$$\mathcal{L}_{\text{reg}}(t+1) - \mathcal{L}_{\text{reg}}(t) = 2a\sum_{j=1}^{3}\langle\Delta_{j,j+1}(t+1) - \Delta_{j,j+1}(t), \Delta_{j,j+1}(t)\rangle$$
$$+ a\sum_{j=1}^{3}\|\Delta_{j,j+1}(t+1) - \Delta_{j,j+1}(t)\|_F^2$$
$$= -4\eta a^2 \sum_{j=1}^{4}\|\Delta_{j,j+1}(t)W_j(t) - W_j(t)\Delta_{j-1,j}(t)\|_F^2$$
$$+ O\left(\eta^2 \sqrt{a\mathcal{L}_{\text{reg}}(t)}\left(a\mu_{\max}^2 \mathcal{L}_{\text{reg}}(t) + \mu_{\max}^6 \mathcal{L}_{\text{ori}}(t)\right)\right) \quad (113)$$
$$+ O\left(\eta^2 a^2 \mu_{\max}^4 \mathcal{L}_{\text{reg}}(t) + \eta^4 a\mu_{\max}^{12}\mathcal{L}_{\text{ori}}(t)^2 + \eta^4 a^3 \mu_{\max}^4 \mathcal{L}_{\text{reg}}(t)^2\right)$$
$$= -4\eta a^2 \sum_{j=1}^{4}\|\Delta_{j,j+1}(t)W_j(t) - W_j(t)\Delta_{j-1,j}(t)\|_F^2$$
$$+ \eta^2 O\left(a^2 \mu_{\max}^4 \mathcal{L}_{\text{reg}}(t) + \sqrt{a\mathcal{L}_{\text{reg}}(t)}\mu_{\max}^6 \mathcal{L}_{\text{ori}}(t)\right)$$
$$+ \eta^4 O\left(a\mu_{\max}^{12}\mathcal{L}_{\text{ori}}(t)^2 + a^3 \mu_{\max}^4 \mathcal{L}_{\text{reg}}(t)^2\right).$$

Follow previous analysis in continuous case,

$$\sum_{j=1}^{4}\|\Delta_{j,j+1}(t)W_j(t) - W_j(t)\Delta_{j-1,j}(t)\|^2 \geq \frac{2}{3}\frac{\mu_{\min}^4}{\mu_{\max}^2 + \mu_{\min}^2}\sum_{i=1}^{3}\|\Delta_{i,i+1}(t)\|_F^2. \quad (114)$$

Then the proof is done.

$\square$

**Theorem 30.** *The maximum and minimum singular values of $W_j$s are irrelevant to the regularization term.*

*Under $\eta \leq \min\left(\frac{1}{18a \max_{j,k} \sigma_k^2(W_j(t))}, \frac{\min_{j,k} \sigma_k(W_j(t))}{3 \max_j \left\|\nabla_{W_j}\mathcal{L}_{\mathrm{ori}}(t)\right\|_{op}}\right),$*

$$
\begin{aligned}
\max_{j,k} \sigma_k^2(W_j(t+1)) - \max_{j,k} \sigma_k^2(W_j(t)) &\leq 2\eta \max_{j,k} \sigma_k(W_j(t)) \max_j \left\|\nabla_{W_j}\mathcal{L}_{\mathrm{ori}}(t)\right\|_{op} \\
&+ \eta^2 O\left(\left\|\nabla_{W_j}\mathcal{L}_{\mathrm{ori}}(t)\right\|_{op}^2 + a^2 \max_{j,k} \sigma_k^6(W_j(t))\right) \\
\min_{j,k} \sigma_k^2(W_j(t+1)) - \min_{j,k} \sigma_k^2(W_j(t)) &\geq -2\eta \min_{j,k} \sigma_k(W_j(t)) \max_j \left\|\nabla_{W_j}\mathcal{L}_{\mathrm{ori}}(t)\right\|_{op} \\
&+ \eta^2 O\left(\left\|\nabla_{W_j}\mathcal{L}_{\mathrm{ori}}(t)\right\|_{op}^2 + a^2 \max_{j,k} \sigma_k^6(W_j(t))\right).
\end{aligned}
\tag{115}
$$

*Proof.* For simplicity, set $W_0 \equiv W_1$, $W_5 \equiv W_4$.

Generally,

$$
\begin{aligned}
W_j(t+1)W_j(t+1)^H &= W_j(t)W_j(t)^H - \eta W_j(t)\nabla_{W_j}\mathcal{L}(t)^H - \eta \nabla_{W_j}\mathcal{L}(t)W_j(t)^H \\
&+ \eta^2 \nabla_{W_j}\mathcal{L}(t)\nabla_{W_j}\mathcal{L}(t)^H \\
&= W_j(t)W_j(t)^H - \eta W_j(t)\nabla_{W_j}\mathcal{L}_{\mathrm{ori}}(t)^H - \eta \nabla_{W_j}\mathcal{L}_{\mathrm{ori}}(t)W_j(t)^H \\
&+ 2\eta a W_j(t)\Delta_{j-1,j}(t)W_j(t)^H - \eta a W_j(t)W_j(t)^H\Delta_{j,j+1}(t) \\
&- \eta a \Delta_{j,j+1}(t)W_j(t)W_j(t)^H + \eta^2 \nabla_{W_j}\mathcal{L}(t)\nabla_{W_j}\mathcal{L}(t)^H \\
&= \frac{1}{3}W_j(t)\left(I + 3\eta a \Delta_{j-1,j}(t)\right)^2 W_j(t)^H \\
&+ \frac{1}{3}\left(I - 3\eta a \Delta_{j,j+1}(t)\right)W_j(t)W_j(t)^H\left(I - 3\eta a \Delta_{j,j+1}(t)\right) \\
&+ \frac{1}{3}\left(W_j(t) - 3\eta \nabla_{W_j}\mathcal{L}_{\mathrm{ori}}(t)\right)\left(W_j(t) - 3\eta \nabla_{W_j}\mathcal{L}_{\mathrm{ori}}(t)\right)^H \\
&+ \eta^2 \nabla_{W_j}\mathcal{L}(t)\nabla_{W_j}\mathcal{L}(t)^H - 3\eta^2 \nabla_{W_j}\mathcal{L}_{\mathrm{ori}}(t)\nabla_{W_j}\mathcal{L}_{\mathrm{ori}}(t)^H \\
&- 3\eta^2 a^2 W_j(t)\Delta_{j-1,j}(t)^2 W_j(t)^H \\
&- 3\eta^2 a^2 \Delta_{j,j+1}(t)W_j(t)W_j(t)^H\Delta_{j,j+1}(t).
\end{aligned}
\tag{116}
$$

Notice that $W_j(t)\left(I + 3\eta a \Delta_{j-1,j}(t)\right)^2 W_j(t)^H$ and $\left(I + 3\eta a \Delta_{j-1,j}(t)\right)W_j(t)^H W_j(t)\left(I + 3\eta a \Delta_{j-1,j}(t)\right)$ shares the same eigenvalues. Then from Lemma 19, the maximum and minimum singular values of $W_j(t+1)$ satisfy

$$\sigma_{\max}^2(W_j(t+1)) \leq \frac{1}{3}\sigma_{\max}^2(W_j(t)) \left[1 + 3\eta a \left(\sigma_{\max}^2(W_{j-1}(t)) - \sigma_{\max}^2(W_j(t))\right)\right]^2$$

$$+ \frac{1}{3}\sigma_{\max}^2(W_j(t)) \left[1 + 3\eta a \left(\sigma_{\max}^2(W_{j+1}(t)) - \sigma_{\max}^2(W_j(t))\right)\right]^2$$

$$+ \frac{1}{3}\left[\sigma_{\max}(W_j(t)) + 3\eta \left\|\nabla_{W_j}\mathcal{L}_{\text{ori}}(t)\right\|_{op}\right]^2$$

$$+ \eta^2 O \left(\left\|\nabla_{W_j}\mathcal{L}_{\text{ori}}(t)\right\|_{op}^2 + a^2 \max_{j,k} \sigma_k^6(W_j(t))\right)$$

$$= \sigma_{\max}^2(W_j(t)) \left[1 + 3\eta a \left(\sigma_{\max}^2(W_{j+1}(t)) + \sigma_{\max}^2(W_{j-1}(t)) - 2\sigma_{\max}^2(W_j(t))\right)\right]$$

$$+ 2\eta\sigma_{\max}(W_j(t)) \left\|\nabla_{W_j}\mathcal{L}_{\text{ori}}(t)\right\|_{op} + \eta^2 O \left(\left\|\nabla_{W_j}\mathcal{L}_{\text{ori}}(t)\right\|_{op}^2 + a^2 \max_{j,k} \sigma_k^6(W_j(t))\right)$$

$$\sigma_{\min}^2(W_j(t+1)) \geq \frac{1}{3}\sigma_{\min}^2(W_j(t)) \left[1 + 3\eta a \left(\sigma_{\min}^2(W_{j-1}(t)) - \sigma_{\min}^2(W_j(t))\right)\right]^2$$

$$+ \frac{1}{3}\sigma_{\min}^2(W_j(t)) \left[1 + 3\eta a \left(\sigma_{\min}^2(W_{j+1}(t)) - \sigma_{\min}^2(W_j(t))\right)\right]^2$$

$$+ \frac{1}{3}\left[\sigma_{\min}(W_j(t)) - 3\eta \left\|\nabla_{W_j}\mathcal{L}_{\text{ori}}(t)\right\|_{op}\right]^2$$

$$+ \eta^2 O \left(\left\|\nabla_{W_j}\mathcal{L}_{\text{ori}}(t)\right\|_{op}^2 + a^2 \max_{j,k} \sigma_k^6(W_j(t))\right)$$

$$= \sigma_{\min}^2(W_j(t)) \left[1 + 3\eta a \left(\sigma_{\min}^2(W_{j+1}(t)) + \sigma_{\min}^2(W_{j-1}(t)) - 2\sigma_{\min}^2(W_j(t))\right)\right]$$

$$- 2\eta\sigma_{\min}(W_j(t)) \left\|\nabla_{W_j}\mathcal{L}_{\text{ori}}(t)\right\|_{op} + \eta^2 O \left(\left\|\nabla_{W_j}\mathcal{L}_{\text{ori}}(t)\right\|_{op}^2 + a^2 \max_{j,k} \sigma_k^6(W_j(t))\right).$$

$$(117)$$

By taking maximum and minimum over $j \in \{1,2,3,4\}$ (for $\eta \leq \frac{1}{6a \max_{j,k} \sigma_k^2(W_j(t))}$, the first term of R.H.S can be upper bounded by $\max_{j,k} \sigma_k^2(W_j(t))$ or lower bounded by $\min_{j,k} \sigma_k^2(W_j(t))$ respectively), the proof is completed.

$\square$

## E DYNAMICS UNDER BALANCED INITIALIZATION

This section analyzes the training dynamics under balanced initialization.

At the beginning, We derive some properties from Lemma 24. Under balanced condition,

$$W_{\prod_L,j} W_{\prod_L,j}^H = \left(\prod_{k=N}^{j} W_k\right) \left(\prod_{k=N}^{j} W_k\right)^H = \left(W_N W_N^H\right)^{N-j+1}$$

$$W_{\prod_R,j}^H W_{\prod_R,j} = \left(\prod_{k=j}^{1} W_k\right)^H \left(\prod_{k=j}^{1} W_k\right) = \left(W_1^H W_1\right)^{N-j+1}.$$

$$(118)$$

Consider $j = 1$ and $j = N$, then

$$W_N W_N^H = \left(WW^H\right)^{1/N} = U\Sigma_w^2 U^H$$

$$W_1 W_1^H = \left(W^H W\right)^{1/N} = V\Sigma_w^2 V^H.$$

$$(119)$$

Suppose the non-negative ASVD of product matrix is $W = U\Sigma_w^N V^H$, then

$$\frac{\mathrm{d}}{\mathrm{d}t}\left(U\Sigma_w^2 U^H\right) = \frac{\mathrm{d}}{\mathrm{d}t}\left(W_N W_N^H\right) = \Sigma V\Sigma_w^N U^H + U\Sigma_w^N V^H \Sigma^H - 2U\Sigma_w^{2N} U^H$$

$$\frac{\mathrm{d}}{\mathrm{d}t}\left(V\Sigma_w^2 V^H\right) = \frac{\mathrm{d}}{\mathrm{d}t}\left(W_1^H W_1\right) = V\Sigma_w^N U^H \Sigma + \Sigma^H U\Sigma_w^N V^H - 2V\Sigma_w^{2N} V^H \tag{120}$$

$$\frac{\mathrm{d}W}{\mathrm{d}t} = \sum_{j=1}^{N} U\Sigma_w^{2(j-1)} U^H \Sigma V\Sigma_w^{2(N-j)} V^H - NU\Sigma_w^{3N-2} V^H.$$

The dynamics of $\sigma_r := \sigma_{w,r}^N$ is presented in (73).

### E.1 SKEW-HERMITIAN ERROR

This section formally state and prove Theorem 4.

**Theorem 31.** *The skew-Hermitian error is non-increasing.*

*Under balanced Gaussian initialization, for $\mathbb{F} = \mathbb{C}$ or $\mathbb{R}$, suppose the ASVD of the product matrix is $W(t) = U(t)\Sigma_w(t)^N V(t)^H$, furthermore assume that the singular values of the product matrix at initialization $(W(0))$ are distinct and different from zero (refer to Lemma 2 in Arora et al. (2019b)).*

*Denote $\sigma_{w,j} = (\Sigma_w)_{jj}$, $U' = \Sigma^{1/2} U$, $V' = \Sigma^{1/2} V$, $u'_j$ and $v'_j$ are the $j^{th}$ columns of $U'$ and $V'$ respectively, then*

$$\frac{\mathrm{d}}{\mathrm{d}t}\|\Sigma^{1/2}(U-V)\Sigma_w\|_F^2 = -2\sum_j \sigma_{w,j}^N \cdot \left\|\Sigma^{1/2}\left(u'_j - v'_j\right)\right\|^2 - 2\sum_j \sigma_{w,j}^{2N} \cdot \left\|u'_j - v'_j\right\|^2$$

$$- \sum_{j,k} f_N\left(\sigma_{w,j}, \sigma_{w,k}\right)\left|{u'_j}^H v'_k - {v'_j}^H u'_k\right|^2 \tag{121}$$

$$\leq 0,$$

*where $f_N(x,y) = \begin{cases} \frac{x^2 y^2 (x^{N-2} - y^{N-2})}{x^2 - y^2} & , y \neq x \\ \frac{N-2}{2} x^N & , y = x \end{cases}$ is a non-negative real-analytic function on $[0, +\infty)^2$.*

*Proof.* By (73)

$$\frac{\mathrm{d}\sigma_{w,j}}{\mathrm{d}t} = \sigma_{w,j}^{N-1}\left(\frac{\langle u'_j, v'_j\rangle + \langle v'_j, u'_j\rangle}{2} - \sigma_{w,j}^N\right). \tag{122}$$

From Lemma 25,

$$\frac{\mathrm{d}U}{\mathrm{d}t} = U\left(F \odot M_U + D_U\right), \frac{\mathrm{d}V}{\mathrm{d}t} = V\left(F \odot M_V + D_V\right), \tag{123}$$

where

$$\begin{cases} (M_U)_{jk} = \langle v'_k, u'_j\rangle \sigma_{w,k}^N + \langle u'_k, v'_j\rangle \sigma_{w,j}^N - 2\sigma_{w,j}^{2N}\delta_{j,k} \\ (M_V)_{jk} = \langle u'_k, v'_j\rangle \sigma_{w,k}^N + \langle v'_k, u'_j\rangle \sigma_{w,j}^N - 2\sigma_{w,j}^{2N}\delta_{j,k} \end{cases}, \tag{124}$$

$D_{U,V}$ are pure imaginary diagonal matrices defined by

$$(D_U)_{jj} - (D_V)_{jj} = \frac{N}{2}\sigma_{w,j}^{N-2}\left[\langle v'_j, u'_j\rangle - \langle u'_j, v'_j\rangle\right], \Re(D_U) = \Re(D_V) = O. \tag{125}$$

Here $\langle a, b\rangle := b^H a$ follows the standard definition of (complex) inner product. Then

$$\frac{\mathrm{d}U'^H V'}{\mathrm{d}t} = \frac{\mathrm{d}U}{\mathrm{d}t}^H \Sigma V + U^H \Sigma \frac{\mathrm{d}V}{\mathrm{d}t} = \left(F^H \odot M_U^H - D_U\right) U^H \Sigma V + U^H \Sigma V (F \odot M_V + D_V)$$

$$\frac{\mathrm{d}U'^H U'}{\mathrm{d}t} = \frac{\mathrm{d}U}{\mathrm{d}t}^H \Sigma U + U^H \Sigma \frac{\mathrm{d}U}{\mathrm{d}t} = \left(F^H \odot M_U^H - D_U\right) U^H \Sigma U + U^H \Sigma U (F \odot M_U + D_U)$$

$$\frac{\mathrm{d}V'^H V'}{\mathrm{d}t} = \frac{\mathrm{d}V}{\mathrm{d}t}^H \Sigma V + V^H \Sigma \frac{\mathrm{d}V}{\mathrm{d}t} = \left(F^H \odot M_V^H - D_V\right) V^H \Sigma V + V^H \Sigma V (F \odot M_V + D_V). \tag{126}$$

For each diagonal entry,

$$\frac{\mathrm{d}}{\mathrm{d}t} \left\langle v_j', u_j' \right\rangle = \left(\frac{\mathrm{d}U'^H V'}{\mathrm{d}t}\right)_{jj}$$

$$= -\frac{N}{2} \sigma_{w,j}^{N-2} \left\langle v_j', u_j' \right\rangle \left[\left\langle v_j', u_j' \right\rangle - \left\langle u_j', v_j' \right\rangle\right]$$

$$+ \sum_{k \neq j} \frac{1}{\sigma_{w,j}^2 - \sigma_{w,k}^2} \left[\left(\left|\left\langle u_j', v_k' \right\rangle\right|^2 + \left|\left\langle u_k', v_j' \right\rangle\right|^2\right) \sigma_{w,j}^N + 2 \left\langle v_k', u_j' \right\rangle \left\langle v_j', u_k' \right\rangle \sigma_{w,k}^N\right]$$

$$\frac{\mathrm{d}}{\mathrm{d}t} \left\langle u_j', u_j' \right\rangle = \left(\frac{\mathrm{d}U'^H U'}{\mathrm{d}t}\right)_{jj}$$

$$= \sum_{k \neq j} \frac{1}{\sigma_{w,j}^2 - \sigma_{w,k}^2} \left[\left(\left\langle u_k', v_j' \right\rangle \left\langle u_j', u_k' \right\rangle + \left\langle u_k', u_j' \right\rangle \left\langle v_j', u_k' \right\rangle\right) \sigma_{w,j}^N \right. \tag{127}$$

$$\left. + \left(\left\langle v_k', u_j' \right\rangle \left\langle u_j', u_k' \right\rangle + \left\langle u_k', u_j' \right\rangle \left\langle u_j', v_k' \right\rangle\right) \sigma_{w,k}^N\right]$$

$$\frac{\mathrm{d}}{\mathrm{d}t} \left\langle v_j', v_j' \right\rangle = \left(\frac{\mathrm{d}V'^H V'}{\mathrm{d}t}\right)_{jj}$$

$$= \sum_{k \neq j} \frac{1}{\sigma_{w,j}^2 - \sigma_{w,k}^2} \left[\left(\left\langle v_k', u_j' \right\rangle \left\langle v_j', v_k' \right\rangle + \left\langle v_k', v_j' \right\rangle \left\langle u_j', v_k' \right\rangle\right) \sigma_{w,j}^N\right.$$

$$\left. + \left(\left\langle u_k', v_j' \right\rangle \left\langle v_j', v_k' \right\rangle + \left\langle v_k', v_j' \right\rangle \left\langle v_j', u_k' \right\rangle\right) \sigma_{w,k}^N\right].$$

Notice that for the second and third equation, $D_U, D_V$ terms cancel out with each other. This further gives

$$\frac{\mathrm{d}}{\mathrm{d}t} \left\|u_j' - v_j'\right\|^2$$

$$= \frac{N}{2} \sigma_{w,j}^{N-2} \left[\left\langle v_j', u_j' \right\rangle - \left\langle u_j', v_j' \right\rangle\right]^2$$

$$+ \sum_{k \neq j} \frac{\sigma_{w,j}^N}{\sigma_{w,j}^2 - \sigma_{w,k}^2} \cdot \left[-2\left(\left|\left\langle u_j', v_k' \right\rangle\right|^2 + \left|\left\langle u_k', v_j' \right\rangle\right|^2\right)\right.$$

$$\left. + \left(\left\langle u_k', v_j' \right\rangle \left\langle u_j', u_k' \right\rangle + \left\langle u_k', u_j' \right\rangle \left\langle v_j', u_k' \right\rangle\right) + \left(\left\langle v_k', u_j' \right\rangle \left\langle v_j', v_k' \right\rangle + \left\langle v_k', v_j' \right\rangle \left\langle u_j', v_k' \right\rangle\right)\right] \tag{128}$$

$$+ \sum_{k \neq j} \frac{\sigma_{w,k}^N}{\sigma_{w,j}^2 - \sigma_{w,k}^2} \cdot \left[-2\left(\left\langle v_k', u_j' \right\rangle \left\langle v_j', u_k' \right\rangle + \left\langle u_j', v_k' \right\rangle \left\langle u_k', v_j' \right\rangle\right)\right.$$

$$\left. + \left(\left\langle v_k', u_j' \right\rangle \left\langle u_j', u_k' \right\rangle + \left\langle u_k', u_j' \right\rangle \left\langle u_j', v_k' \right\rangle\right) + \left(\left\langle u_k', v_j' \right\rangle \left\langle v_j', v_k' \right\rangle + \left\langle v_k', v_j' \right\rangle \left\langle v_j', u_k' \right\rangle\right)\right].$$

For the L.H.S. of (121),

$$\frac{\mathrm{d}}{\mathrm{d}t} \left\|(U' - V')\Sigma_w\right\|_F^2 = \sum_j \left\|u_j' - v_j'\right\|^2 \frac{\mathrm{d}}{\mathrm{d}t} \sigma_{w,j}^2 + \sum_j \sigma_{w,j}^2 \frac{\mathrm{d}}{\mathrm{d}t} \left\|u_j' - v_j'\right\|^2. \tag{129}$$

The first term can be written by

$$
\begin{aligned}
&\sum_j \left\| u_j' - v_j' \right\|^2 \frac{\mathrm{d}}{\mathrm{d}t} \sigma_{w,j}^2 \\
=&\sum_j \sigma_{w,j}^N \left( \langle u_j', v_j' \rangle + \langle v_j', u_j' \rangle - 2\sigma_{w,j}^N \right) \left\| u_j' - v_j' \right\|^2 \\
=&\sum_j \sigma_{w,j}^N \left( u_j'^{H} u_j' u_j'^{H} v_j' + v_j'^{H} u_j' u_j'^{H} u_j' + u_j'^{H} v_j' v_j'^{H} v_j' + v_j'^{H} v_j' v_j'^{H} u_j' \right) \\
&-\sum_j \sigma_{w,j}^N \left( u_j'^{H} v_j' + v_j'^{H} u_j' \right)^2 - 2\sum_j \sigma_{w,j}^{2N} \cdot \left\| u_j' - v_j' \right\|^2 .
\end{aligned}
\tag{130}
$$

For the second term,

$$
\begin{aligned}
&\sum_j \sigma_{w,j}^2 \frac{\mathrm{d}}{\mathrm{d}t} \left\| u_j' - v_j' \right\|^2 \\
=&\frac{1}{2} \left( \sum_j \sigma_{w,j}^2 \frac{\mathrm{d}}{\mathrm{d}t} \left\| u_j' - v_j' \right\|^2 + \sum_k \sigma_{w,k}^2 \frac{\mathrm{d}}{\mathrm{d}t} \left\| u_k' - v_k' \right\|^2 \right) \\
=&\frac{N}{2} \sum_j \sigma_{w,j}^N \left[ \langle v_j', u_j' \rangle - \langle u_j', v_j' \rangle \right]^2 \\
&-\sum_{j,k,j\neq k} \frac{\sigma_{w,j}^2 \sigma_{w,k}^2 \left( \sigma_{w,j}^{N-2} - \sigma_{w,k}^{N-2} \right)}{\sigma_{w,j}^2 - \sigma_{w,k}^2} \left| \langle v_k', u_j' \rangle - \langle u_k', v_j' \rangle \right|^2 \\
&-2\sum_{j,k,j\neq k} \sigma_{w,j}^N \cdot \left( \left| \langle u_j', v_k' \rangle \right|^2 + \left| \langle u_k', v_j' \rangle \right|^2 \right) \\
&+2\sum_{j,k,j\neq k} \sigma_{w,j}^N \cdot \Re \left( \langle u_k', v_j' \rangle \langle u_j', u_k' \rangle + \langle u_j', v_k' \rangle \langle v_k', v_j' \rangle \right) .
\end{aligned}
\tag{131}
$$

Notice that

$$
\left[ \langle v_j', u_j' \rangle - \langle u_j', v_j' \rangle \right]^2 = 4 \left[ i\Im \left( \langle v_j', u_j' \rangle \right) \right]^2 = -\left| u_j'^{H} v_j' - v_j'^{H} u_j' \right|^2 ,
\tag{132}
$$

$$
\begin{aligned}
&-\sum_j \sigma_{w,j}^N \left( u_j'^{H} v_j' + v_j'^{H} u_j' \right)^2 - 2\sum_{j,k,j\neq k} \sigma_{w,j}^N \left( \left| \langle u_j', v_k' \rangle \right|^2 + \left| \langle u_k', v_j' \rangle \right|^2 \right) \\
=&-\sum_j \sigma_{w,j}^N \left( u_j'^{H} v_j' - v_j'^{H} u_j' \right)^2 - 2\sum_j \sigma_{w,j}^N \left( u_j'^{H} v_j' v_j'^{H} u_j' + v_j'^{H} u_j' u_j'^{H} v_j' \right) \\
&-2\sum_j \sigma_{w,j}^N \cdot \left( u_j'^{H} \left( \sum_{k\neq j} v_k' v_k'^{H} \right) u_j' + v_j'^{H} \left( \sum_{k\neq j} u_k' u_k'^{H} \right) v_j' \right) \\
=&-\sum_j \sigma_{w,j}^N \left( u_j'^{H} v_j' - v_j'^{H} u_j' \right)^2 - 2\sum_j \sigma_{w,j}^N \cdot \left( u_j'^{H} V' V'^{H} u_j' + v_j'^{H} U' U'^{H} v_j' \right) \\
=&\sum_j \sigma_{w,j}^N \left| u_j'^{H} v_j' - v_j'^{H} u_j' \right|^2 - 2\sum_j \sigma_{w,j}^N \cdot \left( u_j'^{H} \Sigma u_j' + v_j'^{H} \Sigma v_j' \right) ,
\end{aligned}
\tag{133}
$$

and

$$\sum_j \sigma_{w,j}^N \left( {u_j'}^H u_j' {u_j'}^H v_j' + {v_j'}^H u_j' {u_j'}^H u_j' + {u_j'}^H v_j' {v_j'}^H v_j' + {v_j'}^H v_j' {v_j'}^H u_j' \right)$$

$$+ 2 \sum_{j,k, j \neq k} \sigma_{w,j}^N \cdot \Re \left( \langle u_k', v_j' \rangle \langle u_j', u_k' \rangle + \langle u_j', v_k' \rangle \langle v_k', v_j' \rangle \right)$$

$$= 2 \sum_{j,k} \sigma_{w,j}^N \cdot \Re \left( {u_j'}^H \left( UU'^H + VV'^H \right) v_j' \right) \tag{134}$$

$$= 2 \sum_j \sigma_{w,j}^N \cdot \left( {u_j'}^H \Sigma v_j' + {v_j'}^H \Sigma u_j' \right).$$

By combining the results above,

$$\frac{\mathrm{d}}{\mathrm{d}t} \left\| (U' - V') \Sigma_w \right\|_F^2$$

$$= -2 \sum_j \sigma_{w,j}^N \cdot \left( {u_j'}^H \Sigma u_j' + {v_j'}^H \Sigma v_j' \right) + 2 \sum_j \sigma_{w,j}^N \cdot \left( {u_j'}^H \Sigma v_j' + {v_j'}^H \Sigma u_j' \right)$$

$$-2 \sum_j \sigma_{w,j}^{2N} \cdot \left\| u_j' - v_j' \right\|^2$$

$$- \sum_{j,k, j \neq k} \frac{\sigma_{w,j}^2 \sigma_{w,k}^2 \left( \sigma_{w,j}^{N-2} - \sigma_{w,k}^{N-2} \right)}{\sigma_{w,j}^2 - \sigma_{w,k}^2} \left| {u_j'}^H v_k' - {v_j'}^H u_k' \right|^2 - \sum_j \frac{N-2}{2} \sigma_{w,j}^N \left| {u_j'}^H v_j' - {v_j'}^H u_j' \right|^2$$

$$= -2 \sum_j \sigma_{w,j}^N \cdot \left\| \Sigma^{1/2} \left( u_j' - v_j' \right) \right\|^2 - 2 \sum_j \sigma_{w,j}^{2N} \cdot \left\| u_j' - v_j' \right\|^2$$

$$- \sum_{j,k} f_N \left( \sigma_{w,j}, \sigma_{w,k} \right) \left| {u_j'}^H v_k' - {v_j'}^H u_k' \right|^2. \tag{135}$$

This completes the proof.

$\square$

For even depth $2 \mid N$, we have a similar result written in matrix form:

**Theorem 32.** *If* $2 \mid N$*, the singular values of the product matrix* $W(0)$ *are different from zero at initialization, then*

$$\frac{\mathrm{d}}{\mathrm{d}t} \left\| \Sigma^{1/2}(U - V)\Sigma_w \right\|_F^2 = -2 \left\| \Sigma(U - V)\Sigma_w^{N/2} \right\|_F^2 - 2 \left\| \Sigma^{1/2}(U - V)\Sigma_w^N \right\|_F^2$$

$$- 2\Re \left( \mathrm{tr} \left( \sum_{j=1}^{N/2-1} \Sigma U \Sigma_w^{2j} \left( U^H \Sigma V - V^H \Sigma U \right) \Sigma_w^{N-2j} V^H \right) \right) \tag{136}$$

$$\leq 0.$$

We present another approach of proof which *takes the inverse* of some terms. This approach *adapts to the skew-Hermitian term in imbalanced initialization*, where the proof of Theorem 31 in does not hold.

To prove the theorem, we introduce the following lemma.

**Lemma 33.** *If* $2 \mid N$*,* $\Sigma_w$ *is full rank at initialization, then* $\forall k = 0, 1, \cdots, N/2$ *we have*

$$\frac{\mathrm{d}}{\mathrm{d}t}(U \pm V)\Sigma_w^{2k}(U \pm V)^H$$

$$= \sum_{j=1}^{k} \left[ U\Sigma_w^{2(j-1)}U^H \Sigma V \Sigma_w^{N+2k-2j}U^H + U\Sigma_w^{N+2(j-1)}V^H \Sigma U \Sigma_w^{2(k-j)}U^H \right.$$

$$\left. + V\Sigma_w^{2(j-1)}V^H \Sigma U \Sigma_w^{N+2k-2j}V^H + V\Sigma_w^{N+2(j-1)}U^H \Sigma V \Sigma_w^{2(k-j)}V^H \right]$$

$$\pm \sum_{j=1}^{N/2+k} \left[ U\Sigma_w^{2(j-1)}U^H \Sigma V \Sigma_w^{N+2k-2j}V^H + V\Sigma_w^{2(j-1)}V^H \Sigma U \Sigma_w^{N+2k-2j}U^H \right]$$

$$\mp \sum_{j=1}^{N/2-k} \left[ U\Sigma_w^{2(j-1+k)}V^H \Sigma U \Sigma_w^{N-2j}V^H + V\Sigma_w^{2(j-1+k)}U^H \Sigma V \Sigma_w^{N-2j}U^H \right]$$

$$-2k(U \pm V)\Sigma_w^{2(N+k-1)}(U \pm V)^H. \tag{137}$$

*Proof.* $\forall l \in \mathbb{N}$ we have

$$\frac{\mathrm{d}}{\mathrm{d}t}\left(U\Sigma_w^{2l}U^H\right) = \sum_{j=1}^{l} U\Sigma_w^{2(j-1)}U^H \left(\frac{\mathrm{d}}{\mathrm{d}t}\left(U\Sigma_w^{2}U^H\right)\right) U\Sigma_w^{2(l-j)}U^H$$

$$= \sum_{j=1}^{l} U\Sigma_w^{2(j-1)}U^H \left(\Sigma V \Sigma_w^{N}U^H + U\Sigma_w^{N}V^H \Sigma^H - 2U\Sigma_w^{2N}U^H\right) U\Sigma_w^{2(l-j)}U^H. \tag{138}$$

$$\frac{\mathrm{d}}{\mathrm{d}t}\left(V\Sigma_w^{2l}V^H\right) = \sum_{j=1}^{l} V\Sigma_w^{2(j-1)}V^H \left(\frac{\mathrm{d}}{\mathrm{d}t}\left(V\Sigma_w^{2}V^H\right)\right) V\Sigma_w^{2(l-j)}V^H$$

$$= \sum_{j=1}^{l} V\Sigma_w^{2(j-1)}V^H \left(\Sigma U \Sigma_w^{N}V^H + V\Sigma_w^{N}U^H \Sigma^H - 2V\Sigma_w^{2N}V^H\right) V\Sigma_w^{2(l-j)}V^H. \tag{139}$$

From Lemma 24, $U\Sigma_w^{N-2k}U^H$ is invertible at arbitrary time $t \in [0, +\infty)$, thus

$$\frac{\mathrm{d}}{\mathrm{d}t}\left(U\Sigma_w^{-(N-2k)}U^H\right) = -\left(U\Sigma_w^{N-2k}U^H\right)^{-1}\left[\frac{\mathrm{d}}{\mathrm{d}t}\left(U\Sigma_w^{N-2k}U^H\right)\right]\left(U\Sigma_w^{N-2k}U^H\right)^{-1}$$

$$= -\left(U\Sigma_w^{-(N-2k)}U^H\right)\left[\frac{\mathrm{d}}{\mathrm{d}t}\left(U\Sigma_w^{N-2k}U^H\right)\right]\left(U\Sigma_w^{-(N-2k)}U^H\right), \tag{140}$$

which further gives

$$\frac{\mathrm{d}}{\mathrm{d}t}\left(U\Sigma_w^{2k}V^H\right)$$

$$= \left[\frac{\mathrm{d}}{\mathrm{d}t}\left(U\Sigma_w^{-(N-2k)}U^H\right)\right]U\Sigma_w^{N}V^H + U\Sigma_w^{-(N-2k)}U^H\left[\frac{\mathrm{d}}{\mathrm{d}t}\left(U\Sigma_w^{N}V^H\right)\right]$$

$$= -\left(U\Sigma_w^{-(N-2k)}U^H\right)\left[\frac{\mathrm{d}}{\mathrm{d}t}\left(U\Sigma_w^{N-2k}U^H\right)\right]\left(U\Sigma_w^{2k}V^H\right)$$

$$+ U\Sigma_w^{-(N-2k)}U^H\left[\frac{\mathrm{d}}{\mathrm{d}t}\left(U\Sigma_w^{N}V^H\right)\right]$$

$$= \sum_{j=1}^{N/2+k} U\Sigma_w^{2(j-1)}U^H \Sigma V \Sigma_w^{N+2(k-j)}V^H + \sum_{j=1}^{N/2-k} U\Sigma_w^{2(k+j-1)}V^H \Sigma^H U \Sigma_w^{N-2j}V^H$$

$$-2kU\Sigma_w^{2(N+k-1)}V^H. \tag{141}$$

Combine (138), (139) and (141) together, then the proof is completed.

□

Now we present the proof of Theorem 32.

*Proof.* Denote $Q = U^H \Sigma V$, calculate the L.H.S. of (136) by setting $k = 1$ in Lemma 33:

$$
\begin{aligned}
&\frac{\mathrm{d}}{\mathrm{d}t} \left\| \Sigma^{1/2}(U-V)\Sigma_w \right\|_F^2 \\
=& \frac{\mathrm{d}}{\mathrm{d}t} \operatorname{tr}\left( \Sigma(U-V)\Sigma_w^2(U-V)^H \right) \\
=& - 2\operatorname{tr}\left( \Sigma^2(U-V)\Sigma_w^N(U-V)^H \right) - 2\operatorname{tr}\left( \Sigma(U-V)\Sigma_w^{2N}(U-V)^H \right) \\
& -2\Re\left( \operatorname{tr}\left( \sum_{j=1}^{N/2-1} \Sigma U \Sigma_w^{2j}\left( U^H\Sigma V - V^H\Sigma U \right)\Sigma_w^{N-2j}V^H \right) \right) \\
=& - 2\left\| \Sigma(U-V)\Sigma_w^{N/2} \right\|_F^2 - 2\left\| \Sigma^{1/2}(U-V)\Sigma_w^N \right\|_F^2 \\
& -2\Re\left( \operatorname{tr}\left( \sum_{j=1}^{N/2-1} \Sigma_w^{2j}(Q-Q^H)\Sigma_w^{N-2j}Q^H \right) \right).
\end{aligned}
\tag{142}
$$

To analyze the last term,

$$
\begin{aligned}
&\Re\left( \operatorname{tr}\left( \sum_{j=1}^{N/2-1} \Sigma_w^{2j}(Q-Q^H)\Sigma_w^{N-2j}Q^H \right) \right) \\
=& \Re\left( \sum_{m,n}\left( \sum_{j=1}^{N/2-1} \sigma_m^{2j}(\Sigma_w)(Q_{mn}-\overline{Q_{nm}})\sigma_n^{N-2j}(\Sigma_w)\overline{Q_{mn}} \right) \right) \\
=& \frac{1}{2}\sum_{m,n}\left( \sum_{j=1}^{N/2-1} \sigma_m^{2j}(\Sigma_w)\sigma_n^{N-2j}(\Sigma_w)(|Q_{mn}|^2 + |Q_{nm}|^2 - 2\Re(Q_{mn}Q_{nm})) \right) \\
=& \frac{1}{2}\sum_{m,n}\left| Q_{mn}-\overline{Q_{nm}} \right|^2 \left( \sum_{j=1}^{N/2-1} \sigma_m^{2j}(\Sigma_w)\sigma_n^{N-2j}(\Sigma_w) \right) \geq 0.
\end{aligned}
\tag{143}
$$

Thus for arbitrary $\Sigma \succ O$ we have

$$
\begin{aligned}
\frac{\mathrm{d}}{\mathrm{d}t}\left\| \Sigma^{1/2}(U-V)\Sigma_w \right\|_F^2 =& -2\left\| \Sigma(U-V)\Sigma_w^{N/2} \right\|_F^2 - 2\left\| \Sigma^{1/2}(U-V)\Sigma_w^N \right\|_F^2 \\
& -\sum_{m,n}\left| Q_{mn}-\overline{Q_{nm}} \right|^2 \left( \sum_{j=1}^{N/2-1} \sigma_m^{2j}(\Sigma_w)\sigma_n^{N-2j}(\Sigma_w) \right) \\
& \leq 0.
\end{aligned}
\tag{144}
$$

which completes the proof.

□

## E.2 HERMITIAN MAIN TERM

This section proves Theorem 5.

*Proof.* Consider

$$\frac{\mathrm{d}}{\mathrm{d}t}(U+V)\Sigma_w^2(U+V)^H$$

$$=\Sigma(U+V)\Sigma_w^N(U+V)^H + (U+V)\Sigma_w^N(U+V)^H\Sigma - 2(U+V)\Sigma_w^{2N}(U+V)^H$$

$$+ \sum_{j=1}^{N/2-1}\left[U\Sigma_w^{2j}\left(U^H\Sigma V - V^H\Sigma U\right)\Sigma_w^{N-2j}V^H + V\Sigma_w^{2j}\left(V^H\Sigma U - U^H\Sigma V\right)\Sigma_w^{N-2j}U^H\right].$$

$$(145)$$

Denote $P = \frac{(U+V)\Sigma_w}{2}, Q = \frac{(U-V)\Sigma_w}{2}$. Then $P^HQ = -Q^HP$, $\Sigma_w^2 = P^HP + Q^HQ$.

From $ABC^H - CBA^H = \frac{1}{2}\left[(A-C)B(A+C)^H - (A+C)B(A-C)^H\right]$ for arbitrary $A, B, C$ we have

$$\frac{\mathrm{d}}{\mathrm{d}t}PP^H = \Sigma P\Sigma_w^{N-2}P^H + P\Sigma_w^{N-2}P^H\Sigma - 2P\Sigma_w^{2N-2}P^H$$

$$+ \sum_{j=1}^{N/2-1}\left[Q\Sigma_w^{2j-2}\left(Q^H\Sigma P - P^H\Sigma Q\right)\Sigma_w^{N-2j-2}P^H\right.$$

$$\left. - P\Sigma_w^{2j-2}\left(Q^H\Sigma P - P^H\Sigma Q\right)\Sigma_w^{N-2j-2}Q^H\right].$$

$$(146)$$

Suppose the $k^{th}$ eigenvalue and eigenvector of $PP^H$ are $x_k^2$ and $\xi_k$ respectively, $P^H\xi_k = x_k\eta_k$, then

$$\frac{\mathrm{d}}{\mathrm{d}t}x_k^2 = \xi_k^H\left(\frac{\mathrm{d}}{\mathrm{d}t}PP^H\right)\xi_k$$

$$= 2\xi_k^H\Sigma P\Sigma_w^{N-2}P^H\xi_k - 2\xi_k^H P\Sigma_w^{2N-2}P^H\xi_k$$

$$+ 2\xi_k^H\left[\sum_{j=1}^{N/2-1}Q\Sigma_w^{2j-2}\left(Q^H\Sigma P - P^H\Sigma Q\right)\Sigma_w^{N-2j-2}P^\top\right]\xi_k.$$

$$(147)$$

We focus on $N = 4, \Sigma = \sigma_1(\Sigma)I$. Then

$$\frac{\mathrm{d}}{\mathrm{d}t}x_k^2 = 2\sigma_1(\Sigma)\xi_k^H P\Sigma_w^2 P^H\xi_k - 2\xi_k^H P\Sigma_w^6 P^H\xi_k + 4\sigma_1(\Sigma)\xi_k^H QQ^H PP^H\xi_k$$

$$= 2\sigma_1(\Sigma)\xi_k^H P\Sigma_w^2 P^H\xi_k - 2\xi_k^H P\Sigma_w^6 P^H\xi_k + 4\sigma_1(\Sigma)x_k^2\xi_k^H QQ^H\xi_k.$$

$$(148)$$

For the second term:

$$\xi_k^H P\Sigma_w^6 P^H\xi_k = \xi_k^H P\left(P^HP + Q^HQ\right)\Sigma_w^2\left(P^HP + Q^HQ\right)P^H\xi_k$$

$$= x_k^4\xi_k^H P\Sigma_w^2 P^H\xi_k + 2x_k^2\xi_k^H P\Sigma_w^2 Q^H QP^H\xi_k + \xi_k^H PQ^HQ\Sigma_w^2 Q^H QP^H\xi_k$$

$$\leq x_k^4\xi_k^H P\Sigma_w^2 P^H\xi_k + 2x_k^4\|Q\|_{op}^2\|\Sigma_w\|_{op}^2 + x_k^2\|Q\|_{op}^4\|\Sigma_w\|_{op}^2.$$

$$(149)$$

From Theorem 32, $\|Q\|_{op} \leq \|Q\|_F \leq \|Q(t=0)\|_F$. Then

$$\frac{\mathrm{d}}{\mathrm{d}t}x_k^2 \geq \left(2\sigma_1(\Sigma) - x_k^4\right)\xi_k^H P\Sigma_w^2 P^H\xi_k - 2x_k^4\|Q\|_{op}^2\|\Sigma_w\|_{op}^2 - x_k^2\|Q\|_{op}^4\|\Sigma_w\|_{op}^2$$

$$\geq \left(2\sigma_1(\Sigma) - x_k^4 - \frac{1}{2}\|\Sigma_w\|_{op}^2\|((U-V)\Sigma_w)|_{t=0}\|_F^2\right)x_k^4 - \frac{1}{16}x_k^2\|\Sigma_w\|_{op}^2\|((U-V)\Sigma_w)|_{t=0}\|_F^4.$$

$$(150)$$

The lower bound is proved.

For the upper bound,

$$\frac{\mathrm{d}}{\mathrm{d}t}x_k^2 \leq 2\sigma_1(\Sigma)x_k^2\|\Sigma_w\|_{op}^2 + 4\sigma_1(\Sigma)x_k^2\|Q\|_{op}^2. \tag{151}$$

This completes the proof.

$\square$

**Corollary 34.** *If for some $k$, $\sigma_k((U+V)\Sigma_w)|_{t=0} = 0$, then $\sigma_k((U+V)\Sigma_w) \equiv 0$ for finite time $t \in [0, +\infty)$.*

*Proof.* Denote $x_k \equiv \frac{1}{2}\sigma_k((U+V)\Sigma_w)$. By Lemma 23, $\|\Sigma - W\|_F \leq \|\Sigma - W(0)\|_F$. Then $\|\Sigma_w\|_{op}$ is bounded:

$$\begin{aligned}\|\Sigma_w\|_{op} &= \|W\|_{op}^{1/N} \leq (\|\Sigma\|_{op} + \|\Sigma - W\|_{op})^{1/N} \leq (\|\Sigma\|_{op} + \|\Sigma - W\|_F)^{1/N} \\ &\leq (\|\Sigma\|_{op} + \|\Sigma - W(0)\|_F)^{1/N}.\end{aligned} \tag{152}$$

Then from Theorem 5, there exists some $C \in (0, +\infty)$ such that

$$\frac{\mathrm{d}}{\mathrm{d}t}x_k^2 \leq \sigma_1(\Sigma)\left(2\|\Sigma_w\|_{op}^2 + \|((U-V)\Sigma_w)|_{t=0}\|_F^2\right)x_k^2 \leq Cx_k^2. \tag{153}$$

Giving

$$x_k^2(t) \leq x_k^2(0)e^{Ct} = 0. \tag{154}$$

This completes the proof.

$\square$

### E.3 Convergence proof

This section states the global convergence guarantee under balanced Gaussian initialization, with gradient flow. Below we omit the confidence level $\delta$ in $f_1(\delta) = O\left(\frac{1}{\delta}\right)$ and $f_2'(\delta) = O\left(\frac{1}{\delta^2}\right)$ for simplicity.

**Theorem 35.** *Global convergence bound under balanced Gaussian initialization, gradient flow.*

*For four-layer matrix factorization under gradient flow, balanced Gaussian initialization with scaling factor $\epsilon \leq \frac{\sigma_1^{1/4}(\Sigma)}{4f_1^2 f_2' d^{29/8}}$, then for target matrix with identical singular values,*

*1. For $\mathbb{F} = \mathbb{R}$, with probability at least $\frac{1}{2}$ the loss does not converge to zero. Specifically,*

$$\mathcal{L}(t) \geq \frac{1}{2}\sigma_1^2(\Sigma), \forall t \in [0, +\infty). \tag{155}$$

*2. For $\mathbb{F} = \mathbb{C}$ with high probability and for $\mathbb{F} = \mathbb{R}$ with probability close to $\frac{1}{2}$, there exists $T(\epsilon_{\mathrm{conv}}) = \frac{16 f_2'^2 d^3}{\sigma_1(\Sigma)\epsilon^2} + \frac{1}{8\sigma_1^{3/2}(\Sigma)}\ln\left(\frac{d\sigma_1^2(\Sigma)}{\epsilon_{\mathrm{conv}}}\right)$, such that for any $\epsilon_{\mathrm{conv}} > 0$, when $t > T(\epsilon_{\mathrm{conv}})$, $\mathcal{L}(t) < \epsilon_{\mathrm{conv}}$.*

**Remark 16.** *The first part of this Theorem can be generalized to general (bounded) balanced initialization.*

*Proof.* For the first conclusion, by Theorem 3 and Corollary 34, for $\mathbb{F} = \mathbb{R}$, $\sigma_{\min}((U+V)\Sigma_w) \equiv 0$ with probability at least $\frac{1}{2}$. Consequently $\sigma_{\min}((U+V)\Sigma_w^N) \equiv 0$.

Suppose at time $t$, for some unit vector $y$, $(U + V)\Sigma_w^N y(t) = 0$. Then

$$
\begin{aligned}
\|\Sigma - W\|_F = \|\sigma_1(\Sigma)I - U\Sigma_w^N V^\top\|_F &= \|\sigma_1(\Sigma)V - U\Sigma_w^N\|_F \\
&\geq \|\sigma_1(\Sigma)V - U\Sigma_w^N\|_{op} \geq \left\|(\sigma_1(\Sigma)V - U\Sigma_w^N)y\right\| \\
&= \left\|(\sigma_1(\Sigma)V + V\Sigma_w^N)y\right\| = \left\|(\sigma_1(\Sigma) + \Sigma_w^N)y\right\| \geq \sigma_1(\Sigma).
\end{aligned}
\tag{156}
$$

For the second part:

From Lemma 23, $\|\Sigma - W\|_F \leq \|\Sigma - W(0)\|_F < 2\sqrt{d}\sigma_1(\Sigma)$. Thus for any time $t$,

$$
\begin{aligned}
\|\Sigma_w\|_{op} = \|W\|_{op}^{1/4} &\leq \left(\|\Sigma\|_{op} + \|\Sigma - W\|_{op}\right)^{1/4} \leq \left(\|\Sigma\|_{op} + \|\Sigma - W\|_F\right)^{1/N} \\
&\leq \left(\|\Sigma\|_{op} + \|\Sigma - W(0)\|_F\right)^{1/4} \leq \sqrt{2}d^{1/8}\sigma_1^{1/4}(\Sigma).
\end{aligned}
\tag{157}
$$

From Theorem 3, for $\mathbb{F} = \mathbb{C}$ with high probability (while for $\mathbb{F} = \mathbb{R}$ with probability close to $\frac{1}{2}$), $x_k(t = 0) \geq \frac{\epsilon}{2f_2'd^{3/2}}$, $\|(U - V)\Sigma_w\|_F|_{t=0} \leq 2f_1 d\epsilon$. Thus by taking $\epsilon \leq \frac{\sigma_1^{1/4}(\Sigma)}{4f_1^2 f_2' d^{29/8}}$, for $t$ such that $x_k(t) \geq x_k(0)$,

$$
\frac{\mathrm{d}}{\mathrm{d}t}x_k^2 \geq \left(2\sigma_1(\Sigma) - \left(4f_1^2 d^{9/4} + 8f_1^4 f_2'^2 d^{29/4}\right)\epsilon^2\sigma_1^{1/2}(\Sigma) - x_k^4\right)x_k^4 \geq \left(\frac{5}{4}\sigma_1(\Sigma) - x_k^4\right)x_k^4.
\tag{158}
$$

This indicates that all $x_k$ monotonically increase to $\sigma_1^{1/4}(\Sigma)$ in $T_1 = \frac{4}{\sigma_1(\Sigma)} \cdot x_k(0)^{-2} = \frac{16f_2'^2 d^3}{\sigma_1(\Sigma)\epsilon^2}$, and never decrease to below $\sigma_1^{1/4}(\Sigma)$ for $t > T_1$.

By Theorem 18, $\sigma_{\min}(\Sigma_w) \geq x_k$. Then combine with Lemma 23,

$$
\mathcal{L}_{\text{ori}}(t) \leq \mathcal{L}_{\text{ori}}(0)e^{-8\sigma_{\min}^6(\Sigma_w(T_1))(t-T_1)} \leq d\sigma_1^2(\Sigma)e^{-8\sigma_1^{3/2}(\Sigma)(t-T_1)}.
\tag{159}
$$

Thus it takes at most $t = T_1 + \frac{1}{8\sigma_1^{3/2}(\Sigma)}\ln\left(\frac{d\sigma_1^2(\Sigma)}{\epsilon_{\text{conv}}}\right)$ to reach $\epsilon_{\text{conv}}$-convergence.

$\square$

# F  NOTATIONS AND PRELIMINARIES UNDER THE DEPTH OF FOUR, IMBALANCED

To tackle the imbalanced initialization with depth $N = 4$, we make the following notations and derive some basic properties.

Below we denote $R = W_2^{-1}W_3^H$, $W_1' = RW_4^H$, $W = W_4 W_3 W_2 W_1$, $M_2 = W_2^H W_2$, $M_1 = W_1 W_1^H$, $M_{\Delta 1234} = W_2 W_1 W_1^H W_2^H - W_3^H W_4^H W_4 W_3$ $M_1' = W_1' W_1'^H$, $e_\Delta = \sqrt{\sum_{i=1}^3 \|\Delta_{i,i+1}\|_F^2}$. Then:

$$
W = W_1'^H M_2 W_1,
\tag{160}
$$

$$
RR^H = W_2^{-1}W_3^H W_3 W_2^{H-1} = I - W_2^{-1}\Delta_{23}W_2^{H-1},
\tag{161}
$$

$$
R^{-1}R^{H-1} = W_3^{H-1}W_2 W_2^H W_3^{-1} = I + W_3^{H-1}\Delta_{23}W_3^{-1},
\tag{162}
$$

$$M_{\Delta 1234} = \left( \left( W_2^H W_2 \right)^2 - \left( W_3 W_3^H \right)^2 \right) + W_3^H \Delta_{34} W_3 + W_2 \Delta_{12} W_2^H$$
$$= \frac{1}{2} \left( \Delta_{23} \left( W_3^H W_3 + W_2 W_2^H \right) + \left( W_3^H W_3 + W_2 W_2^H \right) \Delta_{23} \right) \tag{163}$$
$$+ W_3^H \Delta_{34} W_3 + W_2 \Delta_{12} W_2^H,$$

$$M_1' - M_1 = W_2^{-1} M_{\Delta 1234} W_2^{H-1}. \tag{164}$$

Deducing that

$$\|R\|_{op} \leq \sqrt{1 + \frac{1}{\sigma_{\min}^2(W_2)} \cdot \|\Delta_{23}\|_{op}} \leq \sqrt{1 + \frac{1}{\min_{j,k} \sigma_k^2(W_j)} \cdot e_\Delta}, \tag{165}$$

$$\left\|R^{-1}\right\|_{op} \leq \sqrt{1 + \frac{1}{\sigma_{\min}^2(W_3)} \cdot \|\Delta_{23}\|_{op}} \leq \sqrt{1 + \frac{1}{\min_{j,k} \sigma_k^2(W_j)} \cdot e_\Delta}, \tag{166}$$

$$\left\|I - RR^H\right\|_{op} \leq \frac{1}{\sigma_{\min}^2(W_2)} \cdot \|\Delta_{23}\|_{op} \leq \frac{1}{\min_{j,k} \sigma_k^2(W_j)} \cdot e_\Delta, \tag{167}$$

$$\left\|I - R^{-1}R^{H-1}\right\|_{op} \leq \frac{1}{\sigma_{\min}^2(W_3)} \cdot \|\Delta_{23}\|_{op} \leq \frac{1}{\min_{j,k} \sigma_k^2(W_j)} \cdot e_\Delta, \tag{168}$$

$$\|M_{\Delta 1234}\|_{op} \leq \left( \|W_2\|_{op}^2 + \|W_3\|_{op}^2 \right) \|\Delta_{23}\|_{op} + \|W_3\|_{op}^2 \|\Delta_{34}\|_{op} + \|W_2\|_{op}^2 \|\Delta_{12}\|_{op}$$
$$\leq \sqrt{6} \max_{j,k} \sigma_k^2(W_j) e_\Delta, \tag{169}$$

$$\|M_1' - M_1\|_{op} \leq \sqrt{6} \cdot \frac{\max_{j,k} \sigma_k^2(W_j)}{\sigma_{\min}^2(W_2)} e_\Delta \leq \sqrt{6} \cdot \frac{\max_{j,k} \sigma_k^2(W_j)}{\min_{j,k} \sigma_k^2(W_j)} e_\Delta. \tag{170}$$

Applying Lemma 15,

$$\left\|I - R^H R\right\|_{op} \leq \frac{1}{\sigma_{\min}^2(W_2)} \cdot \|\Delta_{23}\|_{op} \leq \frac{1}{\min_{j,k} \sigma_k^2(W_j)} \cdot e_\Delta, \tag{171}$$

$$\left\|I - R^{H-1} R^{-1}\right\|_{op} \leq \frac{1}{\sigma_{\min}^2(W_3)} \cdot \|\Delta_{23}\|_{op} \leq \frac{1}{\min_{j,k} \sigma_k^2(W_j)} \cdot e_\Delta. \tag{172}$$

# G  SKEW-HERMITIAN ERROR TERM AND HERMITIAN MAIN TERM FOR FOUR-LAYER MATRIX DECOMPOSITION

In this section, we construct skew-Hermitian error term and Hermitian main term to prepare for the convergence proof, under four-layer setting with scaled identical target matrix $\Sigma = \sigma_1(\Sigma)I$.

## G.1  SKEW-HERMITIAN ERROR TERM

The skew-Hermitian error term is defined by $\|W_1 - W_1'\|_F^2$. To address the dynamics:

### G.1.1 GRADIENT FLOW

Consider $\Sigma = \sigma_1(\Sigma)I$. We study $\|W_1 - W_1'\|_F^2$. From the derivative of inverse,

$$\frac{\mathrm{d}W_2^{-1}}{\mathrm{d}t} = -W_2^{-1}\frac{\mathrm{d}W_2}{\mathrm{d}t}W_2^{-1} = -W_1'(\Sigma - W)W_1^H W_2^{-1} - a\Delta_{12}W_2^{-1} + aW_2^{-1}\Delta_{23}, \quad (173)$$

$$\begin{aligned}
\frac{\mathrm{d}R}{\mathrm{d}t} &= \frac{\mathrm{d}W_2^{-1}}{\mathrm{d}t}W_3^H + W_2^{-1}\frac{\mathrm{d}W_3^H}{\mathrm{d}t} \\
&= -RW_4^H(\Sigma - W)W_1^H R + W_1\left(\Sigma - W^H\right)W_4 \\
&\quad - a\Delta_{12}R + 2aW_2^{-1}\Delta_{23}W_3^H - aR\Delta_{34},
\end{aligned} \quad (174)$$

$$\begin{aligned}
\frac{\mathrm{d}W_1'}{\mathrm{d}t} &= \frac{\mathrm{d}W_2^{-1}}{\mathrm{d}t}W_3^H W_4^H + W_2^{-1}\frac{\mathrm{d}W_3^H}{\mathrm{d}t}W_4^H + W_2^{-1}W_3^H\frac{\mathrm{d}W_4^H}{\mathrm{d}t} \\
&= -W_1'(\Sigma - W)W_1^H W_1' + W_1\left(\Sigma - W^H\right)W_1'^H R^{H-1}R^{-1}W_1' \\
&\quad + RR^H W_2^H W_2 W_1\left(\Sigma - W^H\right) - a\Delta_{12}W_1' + 2aW_2^{-1}\Delta_{23}W_2 W_1'.
\end{aligned} \quad (175)$$

From $\Re(\mathrm{tr}(PQ)) = 0$ if $P = P^H$ and $Q = -Q^H$, we have

$$\begin{aligned}
&\Re\left(\mathrm{tr}\left(\left(W_1'W_1^H - W_1 W_1'^H\right)W_1'\left(W_1 - W_1'\right)^H\right)\right) \\
&= -\frac{1}{2}\mathrm{tr}\left(\left(W_1'W_1^H - W_1 W_1'^H\right)\left(W_1'W_1^H - W_1 W_1'^H\right)^H\right).
\end{aligned} \quad (176)$$

Thus

$$\begin{aligned}
\frac{\mathrm{d}}{\mathrm{d}t}\|W_1 - W_1'\|_F^2 &= 2\Re\left(\mathrm{tr}\left(\frac{\mathrm{d}(W_1 - W_1')}{\mathrm{d}t}(W_1 - W_1')^H\right)\right) \\
&= 2\Re\Big(\mathrm{tr}\Big(\left[M_2 W_1'(\Sigma - W) + W_1'(\Sigma - W)W_1^H W_1'\right. \\
&\quad - W_1(\Sigma - W^H)W_1'^H R^{H-1}R^{-1}W_1' - RR^H M_2 W_1(\Sigma - W^H) \\
&\quad \left. -a\Delta_{12}\left(W_1 - W_1'\right) - 2aW_2^{-1}\Delta_{23}W_2 W_1'\right](W_1 - W_1')^H\Big)\Big) \\
&= -2\sigma_1(\Sigma)\mathrm{tr}\left((W_1 - W_1')^H M_2\left(W_1 - W_1'\right)\right) \\
&\quad - \sigma_1(\Sigma)\mathrm{tr}\left(\left(W_1'W_1^H - W_1 W_1'^H\right)\left(W_1'W_1^H - W_1 W_1'^H\right)^H\right) \\
&\quad - \mathrm{tr}\left(M_2\left(M_1' + M_1\right)M_2\left(W_1 - W_1'\right)\left(W_1 - W_1'\right)^H\right) \\
&\quad - \mathrm{tr}\left(M_2\left(M_1' - M_1\right)M_2\left(W_1' + W_1\right)\left(W_1 - W_1'\right)^H\right) \\
&\quad + 2\mathrm{tr}\left(\left[-M_1'M_2 M_1 + M_1 M_2 M_1'\right]W_1'\left(W_1 - W_1'\right)^H\right) \\
&\quad + 2\Re\left(\mathrm{tr}\left(\left[W_1(\Sigma - W^H)W_4\left(R^H R - I\right)W_4^H\right](W_1 - W_1')^H\right)\right) \\
&\quad + 2\Re\left(\mathrm{tr}\left(\left[(I - RR^H)W_2^H W_2 W_1(\Sigma - W^H)\right](W_1 - W_1')^H\right)\right) \\
&\quad - 2a\Re\left(\mathrm{tr}\left(\Delta_{12}\left(W_1 - W_1'\right)\left(W_1 - W_1'\right)^H\right)\right) \\
&\quad - 4a\Re\left(\mathrm{tr}\left(W_2^{-1}\Delta_{23}W_2 W_1'\left(W_1 - W_1'\right)^H\right)\right).
\end{aligned} \quad (177)$$

Note: $-M_1'M_2 M_1 + M_1 M_2 M_1' = \frac{1}{2}\left[\left(M_1 - M_1'\right)M_2\left(M_1 + M_1'\right) - \left(M_1 + M_1'\right)M_2\left(M_1 - M_1'\right)\right]$.

### G.1.2  GRADIENT DESCENT

From Lemma 17,

$$
\begin{aligned}
&\left\| W_2(t+1)^{-1} - W_2(t)^{-1} \right. \\
&- \eta \left[ -W_1'(t)(\Sigma - W(t))W_1(t)^H W_2(t)^{-1} - a\Delta_{12}(t)W_2(t)^{-1} + aW_2(t)^{-1}\Delta_{23}(t) \right] \Big\|_F \\
&\leq \eta^2 \left[ \left( 1 + e_\Delta(t) \left\| W_2(t)^{-1} \right\|_{op}^2 \right) \|W_1(t)\|_{op} \|\Sigma - W(t)\|_F + \sqrt{2}ae_\Delta(t) \left\| W_2(t)^{-1} \right\|_{op} \right] \\
&\quad \cdot \|W_2(t+1)^{-1}\|_{op} \|\nabla_{W_2}\mathcal{L}(t)\|_F.
\end{aligned}
\tag{178}
$$

Under $\|W_j(t+1)\|_{op} = O(\|W_j(t)\|_{op})$, $e_\Delta(t) \left\| W_2(t)^{-1} \right\|_{op}^2 = O(1)$,

$$
\begin{aligned}
&\left\| W_1'(t+1) - W_1'(t) \right. \\
&\quad -\eta \big[ -W_1'(t)(\Sigma - W(t))W_1(t)^H W_1'(t) \\
&\qquad +W_1(t)\left(\Sigma - W(t)^H\right)W_1'(t)^H R(t)^{H-1}R(t)^{-1}W_1'(t) \\
&\qquad +R(t)R(t)^H W_2(t)^H W_2(t)W_1(t)\left(\Sigma - W(t)^H\right) \\
&\qquad - a\Delta_{12}(t)W_1'(t) + 2aW_2^{-1}(t)\Delta_{23}(t)W_2(t)W_1'(t) \big] \Big\|_F \\
&= \eta^2 O\left( \left[ \max_{j\in\{1,2,3,4\}} \|W_j(t)\|_{op} \|\Sigma - W(t)\|_F + ae_\Delta(t) \left\| W_2(t)^{-1} \right\|_{op} \right] \right. \\
&\qquad \left. \cdot \max_{j\in\{1,2,3,4\}} \|W_j(t)\|_{op}^2 \cdot \|W_2(t+1)^{-1}\|_{op} \cdot \max_{j\in\{1,2,3,4\}} \|\nabla_{W_j}\mathcal{L}(t)\|_F \right).
\end{aligned}
\tag{179}
$$

Finally giving

$$
\begin{aligned}
&\left\| W_1(t+1) - W_1'(t+1) \right\|_F^2 - \left\| W_1(t) - W_1'(t) \right\|_F^2 \\
&= \Re\left( \text{tr}\left( [(W_1(t+1) - W_1'(t+1)) + (W_1(t) - W_1'(t))] \right. \right. \\
&\qquad \left. \left. \cdot \left[ (W_1(t+1) - W_1'(t+1)) - (W_1(t) - W_1'(t)) \right]^H \right) \right) \\
&= -2\eta\sigma_1(\Sigma)\text{tr}\left( (W_1(t) - W_1'(t))^H M_2(t) (W_1(t) - W_1'(t)) \right) \\
&\quad -\eta\sigma_1(\Sigma)\text{tr}\left( \left( W_1'(t)W_1(t)^H - W_1(t)W_1'(t)^H \right) \left( W_1'(t)W_1(t)^H - W_1(t)W_1'(t)^H \right)^H \right) \\
&\quad -\eta\text{tr}\left( M_2(t) (M_1'(t) + M_1(t)) M_2(t) (W_1(t) - W_1'(t))(W_1(t) - W_1'(t))^H \right) \\
&\quad -\eta\text{tr}\left( M_2(t) (M_1'(t) - M_1(t)) M_2(t) (W_1'(t) + W_1(t))(W_1(t) - W_1'(t))^H \right) \\
&\quad +2\eta\text{tr}\left( \left[ -M_1'(t)M_2(t)M_1(t) + M_1(t)M_2(t)M_1'(t) \right] W_1'(t) (W_1(t) - W_1'(t))^H \right) \\
&\quad +2\eta\Re\left( \text{tr}\left( \left[ W_1(t)(\Sigma - W(t)^H)W_4(t)\left( R(t)^H R(t) - I \right)W_4(t)^H \right] (W_1(t) - W_1'(t))^H \right) \right) \\
&\quad +2\eta\Re\left( \text{tr}\left( \left[ \left( I - R(t)R(t)^H \right)W_2(t)^H W_2(t)W_1(t)(\Sigma - W(t)^H) \right] (W_1(t) - W_1'(t))^H \right) \right) \\
&\quad -2\eta a\Re\left( \text{tr}\left( \Delta_{12}(t) (W_1(t) - W_1'(t))(W_1(t) - W_1'(t))^H \right) \right) \\
&\quad -4\eta a\Re\left( \text{tr}\left( W_2^{-1}(t)\Delta_{23}(t)W_2(t)W_1'(t) (W_1(t) - W_1'(t))^H \right) \right) \\
&\quad +\eta^2 O\left( \left[ \max_{j\in\{1,2,3,4\}} \|W_j(t)\|_{op} \|\Sigma - W(t)\|_F + ae_\Delta(t) \left\| W_2(t)^{-1} \right\|_{op} \right]^2 \right. \\
&\qquad \left. \cdot \max_{j\in\{1,2,3,4\}} \|W_j(t)\|_{op}^5 \cdot \|W_2(t+1)^{-1}\|_{op} \right).
\end{aligned}
\tag{180}
$$

### G.2 SKEW-HERMITIAN ERROR TERM

#### G.2.1 GRADIENT FLOW

For gradient flow, we study the $k^{th}$ singular value of $W_1 + W_1'$, or equivalently $\lambda_k\left((W_1 + W_1')^H (W_1 + W_1')\right) = \sigma_k^2(W_1 + W_1')$. To address the dynamics:

Suppose the left and right singular vector of $W_1 + W_1'$ corresponding to $\sigma_k(t) = \sigma_k(W_1 + W_1')(t)$ are $\eta_k(t)$ and $\chi_k(t)$ respectively, $(W_1 + W_1')\chi_k = \sigma_k \eta_k$, $\eta_k^H(W_1 + W_1') = \sigma_k \chi_k$, $\|\chi_k\| = \|\eta_k\| = 1$. Then from Lemma 22,

$$
\begin{aligned}
\frac{\mathrm{d}}{\mathrm{d}t}\lambda_k\left((W_1 + W_1')^H (W_1 + W_1')\right) &= \chi_k^H\left(\frac{\mathrm{d}}{\mathrm{d}t}(W_1 + W_1')^H(W_1 + W_1')\right)\chi_k \\
&= 2\Re\left(\chi_k^H(W_1 + W_1')^H\left(\frac{\mathrm{d}}{\mathrm{d}t}(W_1 + W_1')\right)\chi_k\right),
\end{aligned}
\tag{181}
$$

where

$$
\begin{aligned}
\frac{\mathrm{d}}{\mathrm{d}t}(W_1 + W_1') &= M_2 W_1'(\Sigma - W) - W_1'(\Sigma - W)W_1^H W_1' \\
&\quad + W_1(\Sigma - W^H)W_1'^H R^{H-1}R^{-1}W_1' + RR^H M_2 W_1(\Sigma - W^H) \\
&\quad - a\Delta_{12}(W_1 + W_1') + 2aW_2^{-1}\Delta_{23}W_2 W_1' \\
&= M_2(W_1 + W_1')\Sigma + \left(W_1\Sigma W_1'^H - W_1'\Sigma W_1^H\right)W_1' \\
&\quad - M_2\left(\frac{M_1 + M_1'}{2}M_2(W_1 + W_1') + \frac{M_1 - M_1'}{2}M_2(W_1 - W_1')\right) \\
&\quad + \left(M_1' M_2 M_1 - M_1 M_2 M_1'\right)W_1' \\
&\quad - W_1(\Sigma - W^H)W_1'^H\left(I - R^{H-1}R^{-1}\right)W_1' \\
&\quad - \left(I - RR^H\right)M_2 W_1(\Sigma - W^H) \\
&\quad - a\Delta_{12}(W_1 + W_1') + 2aW_2^{-1}\Delta_{23}W_2 W_1'.
\end{aligned}
\tag{182}
$$

Consider arbitrary $\chi \in \mathbb{F}^d$. Notice that $\left(W_1\Sigma W_1'^H - W_1'\Sigma W_1^H\right)$ is a skew-Hermitian matrix:

$$
\begin{aligned}
&\Re\left(2\chi^H(W_1 + W_1')^H\left(W_1\Sigma W_1'^H - W_1'\Sigma W_1^H\right)W_1'\chi\right) \\
=&\Re\left(\chi^H(W_1 + W_1')^H\left(W_1\Sigma W_1'^H - W_1'\Sigma W_1^H\right)W_1'\chi\right) \\
&-\Re\left(\chi^H W_1'^H\left(W_1\Sigma W_1'^H - W_1'\Sigma W_1^H\right)W_1\chi\right) \\
&-\Re\left(\chi^H W_1^H\left(W_1\Sigma W_1'^H - W_1'\Sigma W_1^H\right)W_1\chi\right) \\
=&\Re\left(\chi^H(W_1 + W_1')^H\left(-W_1\Sigma W_1'^H + W_1'\Sigma W_1^H\right)(W_1 - W_1')\chi\right).
\end{aligned}
\tag{183}
$$

From $\Sigma = \sigma_1(\Sigma)I$,

$$
-W_1\Sigma W_1'^H + W_1'\Sigma W_1^H = \sigma_1(\Sigma)(W_1 + W_1')(W_1 - W_1')^H + \sigma_1(\Sigma)(M_1' - M_1).
\tag{184}
$$

Likewise,

$$
\begin{aligned}
&\Re\left(2\chi^H(W_1 + W_1')^H\left(M_1' M_2 M_1 - M_1 M_2 M_1'\right)W_1'\chi\right) \\
=&\Re\left(\chi^H(W_1 + W_1')^H\left(M_1' M_2 M_1 - M_1 M_2 M_1'\right)(W_1' - W_1)\chi\right).
\end{aligned}
\tag{185}
$$

Thus

$$
\frac{\mathrm{d}}{\mathrm{d}t}\sigma_k^2 = 2\sigma_1(\Sigma)\sigma_k^2\eta_k^H M_2\eta_k + \sigma_1(\Sigma)\sigma_k^2\chi_k^H\left(W_1 - W_1'\right)^H\left(W_1 - W_1'\right)\chi_k
$$
$$
+ \sigma_1(\Sigma)\sigma_k\Re\left(\eta_k^H\left(M_1' - M_1\right)\left(W_1 - W_1'\right)\chi_k\right)
$$
$$
- \sigma_k^2\eta_k^H M_2(M_1 + M_1')M_2\eta_k - \sigma_k\Re\left(\eta_k^H M_2(M_1 - M_1')M_2(W_1 - W_1')\chi_k\right)
$$
$$
+ \sigma_k\Re\left(\eta_k^H\left(M_1'M_2M_1 - M_1M_2M_1'\right)\left(W_1' - W_1\right)\chi_k\right)
$$
$$
- 2\sigma_k\Re\left(\eta_k^H W_1(\Sigma - W^H)W_4\left(R^H R - I\right)W_4^H\chi_k\right)
$$
$$
- 2\sigma_k\Re\left(\eta_k^H\left(I - RR^H\right)M_2W_1(\Sigma - W^H)\chi_k\right)
$$
$$
- 2a\sigma_k^2\Re\left(\eta_k^H\Delta_{12}\eta_k\right) + 4a\sigma_k\Re\left(\eta_k^H W_2^{-1}\Delta_{23}W_2W_1'\chi_k\right). \tag{186}
$$

### G.2.2 GRADIENT DESCENT

For gradient descent, we study $\lambda_{\min}\left(\left(W_1 + W_1'\right)^H\left(W_1 + W_1'\right)\right) = \sigma_{\min}^2\left(W_1 + W_1'\right)$. To address the dynamics:

$$
\left(W_1(t+1) + W_1'(t+1)\right)
$$
$$
= W_1(t) + W_1'(t)
$$
$$
+ \eta\left[\sigma_1(\Sigma)M_2(t) - M_2(t)\frac{M_1(t) + M_1'(t)}{2}M_2(t)\right]\left(W_1(t) + W_1'(t)\right) \tag{187}
$$
$$
+ \eta\left(M_1'(t)M_2(t)M_1(t) - M_1(t)M_2(t)M_1'(t)\right)W_1'(t)
$$
$$
+ \eta\sigma_1(\Sigma)\left(W_1(t)W_1'(t)^H - W_1'(t)W_1(t)^H\right)W_1'(t) + \eta E_1(t),
$$

where the error term is bounded by

$$
\|E_1(t)\|_{op} \le \frac{1}{2}\max_{j\in\{1,2,3,4\}}\|W_j(t)\|_{op}^4\|W_1(t) - W_1'(t)\|_{op}\|M_1(t) - M_1'(t)\|_{op}
$$
$$
+ \left(\left\|R(t)^H R(t) - I\right\|_{op} + \left\|I - R(t)R(t)^H\right\|_{op}\right)\max_{j\in\{1,2,3,4\}}\|W_j(t)\|_{op}^3\|\Sigma - W(t)\|_{op}
$$
$$
+ a e_\Delta(t)\left(\|W_1(t) + W_1'(t)\|_{op} + 2\|R(t)\|_{op}\left\|W_2(t)^{-1}\right\|_{op}\max_{j\in\{1,2,3,4\}}\|W_j(t)\|_{op}^2\right)
$$
$$
+ \eta O\left(\left[\max_{j\in\{1,2,3,4\}}\|W_j(t)\|_{op}\|\Sigma - W(t)\|_F + a e_\Delta(t)\left\|W_2(t)^{-1}\right\|_{op}\right]\right.
$$
$$
\left.\cdot \max_{j\in\{1,2,3,4\}}\|W_j(t)\|_{op}^2\cdot\|W_2(t+1)^{-1}\|_{op}\cdot\max_{j\in\{1,2,3,4\}}\|\nabla_{W_j}\mathcal{L}(t)\|_F\right). \tag{188}
$$

Follow the tricks in Lemma 19,

$$
\lambda_{\min}\left(\left(W_1(t+1) + W_1'(t+1)\right)^H\left(W_1(t+1) + W_1'(t+1)\right)\right)
$$
$$
\ge \lambda_{\min}\left(\left(W_1(t) + W_1'(t)\right)^H\left(I + \eta\left[\sigma_1(\Sigma)M_2(t) - M_2(t)\frac{M_1(t) + M_1'(t)}{2}M_2(t)\right]\right)^2\left(W_1(t) + W_1'(t)\right)\right)
$$
$$
+ \eta\|E_2(t)\|_{op} + \eta^2 O\left(\|\left(W_1(t+1) + W_1'(t+1)\right) - \left(W_1(t) + W_1'(t)\right)\|_{op}^2\right), \tag{189}
$$

where

$$\|E_2(t)\|_{op} = \sigma_{\min}\left(W_1(t+1) + W_1'(t+1)\right)$$

$$\cdot \left[\|E_1(t)\|_{op} + \|W_2(t)\|_{op}^2 \|M_1(t) + M_1'(t)\|_{op} \|M_1(t) - M_1'(t)\|_{op} \|W_1(t) - W_1'(t)\|_{op}\right].$$

$$(190)$$

# H  CONVERGENCE UNDER GRADIENT FLOW, STAGED ANALYSIS

In order to present the proof more clearly, we state the complete proof of convergence under Random Gaussian Initialization C.2 and gradient flow, before tackling gradient descent.

At the beginning we assume (49) holds. (For the complex case, it holds with high probability $1 - \delta$; for the real case, it holds with probability $\frac{1}{2}(1-\delta)$. ) We omit the confidence level $\delta$ in $f_1(\delta) = O(\frac{1}{\delta})$ and $f_2(\delta) = O(\frac{1}{\delta^5})$ for simplicity.

## H.1  STAGE 1: ALIGNMENT STAGE

In this section, we set $\epsilon \leq \frac{\sigma_1^{1/4}(\Sigma)}{2f_1\sqrt{d}}$, $a \geq 2^5 f_1^{20} f_2 d^{13} \sigma_1(\Sigma) b$, where $b \geq 2^4 \ln(4f_1 d) + \ln f_2$.

Without loss of generality, $f_1 \geq 2$, and for simplicity we can further relax $f_2$ appearing in the lower bounds to $f_2 \geq f_1^6$ (now $f_2 = O\left(\frac{1}{\delta^6}\right)$).

**Theorem 36.** *At $T_1 = \frac{1}{32f_1^{14}f_2 d^{10}\epsilon^2\sigma_1(\Sigma)}$, the following conclusions hold:*

$$\sigma_{\min}\left(W_1 + W_1'\right)|_{t=T_1} \geq \frac{\epsilon}{2f_1^3 f_2 d^{9/2}}$$

$$e_\Delta(T_1) \leq 2\sqrt{3}f_1^2 d^{3/2}\epsilon^2 \exp\left(-\frac{a}{32f_1^{20}f_2 d^{13}\sigma_1(\Sigma)}\right)$$

$$\max_{j,k}|\sigma_k(W_j(T_1))| \leq (1 + 2^{-21})f_1\sqrt{d}\epsilon$$

$$\min_{j,k}|\sigma_k(W_j(T_1))| \geq (1 - 2^{-17})\frac{\epsilon}{f_1\sqrt{d}}.$$

$$(191)$$

This section proves the theorem above by following Lemmas and Corollaries.

**Lemma 37.** *Maximum and minimum singular value bound of weight matrices in alignment stage.*

*For $t \in \left[0, \frac{1}{16f_1^4 d^2\epsilon^2\sigma_1(\Sigma)}\right]$,*

$$\min_{j,k}\sigma_k(W_j) \geq \frac{\epsilon}{f_1\sqrt{d}} - 16f_1^3 d^{3/2}\epsilon^3\sigma_1(\Sigma)t, \ \max_{j,k}\sigma_k(W_j) \leq \frac{f_1\sqrt{d}\epsilon}{\sqrt{1 - 4f_1^2 d\epsilon^2\sigma_1(\Sigma)t}}. \quad (192)$$

*Proof.* For $t \geq 0$ such that $\max_{j,k}\sigma_k(W_j) \leq 2f_1\sqrt{d}\epsilon \leq \sigma_1^{1/4}(\Sigma)$,

$$\max_j\left\|\nabla_{W_j}\mathcal{L}_{\text{ori}}\right\|_{op} \leq \max_{j,k}|\sigma_k(W_j)|^3\left(\sigma_1(\Sigma) + \max_{j,k}|\sigma_k(W_j)|^4\right) \leq 2\sigma_1(\Sigma)\max_{j,k}|\sigma_k(W_j)|^3.$$

$$(193)$$

By invoking Theorem 28,

$$\frac{d\max_{j,k}\sigma_k^2(W_j)}{dt} \leq 4\max_{j,k}|\sigma_k(W_j)|^4\sigma_1(\Sigma)$$

$$\frac{d\min_{j,k}\sigma_k^2(W_j)}{dt} \geq -4\min_{j,k}|\sigma_k(W_j)|\max_{j,k}|\sigma_k(W_j)|^3\sigma_1(\Sigma).$$

$$(194)$$

By solving the differential inequality,

$$\max_{j,k} \sigma_k |W_j| \leq \frac{\max_{j,k} \sigma_k |W_j(0)|}{\sqrt{1 - 4\sigma_1(\Sigma) \max_{j,k} \sigma_k |W_j(0)|^2 \cdot t}} \leq \frac{f_1 \sqrt{d}\epsilon}{\sqrt{1 - 4f_1^2 d\epsilon^2 \sigma_1(\Sigma) t}}, \ t \in \left[0, \frac{3}{16 f_1^2 d\epsilon^2 \sigma_1(\Sigma)}\right]. \tag{195}$$

$$\min_{j,k} |\sigma_k(W_j)| \geq \frac{\epsilon}{f_1 \sqrt{d}} - 16 f_1^3 d^{3/2} \epsilon^3 \sigma_1(\Sigma) t, \ t \in \left[0, \frac{1}{16 f_1^4 d^2 \epsilon^2 \sigma_1(\Sigma)}\right]. \tag{196}$$

This completes the proof.

$\square$

Notice that

$$\max_{j,k} |\sigma_k(W_j(t \leq T_1))| \leq \frac{f_1 \sqrt{d}\epsilon}{\sqrt{1 - \frac{1}{8 f_1^{12} f_2}}} \leq (1 + 2^{-21}) f_1 \sqrt{d}\epsilon \tag{197}$$

$$\min_{j,k} |\sigma_k(W_j(t \leq T_1))| \geq \left(1 - \frac{1}{2 f_1^{10} f_2}\right) \cdot \frac{\epsilon}{f_1 \sqrt{d}} \geq (1 - 2^{-17}) \frac{\epsilon}{f_1 \sqrt{d}}.$$

**Corollary 38.** *Balanced term error in alignment stage.*

*For $t \in [0, T_1]$,*

$$e_\Delta(t) \leq 2\sqrt{3} f_1^2 d^{3/2} \epsilon^2 \exp\left(-\frac{a\epsilon^2}{f_1^6 d^3} t\right). \tag{198}$$

*Specially, at $t = T_1$,*

$$e_\Delta(T_1) \leq 2\sqrt{3} f_1^2 d^{3/2} \epsilon^2 \exp\left(-\frac{a}{32 f_1^{20} f_2 d^{13} \sigma_1(\Sigma)}\right) \leq \sqrt{3} \cdot 2^{-31} f_1^{-14} f_2^{-1} d^{-29/2} \epsilon^2. \tag{199}$$

*Proof.* By simply combining Theorem 27 and Lemma 37.

$\square$

**Corollary 39.** *Main term at the end of alignment stage.*

*At $t = T_1$,*

$$\sigma_{\min}(W_1 + W_1')|_{t=T_1} \geq \frac{\epsilon}{2 f_1^3 f_2 d^{9/2}}. \tag{200}$$

*Proof.* For simplicity, denote $\Delta_X(t) = X(t) - X(0)$ for arbitrary $X$. Note: $\Delta_{X^H} = \Delta_X^H$.

At $t = T_1$,

$$\|\Delta_W(T_1)\|_{op} = \left\| \int_0^{T_1} \sum_{j=1}^{4} \left[ W_{\Pi_L, j+1}(t') W_{\Pi_L, j+1}(t')^H (\Sigma - W(t')) W_{\Pi_R, j-1}^H(t') W_{\Pi_R, j-1}(t') \right] dt' \right\|_{op}$$

$$\leq \int_0^{T_1} \sum_{j=1}^{4} \left\| W_{\Pi_L, j+1}(t') W_{\Pi_L, j+1}(t')^H (\Sigma - W(t')) W_{\Pi_R, j-1}^H(t') W_{\Pi_R, j-1}(t') \right\|_{op} dt'$$

$$\leq \int_0^{T_1} \sum_{j=1}^{4} \left( \|\Sigma\|_{op} + \|W(t')\|_{op} \right) \left( \prod_{k \in \{1,2,3,4\},\, k \neq j} \|W_i(t')\|_{op}^2 \right) dt'$$

$$\leq \int_0^{T_1} 4 \cdot 2\sigma_1(\Sigma) \cdot \left( \left(1 + 2^{-21}\right) f_1 \sqrt{d}\epsilon \right)^6 dt'$$

$$\leq 8 \left(1 + 2^{-18}\right) f_1^6 d^3 \epsilon^6 \sigma_1(\Sigma) T_1 = \left(1 + 2^{-18}\right) \cdot \frac{1}{4} f_1^{-8} f_2^{-1} d^{-7} \epsilon^4.$$

(201)

Thus

$$\|\Delta_{W^H W}(T_1)\|_{op} = \left\| \frac{1}{2} \left[ (W(T_1) + W(0))^H \Delta_W(T_1) + \Delta_W(T_1)^H (W(T_1) + W(0)) \right] \right\|_{op}$$

$$\leq \left( \|W(T_1)\|_{op} + \|W(0)\|_{op} \right) \|\Delta_W(T_1)\|_{op}$$

$$\leq \left[ 1 + \left(1 + 2^{-21}\right)^4 \right] f_1^4 d^2 \epsilon^4 \cdot \|\Delta_W(T_1)\|_{op} = \left(1 + 2^{-17}\right) \cdot \frac{1}{2} f_1^{-4} f_2^{-1} d^{-5} \epsilon^8.$$

(202)

From Corollary 38,

$$\left\| \left( W_1(T_1)^H W_2(T_1)^H W_2(T_1) W_1(T_1) \right)^2 - W(T_1)^H W(T_1) \right\|_{op}$$

$$= \left\| W_1(T_1)^H W_2(T_1)^H M_{\Delta 1234}(T_1) W_2(T_1) W_1(T_1) \right\|_{op}$$

$$\leq \left\| W_1(T_1)^H W_2(T_1)^H \right\|_{op} \left\| M_{\Delta 1234}(T_1) \right\|_{op} \left\| W_2(T_1) W_1(T_1) \right\|_{op}$$

(203)

$$\leq \left( \left(1 + 2^{-21}\right) f_1 \sqrt{d}\epsilon \right)^4 \cdot \sqrt{6} \left( \left(1 + 2^{-21}\right) f_1 \sqrt{d}\epsilon \right)^2 \cdot e_\Delta(T_1)$$

$$\leq \sqrt{6}(1 + 2^{-18}) f_1^6 d^3 \epsilon^6 e_\Delta(T_1) \leq 2^{-28} f_1^{-8} f_2^{-16} d^{-23/2} \epsilon^8.$$

Thus

$$\left\| \left( W_1(T_1)^H W_2(T_1)^H W_2(T_1) W_1(T_1) \right)^2 - W(T_0)^H W(T_0) \right\|_{op}$$

$$\leq \left\| \left( W_1(T_1)^H W_2(T_1)^H W_2(T_1) W_1(T_1) \right)^2 - W(T_1)^H W(T_1) \right\|_{op} + \|\Delta_{W^H W}(T_1)\|_{op}$$

(204)

$$\leq (1 + 2^{-16}) \cdot \frac{1}{2} f_1^{-4} f_2^{-1} d^{-5} \epsilon^8.$$

From Lemma 16,

$$\left\| W_1(T_1)^H W_2(T_1)^H W_2(T_1) W_1(T_1) - \left( W(T_0)^H W(T_0) \right)^{1/2} \right\|_{op}$$

$$\leq \frac{\left\| \left( W_1(T_1)^H W_2(T_1)^H W_2(T_1) W_1(T_1) \right)^2 - W(T_0)^H W(T_0) \right\|_{op}}{2\sqrt{\lambda_{\min} \left( W(T_0)^H W(T_0) \right) - \left\| \left( W_1(T_1)^H W_2(T_1)^H W_2(T_1) W_1(T_1) \right)^2 - W(T_0)^H W(T_0) \right\|_{op}}}$$

$$\leq \frac{(1 + 2^{-16}) \cdot \frac{1}{2} f_1^{-4} f_2^{-1} d^{-5} \epsilon^8}{2\sqrt{\left( \frac{\epsilon}{f_1 \sqrt{d}} \right)^8 - (1 + 2^{-16}) \cdot \frac{1}{2} f_1^{-4} f_2^{-1} d^{-5} \epsilon^8}} \leq 0.27 f_2^{-1} d^{-3} \epsilon^4. \tag{205}$$

By (C.2),

$$\sigma_{\min} \left( W_1(T_1)^H W_2(T_1)^H W_2(T_1) W_1(T_1) + W(T_1)^H \right)$$

$$\geq \sigma_{\min} \left( \left( W(T_0)^H W(T_0) \right)^{1/2} + W(0)^H \right)$$

$$- \left\| W_1(T_1)^H W_2(T_1)^H W_2(T_1) W_1(T_1) - \left( W(T_0)^H W(T_0) \right)^{1/2} \right\|_{op} - \| \Delta_W(T_1) \|_{op} \tag{206}$$

$$\geq f_2^{-1} d^{-3} \epsilon^4 - 0.27 f_2^{-1} d^{-3} \epsilon^4 - \left( 1 + 2^{-18} \right) \cdot \frac{1}{4} f_1^{-8} f_2^{-1} d^{-7} \epsilon^4$$

$$\geq 0.72 f_2^{-1} d^{-3} \epsilon^4,$$

which further gives

$$\sigma_{\min} \left( W_1 + W_1' \right) \big|_{t=T_1}$$

$$= \sigma_{\min} \left( \left( W_1(T_1)^H W_2(T_1)^H W_2(T_1) \right)^{-1} \left( W_1(T_1)^H W_2(T_1)^H W_2(T_1) W_1(T_1) + W(T_1)^H \right) \right)$$

$$\geq \left( \frac{1}{\max_{j,k} |\sigma_k(W_j(T_1))|} \right)^3 \cdot \sigma_{\min} \left( W_1(T_1)^H W_2(T_1)^H W_2(T_1) W_1(T_1) + W(T_1)^H \right)$$

$$\geq \frac{\epsilon}{2 f_1^3 f_2 d^{9/2}}. \tag{207}$$

$\square$

## H.2 STAGE 2: SADDLE AVOIDANCE STAGE

In this stage, we further assume $a \geq 32 f_1^{20} f_2 d^{13} \sigma_1(\Sigma) \left( 5 \ln \left( \frac{\sigma_1^{1/4}(\Sigma)}{\epsilon} \right) + \frac{281}{8} \ln d + 23 \ln(4f_1) + 7 \ln f_2 \right)$, while $\frac{\epsilon}{\sigma_1^{1/4}(\Sigma)} \leq \frac{1}{32 f_1^5 f_2 d^{53/8}}$. From Lemma 26 and Theorem 36,

$$e_\Delta(t \in [T_1, +\infty)) \leq e_\Delta(T_1) \leq 2\sqrt{3} f_1^2 d^{3/2} \epsilon^2 \exp \left( -\frac{a}{32 f_1^{20} f_2 d^{13} \sigma_1(\Sigma)} \right)$$

$$\leq \sqrt{3} \cdot 2^{-45} f_1^{-21} f_2^{-7} d^{-269/8} \epsilon^7 \sigma_1^{-5/4}(\Sigma). \tag{208}$$

Moreover, $a \geq 32 f_1^{20} f_2 d^{13} \sigma_1(\Sigma) b$, where $b - \ln b \geq 3 \ln \left( \frac{\sigma_1^{1/4}(\Sigma)}{\epsilon} \right) + \frac{303}{8} \ln d + 37 \ln(2f_1) + 6 \ln f_2$. Thus

$$ae_\Delta(t \in [T_1, +\infty)) \leq ae_\Delta(T_1) \leq 2^6 \sqrt{3} f_1^{22} f_2 d^{29/2} \epsilon^2 \sigma_1(\Sigma) \exp(-(b - \ln b))$$

$$\leq \sqrt{3} \cdot 2^{-31} f_1^{-15} f_2^{-5} d^{-187/8} \epsilon^5 \sigma_1^{1/4}(\Sigma). \tag{209}$$

**Theorem 40.** *At $T_1 + T_2$, $T_2 = \frac{32f_1^6 f_2^2 d^9}{\sigma_1(\Sigma)\epsilon^2}$, the following conclusions hold:*

$$\|W_1(T_1 + T_2) - W_1'(T_1 + T_2)\|_F \leq 3f_1 d\epsilon$$
$$\sigma_{\min}(W_1 + W_1')(T_1 + T_2) \geq 2^{3/4}\sigma_1^{1/4}(\Sigma). \tag{210}$$

**Lemma 41.** *Bound of operator norms throughout the optimization process.*

*For $t \in [0, +\infty)$,*

$$\|\Sigma - W(t)\|_{op} \leq \|\Sigma - W(t)\|_F \leq 1.01\sqrt{d}\sigma_1(\Sigma)$$
$$\|W\|_{op} \leq \|W\|_F \leq 3\sqrt{d}\sigma_1(\Sigma)$$
$$\max_j \|W_j\|_{op} \leq \max_j \|W_j\|_F \leq \sqrt{2}d^{1/8}\sigma_1^{1/4}(\Sigma). \tag{211}$$

*Proof.* For $t \in [0, T_1]$, the result is obvious from Theorem 36 and Lemma 37.

For $t \in (T_1, +\infty)$: from Lemma 23,

$$\|\Sigma - W(t)\|_{op} \leq \|\Sigma - W(t)\|_F \leq \|\Sigma - W(0)\|_F \leq \|\Sigma\|_F + \|W(0)\|_F \leq \sqrt{2d}\sigma_1(\Sigma). \tag{212}$$

Giving

$$\|W(t)\|_{op} \leq \|W(t)\|_F \leq \|\Sigma - W(t)\|_F + \|\Sigma\|_F \leq 3\sqrt{d}\sigma_1(\Sigma). \tag{213}$$

For the last inequality, prove by contradiction.

Suppose $\max_j \|W_j\|_{op} \geq \sqrt{2}d^{1/8}\sigma_1^{1/4}(\Sigma)$, then by invoking Corollary 38,

$$e_\Delta(t) \leq e_\Delta(T_1) \leq \sqrt{3} \cdot 2^{-15} f_1^{-14} f_2^{-16} d^{-29/2}\epsilon^2 \leq 2^{-15} \max_j \|W_j\|_{op}^2. \tag{214}$$

Thus for $t > T_1$,

$$\|W\|_{op}^2 = \|WW^H\|_{op} = \|W_4 W_3 W_2 W_1 W_1^H W_2^H W_3^H W_4^H\|_{op}$$
$$\geq \|W_4 W_4^H\|_{op} - \|W_4 W_3 W_2 \Delta_{12} W_2^H W_3^H W_4^H\|_{op}$$
$$- \|W_4 W_3 \Delta_{23} W_2 W_2^H W_3^H W_4^H\|_{op} - \|W_4 W_3 W_2 W_2^H \Delta_{23} W_3^H W_4^H\|_{op}$$
$$- \left\|W_4 \Delta_{34} \left(W_3 W_3^H\right)^2 W_4^H\right\|_{op} - \|W_4 W_3 W_3^H \Delta_{34} W_3 W_3^H W_4^H\|_{op} - \left\|W_4 \left(W_3 W_3^H\right)^2 \Delta_{34} W_4^H\right\|_{op}$$
$$\geq \left(\max_j \|W_j\|_{op}^2 - 3e_\Delta\right)^4 - 6e_\Delta \max_j \|W_j\|_{op}^6 > 15\sqrt{d}\sigma_1(\Sigma). \tag{215}$$

which contradicts inequality (213). This completes the proof.

$\square$

**Lemma 42.** *Bound of $\left\|W_2^{-1}\right\|_{op}$, $\left\|W_3^{-1}\right\|_{op}$, and relevant terms.*

*For $t \in [T_1, T_1 + T_2]$,*

$$\max\left(\left\|W_2^{-1}(t)\right\|_{op}, \left\|W_3^{-1}(t)\right\|_{op}\right) \leq 128 f_1^6 f_2^2 d^{77/8}\epsilon^{-2}\sigma_1^{1/4}(\Sigma), \tag{216}$$

$$\max\left(e_\Delta(t)\left\|W_2^{-1}(t)\right\|_{op}^2, e_\Delta(t)\left\|W_3^{-1}(t)\right\|_{op}^2\right) \leq \sqrt{3} \cdot 2^{-31} f_1^{-9} f_2^{-3} d^{-115/8}\epsilon^3\sigma_1^{-3/4}(\Sigma). \tag{217}$$

*Proof.* We begin with the time derivative of $W_2^{-1}$ and $W_3^{-1}$:

$$\frac{\mathrm{d}W_2^{-1}}{\mathrm{d}t} = -RW_4^H(\Sigma - W)W_1^H W_2^{-1} - a\Delta_{12}W_2^{-1} + aW_2^{-1}\Delta_{23}$$

$$\frac{\mathrm{d}W_3^{-1}}{\mathrm{d}t} = -W_3^{-1}W_4^H(\Sigma - W)W_1^H R^{H-1} - a\Delta_{23}W_3^{-1} + aW_3^{-1}\Delta_{34}. \tag{218}$$

From $\frac{\mathrm{d}}{\mathrm{d}t}\|M\|_{op} \leq \left\|\frac{\mathrm{d}}{\mathrm{d}t}M\right\|_{op}$ (this in equality is from triangular inequality and standard calculus analysis),

$$\frac{\mathrm{d}}{\mathrm{d}t}\left\|W_2^{-1}\right\|_{op} \leq \|R\|_{op}\|W_4\|_{op}\|\Sigma - W\|_{op}\left\|W_1^H W_2^{-1}\right\|_{op}$$
$$+ a\|\Delta_{12}\|_{op}\left\|W_2^{-1}\right\|_{op} + a\left\|W_2^{-1}\right\|_{op}\|\Delta_{23}\|_{op}$$

$$\frac{\mathrm{d}}{\mathrm{d}t}\left\|W_3^{-1}\right\|_{op} \leq \left\|W_3^{-1}W_4^H\right\|_{op}\|\Sigma - W\|_{op}\|W_1\|_{op}\|R\|_{op}$$
$$+ a\|\Delta_{23}\|_{op}\left\|W_3^{-1}\right\|_{op} + a\left\|W_3^{-1}\right\|_{op}\|\Delta_{34}\|_{op}. \tag{219}$$

From Lemma 41 and

$$\|R\|_{op} \leq \sqrt{1 + \frac{1}{\sigma_{\min}^2(W_2)} \cdot \|\Delta_{23}\|_{op}}$$

$$\left\|R^{-1}\right\|_{op} \leq \sqrt{1 + \frac{1}{\sigma_{\min}^2(W_3)} \cdot \|\Delta_{23}\|_{op}}$$

$$\left\|W_1^H W_2^{-1}\right\|_{op} = \sqrt{\left\|W_2^{H-1}W_1 W_1^H W_2^{-1}\right\|_{op}} = \sqrt{\left\|I + W_2^{H-1}\Delta_{12}W_2^{-1}\right\|}$$

$$\leq \sqrt{1 + e_\Delta \left\|W_2^{-1}\right\|_{op}^2}$$

$$\left\|W_3^{-1}W_4^H\right\|_{op} = \sqrt{\left\|W_3^{-1}W_4^H W_4 W_3^{H-1}\right\|_{op}} = \sqrt{\left\|I - W_3^{-1}\Delta_{34}W_3^{H-1}\right\|}$$

$$\leq \sqrt{1 + e_\Delta \left\|W_3^{-1}\right\|_{op}^2}. \tag{220}$$

Further we have

$$\frac{\mathrm{d}}{\mathrm{d}t}\left\|W_2^{-1}\right\|_{op} \leq 2\sqrt{2}\left(1 + e_\Delta \left\|W_2^{-1}\right\|_{op}^2\right)d^{5/8}\sigma_1^{5/4}(\Sigma) + \sqrt{2}ae_\Delta\left\|W_2^{-1}\right\|_{op}$$

$$\frac{\mathrm{d}}{\mathrm{d}t}\left\|W_3^{-1}\right\|_{op} \leq 2\sqrt{2}\left(1 + e_\Delta \left\|W_3^{-1}\right\|_{op}^2\right)d^{5/8}\sigma_1^{5/4}(\Sigma) + \sqrt{2}ae_\Delta\left\|W_3^{-1}\right\|_{op}. \tag{221}$$

Combine with (208) and (209), for $t \geq T_1$ such that (216) holds,

$$\max\left(\frac{\mathrm{d}}{\mathrm{d}t}\left\|W_2^{-1}\right\|_{op}, \frac{\mathrm{d}}{\mathrm{d}t}\left\|W_3^{-1}\right\|_{op}\right)$$

$$\leq 2\sqrt{2}(1 + \sqrt{3} \cdot 2^{-31})d^{5/8}\sigma_1^{5/4}(\Sigma) + 2^{-22}f_1^{-9}f_2^{-3}d^{-55/4}\epsilon^3\sigma_1^{1/2}(\Sigma) \tag{222}$$

$$\leq 2\sqrt{2}(1 + 2^{-20})d^{5/8}\sigma_1^{5/4}(\Sigma).$$

From Theorem 36, $\max\left(\left\|W_2(T_1)^{-1}\right\|_{op}, \left\|W_3(T_1)^{-1}\right\|_{op}\right) \leq \frac{1}{\min_{j,k}|\sigma_k(W_j(T_1))|} \leq \frac{f_1\sqrt{d}}{(1-2^{-17})\epsilon}$, then the proof of the first inequality is completed via integration during the time interval $[T_1, T_1 + T_2]$. The second inequality follows immediately.

$\square$

**Remark 17.** *This Lemma verifies that $W_{2,3}^{-1}$ are bounded (consequently $W_{2,3}$ are full rank), then $R$ is well defined throughout this stage. For $t > T_1 + T_2$, further analysis shows that the minimum singular values of $W_2$ and $W_3$ are lower bounded by $\Omega(\sigma_1^{1/4}(\Sigma))$.*

**Lemma 43.** *Skew-Hermitian error.*

*For $t \in [T_1, T_1 + T_2]$,*

$$\|W_1 - W_1'\|_F \leq 3f_1 d\epsilon. \tag{223}$$

*Proof.* From section G.1.1,

$$
\begin{aligned}
\frac{\mathrm{d}}{\mathrm{d}t}\|W_1 - W_1'\|_F^2 &= -2\sigma_1(\Sigma)\mathrm{tr}\left((W_1 - W_1')^H M_2 (W_1 - W_1')\right) \\
&\quad - \sigma_1(\Sigma)\mathrm{tr}\left((W_1'W_1^H - W_1W_1'^H)(W_1'W_1^H - W_1W_1'^H)^H\right) \\
&\quad - \mathrm{tr}\left(M_2(M_1' + M_1)M_2(W_1 - W_1')(W_1 - W_1')^H\right) \\
&\quad - \mathrm{tr}\left(M_2(M_1' - M_1)M_2(W_1' + W_1)(W_1 - W_1')^H\right) \\
&\quad + 2\mathrm{tr}\left([-M_1'M_2M_1 + M_1M_2M_1']W_1'(W_1 - W_1')^H\right) \\
&\quad + 2\Re\left(\mathrm{tr}\left([W_1(\Sigma - W^H)W_4(R^HR - I)W_4^H](W_1 - W_1')^H\right)\right) \\
&\quad + 2\Re\left(\mathrm{tr}\left([(I - RR^H)W_2^HW_2W_1(\Sigma - W^H)](W_1 - W_1')^H\right)\right) \\
&\quad - 2a\Re\left(\mathrm{tr}\left(\Delta_{12}(W_1 - W_1')(W_1 - W_1')^H\right)\right) \\
&\quad - 4a\Re\left(\mathrm{tr}\left(W_2^{-1}\Delta_{23}W_2W_1'(W_1 - W_1')^H\right)\right).
\end{aligned}
\tag{224}
$$

Note: $-M_1'M_2M_1 + M_1M_2M_1' = \frac{1}{2}\left[(M_1 - M_1')M_2(M_1 + M_1') - (M_1 + M_1')M_2(M_1 - M_1')\right]$. From Lemma 42, for $t \in [T_1, T_1 + T_2]$,

$$
\begin{aligned}
\max\left(\|R^HR - I\|_{op}, \|I - RR^H\|_{op}\right) &\leq e_\Delta \|W_2^{-1}\|_{op}^2 \\
&\leq \sqrt{3} \cdot 2^{-31} f_1^{-9} f_2^{-3} d^{-115/8} \epsilon^3 \sigma_1^{-3/4}(\Sigma),
\end{aligned}
\tag{225}
$$

$$
\begin{aligned}
\|M_1 - M_1'\|_{op} &\leq \sqrt{6} \cdot \frac{\max_{j,k}\sigma_k^2(W_j)}{\sigma_{\min}^2(W_2)} e_\Delta \\
&\leq 2^{-27} f_1^{-9} f_2^{-3} d^{-113/8} \epsilon^3 \sigma_1^{-1/4}(\Sigma),
\end{aligned}
\tag{226}
$$

$$
\begin{aligned}
\left\|M_2 - \frac{M_1 + M_1'}{2}\right\|_{op} &\leq \|\Delta_{12}\|_{op} + \frac{1}{2}\|M_1 - M_1'\|_{op} \leq \left[1 + \frac{\sqrt{6}}{2} \cdot \frac{\max_{j,k}\sigma_k^2(W_j)}{\sigma_{\min}^2(W_2)}\right]e_\Delta \\
&\leq 2^{-28} f_1^{-9} f_2^{-3} d^{-113/8} \epsilon^3 \sigma_1^{-1/4}(\Sigma).
\end{aligned}
\tag{227}
$$

Consequently:

$$\|R\|_{op} \leq \sqrt{1 + e_\Delta \|W_2^{-1}\|_{op}^2} \leq 1 + \sqrt{3} \cdot 2^{-32} f_1^{-9} f_2^{-3} d^{-115/8} \epsilon^3 \sigma_1^{-3/4}(\Sigma), \tag{228}$$

$$\|W_1'\|_{op} \leq \|W_1'\|_F \leq \sqrt{2}d^{1/8}\sigma_1^{1/4}(\Sigma)\|R\|_{op} \leq (1 + 2^{-31})\sqrt{2}d^{1/8}\sigma_1^{1/4}(\Sigma), \tag{229}$$

$$\left\|\frac{M_1 + M_1'}{2}\right\|_{op} \le \|M_2\|_{op} + \left\|M_2 - \frac{M_1 + M_1'}{2}\right\|_{op} \le \left(1 + 2^{-29}\right) 2d^{1/4} \sigma_1^{1/2}(\Sigma), \qquad (230)$$

$$\begin{aligned}
\|M_1' M_2 M_1 - M_1 M_2 M_1'\|_{op} &\le \|M_1 - M_1'\| \|M_2\| \|M_1 + M_1'\| \\
&\le \left(1 + 2^{-29}\right) 2^{-25} f_1^{-9} f_2^{-3} d^{-109/8} \epsilon^3 \sigma_1^{3/4}(\Sigma).
\end{aligned} \qquad (231)$$

By combining all results above, for $t \in [T_1, T_1 + T_2]$ such that $\|W_1 - W_1'\|_F \le 3 f_1 d\epsilon$ holds,

$$\begin{aligned}
\frac{\mathrm{d}}{\mathrm{d}t} \|W_1 - W_1'\|_F^2 \le\ & -0 - 0 - 0 \\
& + \|M_2\|_F \|M_1' - M_1\|_{op} \|M_2\|_{op} \left(\|W_1'\|_{op} + \|W_1\|_{op}\right) \|W_1 - W_1'\|_F \\
& + 2 \|-M_1' M_2 M_1 + M_1 M_2 M_1'\|_{op} \|W_1'\|_F \|W_1 - W_1'\|_F \\
& + 2 \max_j \|W_j\|_{op}^3 \|\Sigma - W\|_F \left(\|R^H R - I\|_{op} + \|I - R R^H\|_{op}\right) \|W_1 - W_1'\|_F \\
& + 2 a e_\Delta \|W_1 - W_1'\|_F^2 \\
& + 4 a e_\Delta \|W_2^{-1}\|_{op} \|W_2\|_F \|W_1'\|_{op} \|W_1 - W_1'\|_F \\
\le\ & 2^{-22} f_1^{-8} f_2^{-3} d^{-25/2} \epsilon^4 \sigma_1(\Sigma) \\
& + 2^{-21} f_1^{-8} f_2^{-3} d^{-25/2} \epsilon^4 \sigma_1(\Sigma) \\
& + 2^{-24} f_1^{-8} f_2^{-3} d^{-25/2} \epsilon^4 \sigma_1(\Sigma) \\
& + 2^{-26} f_1^{-13} f_2^{-5} d^{-171/8} \epsilon^7 \sigma_1^{1/4}(\Sigma) \\
& + 2^{-18} f_1^{-8} f_2^{-3} d^{-25/2} \epsilon^4 \sigma_1(\Sigma) \\
\le\ & 2^{-17} f_1^{-8} f_2^{-3} d^{-25/2} \epsilon^4 \sigma_1(\Sigma).
\end{aligned} \qquad (232)$$

From Theorem 36, at $t = T_1$,

$$\begin{aligned}
\|W_1(T_1) - W_1'(T_1)\|_F &\le \|W_1(T_1)\|_F + \|W_1'(T_1)\|_F \le \|W_1(T_1)\|_F + \|W_4(T_1)\|_F \|R(T_1)\|_{op} \\
&\le \left(1 + 2^{-32}\right) 2\sqrt{d} \cdot \left(1 + 2^{-21}\right) f_1 \sqrt{d} \epsilon \le \left(1 + 2^{-20}\right) 2 f_1 d\epsilon.
\end{aligned} \qquad (233)$$

Thus $\|W_1 - W_1'\|_F^2 \le \sqrt{\left[\left(1 + 2^{-20}\right) 2 f_1 d\epsilon\right]^2 + 2^{-17} f_1^{-8} f_2^{-3} d^{-25/2} \epsilon^4 \sigma_1(\Sigma)(t - T_1)}$ , when both $t \in [T_1, T_1 + T_2]$ and $\|W_1 - W_1'\|_F^2 \le 3 f_1 d\epsilon$ hold. Then

$$\begin{aligned}
&\|W_1(T_1 + T_2) - W_1'(T_1 + T_2)\|_F^2 \\
&\le \sqrt{\left[\left(1 + 2^{-20}\right) 2 f_1 d\epsilon\right]^2 + 2^{-17} f_1^{-8} f_2^{-3} d^{-25/2} \epsilon^4 \sigma_1(\Sigma) T_2} \\
&\le \sqrt{\left[\left(1 + 2^{-20}\right) 2 f_1 d\epsilon\right]^2 + 2^{-12} f_1^{-2} f_2^{-1} d^{-7/2} \epsilon^2} < 3 f_1 d\epsilon.
\end{aligned} \qquad (234)$$

which completes the proof.

$\square$

**Corollary 44.** *The minimum eigenvalue of Hermitian term.*

*For any $\sigma_k(W_1 + W_1')(T_1) \ge \frac{\epsilon}{2 f_1^3 f_2 d^{9/2}}$, it takes at most time $T_2$ to increase to $2^{3/4} \sigma_1^{1/4}(\Sigma)$.*

*Proof.* We analyze the dynamics of $\lambda_k \left((W_1 + W_1')^H (W_1 + W_1')\right) = \sigma_k^2$. The definition of $\eta_k(t)$ and $\chi_k(t)$ follows section G.2.1. The dynamics can be expressed as below:

$$\frac{\mathrm{d}}{\mathrm{d}t}\sigma_k^2 = 2\sigma_1(\Sigma)\sigma_k^2\eta_k^H M_2\eta_k + \sigma_1(\Sigma)\sigma_k^2\chi_k^H (W_1 - W_1')^H (W_1 - W_1')\chi_k$$
$$+ \sigma_1(\Sigma)\sigma_k\Re\left(\eta_k^H (M_1' - M_1)(W_1 - W_1')\chi_k\right)$$
$$- \sigma_k^2\eta_k^H M_2(M_1 + M_1')M_2\eta_k - \sigma_k\Re\left(\eta_k^H M_2(M_1 - M_1')M_2(W_1 - W_1')\chi_k\right)$$
$$+ \sigma_k\Re\left(\eta_k^H (M_1'M_2M_1 - M_1M_2M_1')(W_1' - W_1)\chi_k\right) \tag{235}$$
$$- 2\sigma_k\Re\left(\eta_k^H W_1(\Sigma - W^H)W_4\left(R^H R - I\right)W_4^H\chi_k\right)$$
$$- 2\sigma_k\Re\left(\eta_k^H \left(I - RR^H\right)M_2W_1(\Sigma - W^H)\chi_k\right)$$
$$- 2a\sigma_k^2\Re\left(\eta_k^H\Delta_{12}\eta_k\right) + 4a\sigma_k\Re\left(\eta_k^H W_2^{-1}\Delta_{23}W_2W_1'\chi_k\right).$$

From $\left\|M_2 - \frac{M_1 + M_1'}{2}\right\|_{op} \leq 2^{-28}f_1^{-9}f_2^{-3}d^{-113/8}\epsilon^3\sigma_1^{-1/4}(\Sigma)$ and $\left\|\frac{M_1 + M_1'}{2}\right\|_{op} \leq \left(1 + 2^{-29}\right)2d^{1/4}\sigma_1^{1/2}(\Sigma),$

$$\eta_k^H M_2\eta_k \geq \eta_k^H\left(\frac{M_1 + M_1'}{2}\right)\eta_k - \left\|M_2 - \frac{M_1 + M_1'}{2}\right\|_{op}$$
$$\geq \eta_k^H\left(\frac{M_1 + M_1'}{2}\right)\eta_k - 2^{-28}f_1^{-9}f_2^{-3}d^{-113/8}\epsilon^3\sigma_1^{-1/4}(\Sigma)$$
$$\eta_k^H M_2(M_1 + M_1')M_2\eta_k \leq \eta_k^H\left(\frac{M_1 + M_1'}{2}\right)(M_1 + M_1')\left(\frac{M_1 + M_1'}{2}\right)\eta_k$$
$$+ 2\left\|M_2 - \frac{M_1 + M_1'}{2}\right\|_{op}\left\|\frac{M_1 + M_1'}{2}\right\|_{op}\left(\|M_2\|_{op} + \left\|\frac{M_1 + M_1'}{2}\right\|_{op}\right)$$
$$\leq \eta_k^H\left(\frac{M_1 + M_1'}{2}\right)(M_1 + M_1')\left(\frac{M_1 + M_1'}{2}\right)\eta_k$$
$$+ \left(1 + 2^{-28}\right)2^{-24}f_1^{-9}f_2^{-3}d^{-109/8}\epsilon^3\sigma_1^{3/4}(\Sigma). \tag{236}$$

By Lemma 43, $\|W_1 - W_1'\|_{op} \leq \|W_1 - W_1'\|_F \leq 3f_1 d\epsilon,$

$$
\begin{aligned}
\frac{\mathrm{d}}{\mathrm{d}t}\sigma_k^2 \geq{}& 2\sigma_1(\Sigma)\sigma_k^2\eta_k^H M_2\eta_k + 0 \\
&- \sigma_1(\Sigma)\sigma_k\left\|M_1' - M_1\right\|_{op}\left\|W_1 - W_1'\right\|_{op} \\
&- \sigma_k^2\eta_k^H M_2(M_1 + M_1')M_2\eta_k - \sigma_k\max_j\|W_j\|_{op}^4\left\|M_1 - M_1'\right\|_{op}\left\|W_1 - W_1'\right\|_{op} \\
&- \sigma_k\left\|M_1'M_2M_1 - M_1M_2M_1'\right\|_{op}\left\|W_1' - W_1\right\|_{op} \\
&- 2\sigma_k\max_j\|W_j\|_{op}^3\|\Sigma - W\|_{op}\left(\left\|R^H R - I\right\|_{op} + \left\|I - RR^H\right\|_{op}\right) \\
&- 2ae_\Delta\sigma_k^2 - 4ae_\Delta\sigma_k\left\|W_2^{-1}\right\|_{op}\max_j\|W_j\|_{op}^2\|R\|_{op} \\
\geq{}& 2\sigma_1(\Sigma)\sigma_k^2\left(\eta_k^H\left(\frac{M_1 + M_1'}{2}\right)\eta_k - 2^{-28}f_1^{-9}f_2^{-3}d^{-113/8}\epsilon^3\sigma_1^{-1/4}(\Sigma)\right) \\
&- \sigma_k\left\|W_1 - W_1'\right\|_{op}\cdot 2^{-27}f_1^{-9}f_2^{-3}d^{-113/8}\epsilon^3\sigma_1^{3/4}(\Sigma) \\
&- \sigma_k^2\left[\eta_k^H\left(\frac{M_1 + M_1'}{2}\right)(M_1 + M_1')\left(\frac{M_1 + M_1'}{2}\right)\eta_k + \left(1 + 2^{-28}\right)2^{-24}f_1^{-9}f_2^{-3}d^{-109/8}\epsilon^3\sigma_1^{3/4}(\Sigma)\right] \\
&- \sigma_k\left\|W_1 - W_1'\right\|_{op}\cdot 2^{-25}f_1^{-9}f_2^{-3}d^{-109/8}\epsilon^3\sigma_1^{3/4}(\Sigma) \\
&- \sigma_k\left\|W_1 - W_1'\right\|_{op}\cdot\left(1 + 2^{-29}\right)2^{-25}f_1^{-9}f_2^{-3}d^{-109/8}\epsilon^3\sigma_1^{3/4}(\Sigma) \\
&- \sigma_k\cdot 2^{-25}f_1^{-9}f_2^{-3}d^{-27/2}\epsilon^3\sigma_1(\Sigma) \\
&- \sigma_k^2\cdot 2^{-29}f_1^{-15}f_2^{-5}d^{-187/8}\epsilon^5\sigma_1^{1/4}(\Sigma) - \sigma_k\cdot 2^{-22}f_1^{-9}f_2^{-3}d^{-27/2}\epsilon^3\sigma_1(\Sigma) \\
\geq{}& 2\sigma_k^2\eta_k^H\left[\sigma_1(\Sigma)\left(\frac{M_1 + M_1'}{2}\right) - \left(\frac{M_1 + M_1'}{2}\right)^3\right]\eta_k \\
&- \sigma_k\cdot\left(1 + 2^{-1}\right)2^{-22}f_1^{-9}f_2^{-3}d^{-27/2}\epsilon^3\sigma_1(\Sigma) - \sigma_k^2\cdot 2^{-23}f_1^{-9}f_2^{-3}d^{-109/8}\epsilon^3\sigma_1(\Sigma).
\end{aligned}
\tag{237}
$$

under $\sigma_k \geq \frac{\epsilon}{2f_1^3 f_2 d^{9/2}}$,

$$
\frac{\mathrm{d}}{\mathrm{d}t}\sigma_k^2 \geq 2\sigma_k^2\eta_k^H\left[\sigma_1(\Sigma)\left(\frac{M_1 + M_1'}{2}\right) - \left(\frac{M_1 + M_1'}{2}\right)^3\right]\eta_k - 2^{-18}\sigma_1(\Sigma)\sigma_k^4.
\tag{238}
$$

Denote $P = \frac{W_1 + W_1'}{2}$, $Q = \frac{W_1 - W_1'}{2}$. Notice that

$$
PP^H + QQ^H = \frac{M_1 + M_1'}{2}, \quad P^H\eta_k = \frac{1}{2}\sigma_k\chi_k,
\tag{239}
$$

$$
\eta_k^H\left(\frac{M_1 + M_1'}{2}\right)\eta_k = \eta_k^H\left(PP^H + QQ^H\right)\eta_k \geq \frac{1}{4}\sigma_k^2,
\tag{240}
$$

$$
\begin{aligned}
\eta_k^H\left(\frac{M_1 + M_1'}{2}\right)^3\eta_k ={}& \eta_k^H\left(PP^H + QQ^H\right)\left(\frac{M_1 + M_1'}{2}\right)\left(PP^H + QQ^H\right)\eta_k \\
={}& \frac{1}{16}\sigma_k^4\eta_k^H\left(\frac{M_1 + M_1'}{2}\right)\eta_k + \eta_k^H QQ^H\left(\frac{M_1 + M_1'}{2}\right)QQ^H\eta_k \\
&+ \frac{1}{4}\sigma_k^2\eta_k^H\left[QQ^H\left(\frac{M_1 + M_1'}{2}\right) + \left(\frac{M_1 + M_1'}{2}\right)QQ^H\right]\eta_k \\
\leq{}& \frac{1}{16}\sigma_k^4\eta_k^H\left(\frac{M_1 + M_1'}{2}\right)\eta_k + \left\|\frac{M_1 + M_1'}{2}\right\|_{op}\left(\frac{1}{2}\sigma_k^2\|Q\|_{op}^2 + \|Q\|_{op}^4\right).
\end{aligned}
\tag{241}
$$

Notice $\|Q\|_{op} = \frac{1}{2}\|W_1 - W_1'\|_F \leq \frac{3}{2}f_1 d\epsilon \leq \sigma_k \cdot 3f_1^4 f_2 d^{11/2}$, $\epsilon \leq \frac{1}{32 f_1^5 f_2 d^{53/8}}\sigma_1^{1/4}(\Sigma)$,

$$
\begin{aligned}
\frac{\mathrm{d}}{\mathrm{d}t}\sigma_k^2 &\geq 2\sigma_k^2\left[\left(\sigma_1(\Sigma) - \frac{1}{16}\sigma_k^4\right)\eta_k^H\left(\frac{M_1 + M_1'}{2}\right)\eta_k - \left\|\frac{M_1 + M_1'}{2}\right\|_{op}\left(\frac{1}{2}\sigma_k^2\|Q\|_{op}^2 + \|Q\|_{op}^4\right)\right] \\
&\quad - 2^{-18}\sigma_1(\Sigma)\sigma_k^4 \\
&\geq \frac{1}{2}\sigma_k^4\left(\sigma_1(\Sigma) - \frac{1}{16}\sigma_k^4\right) - 2\sigma_k^2\left\|\frac{M_1 + M_1'}{2}\right\|_{op}\|Q\|_{op}^2\left(\frac{1}{2}\sigma_k^2 + \|Q\|_{op}^2\right) - 2^{-18}\sigma_1(\Sigma)\sigma_k^4 \\
&\geq \frac{1}{2}\sigma_k^4\sigma_1(\Sigma) - \frac{1}{32}\sigma_k^8 - 81\left(1 + 2^{-5}\right)f_1^{10}f_2^2 d^{53/4}\epsilon^2\sigma_1^{1/2}(\Sigma)\sigma_k^4 - 2^{-18}\sigma_1(\Sigma)\sigma_k^4 \\
&\geq \frac{3}{8}\sigma_k^4\sigma_1(\Sigma) - \frac{1}{32}\sigma_k^8.
\end{aligned}
\tag{242}
$$

This indicates that for $\sigma_k \in \left[\frac{\epsilon}{2f_1^3 f_2 d^{9/2}}, 2^{3/4}\sigma_1^{1/4}(\Sigma)\right]$, $\sigma_k$ is monotonically increasing. By standard calculus, it takes at most time $\Delta t\left(\sigma_k \geq 2^{3/4}\sigma_1^{1/4}(\Sigma)\right) \leq T_2$ for $\sigma_k$ to increase from at least $\frac{\epsilon}{2f_1^3 f_2 d^{9/2}}$ to $2^{3/4}\sigma_1^{1/4}(\Sigma)$:

$$
\begin{aligned}
\Delta t\left(\sigma_k \geq 2^{3/4}\sigma_1^{1/4}(\Sigma)\right) &\leq \int_{\frac{\epsilon}{2f_1^3 f_2 d^{9/2}}}^{2\cdot\sqrt[4]{\frac{\sigma_1(\Sigma)}{2}}}\left(\frac{3}{8}\sigma_1(\Sigma)\sigma_k^4 - \frac{1}{32}\sigma_k^8\right)^{-1}\mathrm{d}\left(\sigma_k^2\right) \\
&= \int_{\frac{\epsilon}{4f_1^6 f_2^2 d^9}}^{4\cdot\sqrt{\frac{\sigma_1(\Sigma)}{2}}}\left(\frac{3}{8}\sigma_1(\Sigma)\lambda_k^2 - \frac{1}{32}\lambda_k^4\right)^{-1}\mathrm{d}\lambda_k \\
&\leq \int_{\frac{\epsilon}{4f_1^6 f_2^2 d^9}}^{4\cdot\sqrt{\frac{\sigma_1(\Sigma)}{2}}}\left(\frac{3}{8}\sigma_1(\Sigma)\lambda_k^2 - \frac{1}{4}\sigma_1(\Sigma)\lambda_k^2\right)^{-1}\mathrm{d}\lambda_k \\
&\leq 8\left[\left(\frac{\epsilon}{4f_1^6 f_2^2 d^9}\right)^{-1} - \left(4\cdot\sqrt{\frac{\sigma_1(\Sigma)}{2}}\right)^{-1}\right]\sigma_1^{-1}(\Sigma) \leq T_2.
\end{aligned}
\tag{243}
$$

And for $t \in \left[T_1 + \Delta t\left(\sigma_k \geq 2^{3/4}\sigma_1^{1/4}(\Sigma)\right), T_1 + T_2\right]$, $\sigma_k$ does not decrease to less than $2^{3/4}\sigma_1^{1/4}(\Sigma)$ if $t \leq T_1 + T_2$. This is from the continuity of $\sigma_k$ and the time derivative of $\sigma_k^2$ at $\sigma_k = 2^{3/4}\sigma_1^{1/4}(\Sigma), t \leq T_1 + T_2$ is positive:

$$
\left.\frac{\mathrm{d}}{\mathrm{d}t}\sigma_k^2\right|_{\sigma_k = 2^{3/4}\sigma_1^{1/4}(\Sigma), t\leq T_1 + T_2} \geq \frac{1}{8}\sigma_1(\Sigma)\cdot\left(2^{3/4}\sigma_1^{1/4}(\Sigma)\right)^4 > 0.
\tag{244}
$$

$\square$

### H.3 Stage 3: local convergence stage

In this stage, we analysis the time to reach $\epsilon_{\mathrm{conv}}$-convergence, that is

$$
T(\epsilon_{\mathrm{conv}}) = \inf_t\{\mathcal{L}(t) \leq \epsilon_{\mathrm{conv}}\}.
\tag{245}
$$

**Lemma 45.** $\sigma_{\min}(W_1 + W_1')$ *is lower bounded, while the skew-Hermitian error is upper bounded. For $t \geq T_1 + T_2$,*

$$\sigma_{\min}\left(W_1 + W_1'\right)(t) \geq 2^{3/4}\sigma_1^{1/4}(\Sigma)$$
$$\|W_1 - W_1'\|_F \leq 3f_1 d\epsilon. \tag{246}$$

*Proof.* (246) holds at $t = T_1 + T_2$. Since both L.H.S. change continuously, it left to prove that the derivatives at these thresholds (to be specific, $t' \geq T_2$ such that $\|W_1 - W_1'\|_F|_{t=t'} = 3f_1 d\epsilon$ or $\sigma_k\left(W_1 + W_1'\right)|_{t=t'} = 2^{3/4}\sigma_1(\Sigma)$) are positive/negative. (If such time does not exist, then the proof is done. )

From

$$\sigma_{\min}^2(W_1) + \sigma_{\min}^2(W_1') \geq \frac{1}{2}\lambda_{\min}\left((W_1 + W_1')(W_1 + W_1')^H + (W_1 - W_1')(W_1 - W_1')^H\right)$$
$$\geq \frac{1}{2}\sigma_{\min}^2\left(W_1 + W_1'\right), \tag{247}$$

and

$$\sigma_{\min}(W_1') \leq \sigma_{\min}(W_1) + \|W_1 - W_1'\|_F. \tag{248}$$

For $t > T_1 + T_2$ such as (246) holds,

$$\sigma_{\min}(W_2) \geq \sigma_{\min}(W_1) - e_\Delta \geq \frac{1}{\sqrt{2}}\sigma_1^{1/4}(\Sigma). \tag{249}$$

Then by following almost the same arguments as Lemma 43 and 44,

$$\frac{d}{dt}\|W_1 - W_1'\|_F^2 \leq -2\sigma_1(\Sigma)\text{tr}\left((W_1 - W_1')^H \sigma_{\min}^2(W_2)(W_1 - W_1')\right) - 0 - 0$$
$$+ 2^{-17}f_1^{-8}f_2^{-3}d^{-25/2}\epsilon^4\sigma_1(\Sigma) \tag{250}$$
$$\leq -\sigma_1^{3/2}(\Sigma)\|W_1 - W_1'\|_F^2 + 2^{-17}f_1^{-8}f_2^{-3}d^{-25/2}\epsilon^4\sigma_1(\Sigma),$$

$$\frac{d}{dt}\sigma_k^2\left(W_1 + W_1'\right) \geq \frac{3}{8}\sigma_k^4\left(W_1 + W_1'\right)\sigma_1(\Sigma) - \frac{1}{32}\sigma_k^8\left(W_1 + W_1'\right). \tag{251}$$

Suppose for some $t_1, t_2 \geq T_1 + T_2$ such that $\|W_1 - W_1'\|_F|_{t=t_1} = 3f_1 d\epsilon$, $\sigma_k\left(W_1 + W_1'\right)|_{t=t_2} = 2^{3/4}\sigma_1(\Sigma)$, then

$$\left.\frac{d}{dt}\|W_1 - W_1'\|_F^2\right|_{t=t_1} \leq 0$$
$$\left.\frac{d}{dt}\sigma_k^2\left(W_1 + W_1'\right)\right|_{t=t_2} \geq 0. \tag{252}$$

This completes the proof.

$\square$

**Theorem 46.** *Global convergence bound.*

*For four-layer matrix factorization under gradient flow, with random Gaussian initialization with scaling factor $\epsilon \leq \frac{\sigma_1^{1/4}(\Sigma)}{32f_1^5 f_2 d^{53/8}}$, regularization factor $a \geq 32f_1^{20}f_2 d^{13}\sigma_1(\Sigma)b$, where b satisfies*

$$b \geq 5 \ln \left( \frac{\sigma_1^{1/4}(\Sigma)}{\epsilon} \right) + \frac{281}{8} \ln d + 23 \ln(4f_1) + 7 \ln f_2$$

$$b - \ln b \geq 3 \ln \left( \frac{\sigma_1^{1/4}(\Sigma)}{\epsilon} \right) + \frac{303}{8} \ln d + 37 \ln(2f_1) + 6 \ln f_2. \tag{253}$$

*Then for target matrix with identical singular values, there exists following $T(\epsilon_{\mathrm{conv}})$, such that for any $\epsilon_{\mathrm{conv}} > 0$, (1) with high probability over the complex initialization (2) with probability close to $\frac{1}{2}$ over the real initialization, when $t > T(\epsilon_{\mathrm{conv}})$, $\mathcal{L}(t) < \epsilon_{\mathrm{conv}}$.*

$$
\begin{aligned}
T(\epsilon_{\mathrm{conv}}) &\leq T_1 + T_2 + \sigma_1^{-3/2}(\Sigma) \ln \left( \frac{d\sigma_1^2(\Sigma)}{\epsilon_{\mathrm{conv}}} \right) \\
&= \frac{1}{32 f_1^{14} f_2 d^{10} \epsilon^2 \sigma_1(\Sigma)} + \frac{32 f_1^6 f_2^2 d^9}{\sigma_1(\Sigma) \epsilon^2} + \sigma_1^{-3/2}(\Sigma) \ln \left( \frac{d\sigma_1^2(\Sigma)}{\epsilon_{\mathrm{conv}}} \right) \\
&= O \left( \frac{f_1^6 f_2^2 d^9}{\sigma_1(\Sigma) \epsilon^2} + \frac{1}{\sigma_1^{3/2}(\Sigma)} \ln \left( \frac{d\sigma_1^2(\Sigma)}{\epsilon_{\mathrm{conv}}} \right) \right).
\end{aligned}
\tag{254}
$$

*Proof.* Following the derivations in Lemma 45,

$$\min_{j,k} \sigma_k(W_j)(t > T_1 + T_2) \geq \frac{1}{\sqrt{2}} \sigma_1^{1/4}(\Sigma). \tag{255}$$

By Lemma 23 and 41,

$$
\begin{aligned}
\mathcal{L}_{\mathrm{ori}}(t) &\leq \mathcal{L}_{\mathrm{ori}}(T_1 + T_2) \exp \left( -8 \min_{j,k} |\sigma_k(W_j)(t > T_1 + T_2)|^6 (t - T_1 - T_2) \right) \\
&\leq \mathcal{L}_{\mathrm{ori}}(0) \exp \left( -8 \min_{j,k} |\sigma_k(W_j)(t > T_1 + T_2)|^6 (t - T_1 - T_2) \right) \\
&\leq 0.52 d\sigma_1^2(\Sigma) \exp \left( -\sigma_1^{3/2}(\Sigma)(t - T_1 - T_2) \right).
\end{aligned}
\tag{256}
$$

For regularization term, by invoking Theorem 27, 36 and Lemma 41,

$$
\begin{aligned}
\mathcal{L}_{\mathrm{reg}}(t) &\leq \mathcal{L}_{\mathrm{reg}}(T_1 + T_2) \exp \left( -\frac{4a}{3} \frac{\min_{j,k} |\sigma_k(W_j)(t > T_1 + T_2)|^4}{\max_{j,k} |\sigma_k(W_j)|^2} \cdot (t - T_1 - T_2) \right) \\
&\leq \frac{a}{4} e_\Delta^2(T_1 + T_2) \exp \left( -\frac{4a}{3} \frac{\min_{j,k} |\sigma_k(W_j)(t > T_1 + T_2)|^4}{\max_{j,k} |\sigma_k(W_j)|^2} \cdot (t - T_1 - T_2) \right) \\
&\leq \frac{a}{4} e_\Delta^2(T_1) \exp \left( -\frac{4a}{3} \frac{\min_{j,k} |\sigma_k(W_j)(t > T_1 + T_2)|^4}{\max_{j,k} |\sigma_k(W_j)|^2} \cdot (t - T_1 - T_2) \right) \\
&\leq 2^{-76} f_1^{-36} f_2^{-12} d^{-57} \epsilon^{12} \sigma_1^{-1}(\Sigma) \exp \left( -16 f_1^{20} f_2 d^{51/4} \sigma_1^{3/2}(\Sigma)(t - T_1 - T_2) \right).
\end{aligned}
\tag{257}
$$

By taking logarithm on the summation of these two inequalities, the proof is completed.

$\square$

# I CONVERGENCE UNDER GRADIENT DESCENT, STAGED ANALYSIS

This section states the complete proof of convergence under Random Gaussian Initialization C.2.

At the beginning we still assume (49) holds. (For the complex case, it holds with high probability $1 - \delta$; for the real case, it holds with probability $\frac{1}{2}(1 - \delta)$. )

**Theorem 47.** *Global convergence bound under random Gaussian initialization, gradient descent.*

*For four-layer matrix factorization under gradient descent, random Gaussian initialization with scaling factor $\epsilon \leq \frac{\sigma_1^{1/4}(\Sigma)}{32 f_1^5 f_2 d^{53/8}}$, regularization factor $a \geq 32 f_1^{20} f_2 d^{13} \sigma_1(\Sigma) b$, where $b$ satisfies*

$$b \geq \max\left( 5 \ln\left( \frac{\sigma_1^{1/4}(\Sigma)}{\epsilon} \right) + \frac{281}{8} \ln d + 23 \ln(4 f_1) + 7 \ln f_2, \ 16 \ln(2 f_1 f_2 d) \right)$$

$$b - \ln b \geq 3 \ln\left( \frac{\sigma_1^{1/4}(\Sigma)}{\epsilon} \right) + \frac{303}{8} \ln d + 37 \ln(2 f_1) + 6 \ln f_2. \tag{258}$$

*Then for target matrix with identical singular values, there exists following learning rate $\eta$ and convergence time $T(\epsilon_{\mathrm{conv}}, \eta)$, such that for any $\epsilon_{\mathrm{conv}} > 0$, (1) with high probability over the complex initialization (2) with probability close to $\frac{1}{2}$ over the real initialization, when $t > T(\epsilon_{\mathrm{conv}}, \eta)$, $\mathcal{L}(t) < \epsilon_{\mathrm{conv}}$.*

$$\eta = O\Big( \min\Big( a^{-2} f_1^{-4} d^{-2} \epsilon^{-2} \sigma_1(\Sigma),$$
$$a f_1^{-56} f_2^{-14} d^{-301/4} \epsilon^8 \sigma_1^{-9/2}(\Sigma), a^{-1} f_1^{-44} f_2^{-10} d^{-219/4} \epsilon^4 \sigma_1^{-3/2}(\Sigma),$$
$$f_1^{-27} f_2^{-9} d^{-355/8} \epsilon^9 \sigma_1^{-15/4}(\Sigma), a^{-1} f_1^{-21} f_2^{-7} d^{-273/8} \epsilon^7 \sigma_1^{-9/4}(\Sigma) \Big) \Big)$$

$$T(\epsilon_{\mathrm{conv}}, \eta) \leq T_1 + T_2 + \eta^{-1} \sigma_1^{-3/2}(\Sigma) \ln\left( \frac{d \sigma_1^2(\Sigma)}{\epsilon_{\mathrm{conv}}} \right)$$

$$= O\left( \frac{f_1^6 f_2^2 d^9}{\eta \sigma_1(\Sigma) \epsilon^2} + \frac{1}{\eta \sigma_1^{3/2}(\Sigma)} \ln\left( \frac{d \sigma_1^2(\Sigma)}{\epsilon_{\mathrm{conv}}} \right) \right). \tag{259}$$

The following section completes the proof.

### I.1 STAGE 1: ALIGNMENT STAGE

In this section, we set $\epsilon \leq \frac{\sigma_1^{1/4}(\Sigma)}{4 f_1 \sqrt{d}}$, $a \geq 2^5 f_1^{20} f_2 d^{13} \sigma_1(\Sigma) b$, where $b \geq 2^4 \ln(4 f_1 d) + \ln f_2$. $\eta = O\left( \frac{\sigma_1(\Sigma)}{a^2 f_1^4 d^2 \epsilon^2} \right)$, with appropriate small constant. Without loss of generality, $f_1 \geq 2$, $f_2 \geq f_1^6$.

**Theorem 48.** *At $T_1 = \frac{1}{32 f_1^{14} f_2 d^{10} \epsilon^2 \sigma_1(\Sigma) \eta}$, the following conclusions hold:*

$$\sigma_{\min}(W_1 + W_1')|_{t=T_1} \geq \frac{\epsilon}{2 f_1^3 f_2 d^{9/2}}$$

$$e_\Delta(T_1) \leq 2\sqrt{3 f_1^4 d^3 \epsilon^4 e^{-2b} + \eta O\left( a^{-1} f_1^{14} d^8 \epsilon^6 \sigma_1^2(\Sigma) \right)}$$

$$\max_{j,k} |\sigma_k(W_j(T_1))| \leq (1 + 2^{-21}) f_1 \sqrt{d} \epsilon \tag{260}$$

$$\min_{j,k} |\sigma_k(W_j(T_1))| \geq (1 - 2^{-17}) \frac{\epsilon}{f_1 \sqrt{d}}.$$

This section proves the theorem above by following Lemmas and Corollaries.

**Lemma 49.** *Maximum and minimum singular value bound of weight matrices in alignment stage.*

*For $t \in \left[0, \frac{1}{32 f_1^4 d^2 \epsilon^2 \sigma_1(\Sigma) \eta}\right]$,*

$$\min_{j,k} \sigma_k(W_j) \geq \frac{\epsilon}{f_1 \sqrt{d}} - 16 f_1^3 d^{3/2} \epsilon^3 \sigma_1(\Sigma) t, \ \max_{j,k} \sigma_k(W_j) \leq \frac{f_1 \sqrt{d} \epsilon}{\sqrt{1 - 4 f_1^2 d \epsilon^2 \sigma_1(\Sigma) t}}. \tag{261}$$

*Proof.* For $t \geq 0$ such that $\max_{j,k} \sigma_k(W_j) \leq 2f_1\sqrt{d}\epsilon \leq \frac{1}{2}\sigma_1^{1/4}(\Sigma)$,

$$\max_j \left\| \nabla_{W_j} \mathcal{L}_{\mathrm{ori}} \right\|_{op} \leq \max_{j,k} |\sigma_k(W_j)|^3 \left( \sigma_1(\Sigma) + \max_{j,k} |\sigma_k(W_j)|^4 \right) \leq \frac{3}{2} \max_{j,k} |\sigma_k(W_j)|^3 \sigma_1(\Sigma).$$

$$(262)$$

By invoking Corollary 30, for $t \geq 0$ such that $\min_{j,k} \sigma_k(W_j(t)) \geq \frac{\epsilon}{2f_1\sqrt{d}}$,

$$\max_{j,k} \sigma_k^2(W_j(t+1)) - \max_{j,k} \sigma_k^2(W_j(t)) \leq 3\eta \max_{j,k} |\sigma_k(W_j(t))|^4 \sigma_1(\Sigma)$$
$$+ \eta^2 O\left( a^2 \left( \epsilon f_1 \sqrt{d} \right)^6 \right)$$
$$\leq 4\eta \max_{j,k} |\sigma_k(W_j(t))|^4 \sigma_1(\Sigma)$$
$$\min_{j,k} \sigma_k^2(W_j(t+1)) - \min_{j,k} \sigma_k^2(W_j(t)) \geq -3\eta \min_{j,k} |\sigma_k(W_j(t))| \max_{j,k} |\sigma_k(W_j(t))|^3 \sigma_1(\Sigma)$$
$$+ \eta^2 O\left( a^2 \left( \epsilon f_1 \sqrt{d} \right)^6 \right)$$
$$\geq -2\eta \left( \min_{j,k} |\sigma_k(W_j(t+1))| + \min_{j,k} |\sigma_k(W_j(t))| \right)$$
$$\cdot \max_{j,k} |\sigma_k(W_j(t))|^3 \sigma_1(\Sigma).$$

$$(263)$$

By solving the differential inequality,

$$\max_{j,k} \sigma_k|W_j(t)| \leq \frac{\max_{j,k} \sigma_k|W_j(0)|}{\sqrt{1 - 4\sigma_1(\Sigma) \max_{j,k} \sigma_k|W_j(0)|^2 \eta t}} \leq \frac{f_1\sqrt{d}\epsilon}{\sqrt{1 - 4f_1^2 d\epsilon^2 \sigma_1(\Sigma)\eta t}}, \ t \in \left[ 0, \frac{3}{16f_1^2 d\epsilon^2 \sigma_1(\Sigma)\eta} \right],$$

$$(264)$$

$$\min_{j,k} |\sigma_k(W_j(t))| \geq \frac{\epsilon}{f_1\sqrt{d}} - 16f_1^3 d^{3/2}\epsilon^3 \sigma_1(\Sigma)\eta t, \ t \in \left[ 0, \frac{1}{32f_1^4 d^2\epsilon^2 \sigma_1(\Sigma)\eta} \right]. \qquad (265)$$

This completes the proof.

$$\square$$

Notice that

$$\max_{j,k} |\sigma_k(W_j(t \leq T_1))| \leq \frac{f_1\sqrt{d}\epsilon}{\sqrt{1 - \frac{1}{8f_1^{12} f_2}}} \leq (1 + 2^{-21})f_1\sqrt{d}\epsilon$$

$$(266)$$

$$\min_{j,k} |\sigma_k(W_j(t \leq T_1))| \geq \left( 1 - \frac{1}{2f_1^{10} f_2} \right) \cdot \frac{\epsilon}{f_1\sqrt{d}} \geq (1 - 2^{-17})\frac{\epsilon}{f_1\sqrt{d}}.$$

**Corollary 50.** *Balanced term error in alignment stage.*

$$e_\Delta(T_1) \leq \sqrt{3} \cdot 2^{-31} f_1^{-14} f_2^{-1} d^{-29/2}\epsilon^2. \qquad (267)$$

*Proof.* By simply combining Theorem 29 and Lemma 49, denote $M = \max_{j,k,t \leq T_1}(W_j(t))$,

$$\mathcal{L}_{\text{reg}}(t+1) \leq \left(1 - 2.509\frac{\eta a\epsilon^2}{f_1^6 d^3}\right) \cdot \mathcal{L}_{\text{reg}}(t) + \eta^2 O\left(a^2 M^4 \mathcal{L}_{\text{reg}}(t) + \sqrt{a\mathcal{L}_{\text{reg}}(t)} M^6 \mathcal{L}_{\text{ori}}(t)\right)$$

$$+ \eta^4 O\left(aM^{12}\mathcal{L}_{\text{ori}}(t)^2 + a^3 M^4 \mathcal{L}_{\text{reg}}(t)^2\right)$$

$$\leq \left(1 - \frac{2\eta a\epsilon^2}{f_1^6 d^3}\right) \cdot \mathcal{L}_{\text{reg}}(t) + \eta^2 O\left(aM^8 \mathcal{L}_{\text{ori}}(t)\right)$$

$$\leq \left(1 - \frac{2\eta a\epsilon^2}{f_1^6 d^3}\right) \cdot \mathcal{L}_{\text{reg}}(t) + \eta^2 O\left(af_1^8 d^5 \epsilon^8 \sigma_1^2(\Sigma)\right),$$

$$(268)$$

giving

$$\mathcal{L}_{\text{reg}}(t) \leq \mathcal{L}_{\text{reg}}(0)e^{-\frac{2\eta a\epsilon^2}{f_1^6 d^3}t} + \eta O\left(f_1^{14} d^8 \epsilon^6 \sigma_1^2(\Sigma)\right). \tag{269}$$

$$\mathcal{L}_{\text{reg}}(T_1) \leq 3af_1^4 d^3 \epsilon^4 e^{-2b} + \eta O\left(f_1^{14} d^8 \epsilon^6 \sigma_1^2(\Sigma)\right), \tag{270}$$

$$e_\Delta(T_1) = 2\sqrt{\frac{\mathcal{L}_{\text{reg}}(T_1)}{a}} \leq \sqrt{3} \cdot 2^{-31} f_1^{-14} f_2^{-1} d^{-29/2} \epsilon^2. \tag{271}$$

$$\square$$

**Corollary 51.** *Main term at the end of alignment stage.*

*At $t = T_1$,*

$$\sigma_{\min}\left(W_1 + W_1'\right)\big|_{t=T_1} \geq \frac{\epsilon}{2f_1^3 f_2 d^{9/2}}. \tag{272}$$

*Proof.* Denote $\Delta_X(t) = X(t) - X(0)$ for arbitrary $X$.

At $t = T_1$,

$$\|\Delta_W(T_1)\|_{op} \leq \left\|\sum_{t'=0}^{T_1-1} \eta \left[\sum_{j=1}^{4} W_{\Pi_L,j+1}(t') W_{\Pi_L,j+1}(t')^H \left(\Sigma - W(t')\right) W_{\Pi_R,j-1}^H(t') W_{\Pi_R,j-1}(t')\right]\right\|_{op}$$

$$+ \eta^2 \sum_{t'=0}^{T_1-1} O\left(\max_{j\in\{1,2,3,4\}} \|\nabla_{W_j}\mathcal{L}(t')\|_F^2 \cdot \max_{j\in\{1,2,3,4\}} \|W_j(t')\|_{op}^2\right)$$

$$\leq \eta T_1 \cdot 6\sigma_1(\Sigma) \cdot \left(\left(1+2^{-21}\right) f_1\sqrt{d}\epsilon\right)^6 + \eta^2 T_1 O\left(a^2 d\left(f_1\sqrt{d}\epsilon\right)^8\right)$$

$$\leq \eta T_1 \cdot 8\sigma_1(\Sigma) \cdot \left(\left(1+2^{-21}\right) f_1\sqrt{d}\epsilon\right)^6$$

$$\leq \left(1+2^{-18}\right) \cdot \frac{1}{4} f_1^{-8} f_2^{-1} d^{-7}\epsilon^4. \tag{273}$$

Thus

$$\|\Delta_{W^H W}(T_1)\|_{op} = \left\|\frac{1}{2}\left[\left(W(T_1) + W(0)\right)^H \Delta_W(T_1) + \Delta_W(T_1)^H \left(W(T_1) + W(0)\right)\right]\right\|_{op}$$

$$\leq \left(1+2^{-17}\right) \cdot \frac{1}{2} f_1^{-4} f_2^{-1} d^{-5}\epsilon^8. \tag{274}$$

From Corollary 50,

$$\left\| \left( W_1(T_1)^H W_2(T_1)^H W_2(T_1) W_1(T_1) \right)^2 - W(T_1)^H W(T_1) \right\|_{op}$$

$$\leq \left\| W_1(T_1)^H W_2(T_1)^H \right\|_{op} \left\| M_{\Delta 1234}(T_1) \right\|_{op} \left\| W_2(T_1) W_1(T_1) \right\|_{op} \tag{275}$$

$$\leq 2^{-12} f_1^{-8} f_2^{-16} d^{-23/2} \epsilon^8.$$

Thus

$$\left\| \left( W_1(T_1)^H W_2(T_1)^H W_2(T_1) W_1(T_1) \right)^2 - W(T_0)^H W(T_0) \right\|_{op}$$

$$\leq \left\| \left( W_1(T_1)^H W_2(T_1)^H W_2(T_1) W_1(T_1) \right)^2 - W(T_1)^H W(T_1) \right\|_{op} + \left\| \Delta_{W^H W}(T_1) \right\|_{op} \tag{276}$$

$$\leq (1 + 2^{-16}) \cdot \frac{1}{2} f_1^{-4} f_2^{-1} d^{-5} \epsilon^8.$$

From Lemma 16,

$$\left\| W_1(T_1)^H W_2(T_1)^H W_2(T_1) W_1(T_1) - \left( W(T_0)^H W(T_0) \right)^{1/2} \right\|_{op}$$

$$\leq \frac{\left\| \left( W_1(T_1)^H W_2(T_1)^H W_2(T_1) W_1(T_1) \right)^2 - W(T_0)^H W(T_0) \right\|_{op}}{2\sqrt{\lambda_{\min}\left( W(T_0)^H W(T_0) \right) - \left\| \left( W_1(T_1)^H W_2(T_1)^H W_2(T_1) W_1(T_1) \right)^2 - W(T_0)^H W(T_0) \right\|_{op}}}$$

$$\leq \frac{(1 + 2^{-16}) \cdot \frac{1}{2} f_1^{-4} f_2^{-1} d^{-5} \epsilon^8}{2\sqrt{\left( \frac{\epsilon}{f_1 \sqrt{d}} \right)^8 - (1 + 2^{-16}) \cdot \frac{1}{2} f_1^{-4} f_2^{-1} d^{-5} \epsilon^8}} \leq 0.27 f_2^{-1} d^{-3} \epsilon^4. \tag{277}$$

By (C.2),

$$\sigma_{\min}\left( W_1(T_1)^H W_2(T_1)^H W_2(T_1) W_1(T_1) + W(T_1)^H \right)$$

$$\geq \sigma_{\min}\left( \left( W(T_0)^H W(T_0) \right)^{1/2} + W(0)^H \right)$$

$$- \left\| W_1(T_1)^H W_2(T_1)^H W_2(T_1) W_1(T_1) - \left( W(T_0)^H W(T_0) \right)^{1/2} \right\|_{op} - \left\| \Delta_W(T_1) \right\|_{op} \tag{278}$$

$$\geq 0.72 f_2^{-1} d^{-3} \epsilon^4,$$

which further gives

$$\sigma_{\min}\left( W_1 + W_1' \right)\big|_{t=T_1}$$

$$= \sigma_{\min}\left( \left( W_1(T_1)^H W_2(T_1)^H W_2(T_1) \right)^{-1} \left( W_1(T_1)^H W_2(T_1)^H W_2(T_1) W_1(T_1) + W(T_1)^H \right) \right)$$

$$\geq \left( \frac{1}{\max_{j,k} |\sigma_k(W_j(T_1))|} \right)^3 \cdot \sigma_{\min}\left( W_1(T_1)^H W_2(T_1)^H W_2(T_1) W_1(T_1) + W(T_1)^H \right)$$

$$\geq \frac{\epsilon}{2 f_1^3 f_2 d^{9/2}}. \tag{279}$$

$\square$

## I.2 Stage 2: saddle avoidance stage

In this stage, we further assume $a \geq 32 f_1^{20} f_2 d^{13} \sigma_1(\Sigma) b$, where $b \geq \left( 5 \ln \left( \frac{\sigma_1^{1/4}(\Sigma)}{\epsilon} \right) + \frac{281}{8} \ln d + 23 \ln(4 f_1) + 7 \ln f_2 \right)$. Meanwhile, $\frac{\epsilon}{\sigma_1^{1/4}(\Sigma)} \leq \frac{1}{32 f_1^5 f_2 d^{53/8}}$.

From Theorem 48, for $\eta = O \left( a f_1^{-56} f_2^{-14} d^{-301/4} \epsilon^8 \sigma_1^{-9/2}(\Sigma) \right)$ with appropriate small constant,

$$
\begin{aligned}
e_\Delta(T_1) &\leq 2\sqrt{3 f_1^4 d^3 \epsilon^4 e^{-2b} + \eta O \left( a^{-1} f_1^{14} d^8 \epsilon^6 \sigma_1^2(\Sigma) \right)} \\
&\leq 2^{-44} f_1^{-21} f_2^{-7} d^{-269/8} \epsilon^7 \sigma_1^{-5/4}(\Sigma).
\end{aligned}
\tag{280}
$$

Moreover, $b - \ln b \geq 3 \ln \left( \frac{\sigma_1^{1/4}(\Sigma)}{\epsilon} \right) + \frac{303}{8} \ln d + 37 \ln(2 f_1) + 6 \ln f_2$. Thus for $\eta = O \left( a^{-1} f_1^{-44} f_2^{-10} d^{-219/4} \epsilon^4 \sigma_1^{-3/2}(\Sigma) \right)$ with appropriate small constant,

$$
\begin{aligned}
a e_\Delta(T_1) &\leq 2\sqrt{3 \cdot 2^{10} f_1^{44} f_2^2 d^{29} \epsilon^4 \sigma_1^2(\Sigma) \exp(-2(b - \ln b)) + \eta O \left( a f_1^{14} d^8 \epsilon^6 \sigma_1^2(\Sigma) \right)} \\
&\leq 2^{-30} f_1^{-15} f_2^{-5} d^{-187/8} \epsilon^5 \sigma_1^{1/4}(\Sigma).
\end{aligned}
\tag{281}
$$

**Theorem 52.** *At* $T_1 + T_2$, $T_2 = \frac{32 f_1^6 f_2^2 d^9}{\eta \sigma_1(\Sigma) \epsilon^2}$, *the following conclusions hold:*

$$
\begin{aligned}
\| W_1(T_1 + T_2) - W_1'(T_1 + T_2) \|_F &\leq 3 f_1 d \epsilon \\
\sigma_{\min}(W_1 + W_1')(T_1 + T_2) &\geq 2^{3/4} \sigma_1^{1/4}(\Sigma).
\end{aligned}
\tag{282}
$$

**Lemma 53.** $\mathcal{L}_{\mathrm{ori}}$ *is approximately non-increasing.*

*For* $t \in [0, +\infty)$, *suppose* $\left\| W_{j \in \{1,2,\cdots,N\}}(t) \right\|_{op} \leq M$, *then*

$$
\begin{aligned}
\mathcal{L}_{\mathrm{ori}}(t+1) - \mathcal{L}_{\mathrm{ori}}(t) &\leq -2\eta N \min_{j,k} |\sigma_k(W_j(t))|^{2(N-1)} \mathcal{L}_{\mathrm{ori}}(t) \\
&\quad + \eta^2 O \left( M^8 \left( M^4 + \sqrt{\mathcal{L}_{\mathrm{ori}}(t)} \right) \mathcal{L}_{\mathrm{ori}}(t) + a M^4 \sqrt{\mathcal{L}_{\mathrm{ori}}(t)} \mathcal{L}_{\mathrm{reg}}(t) \right) \\
&\quad + \eta^4 O \left( M^{16} \mathcal{L}_{\mathrm{ori}}(t)^2 + a^2 M^8 \mathcal{L}_{\mathrm{reg}}(t)^2 \right).
\end{aligned}
\tag{283}
$$

*Proof.* Following the continuous case (75), the change of product matrix satisfy

$$
\begin{aligned}
&\left\| W(t+1) - W(t) - \eta \sum_{j=1}^N W_{\prod_L, j+1}(t) W_{\prod_L, j+1}(t)^H (\Sigma - W(t)) W_{\prod_R, j-1}(t)^H W_{\prod_R, j-1}(t) \right\|_F \\
&= \eta^2 O \left( \max_{j \in \{1,2,3,4\}} \left\| \nabla_{W_j} \mathcal{L}(t) \right\|_F^2 \cdot \max_{j \in \{1,2,3,4\}} \| W_j(t) \|_{op}^2 \right).
\end{aligned}
\tag{284}
$$

Then

$$\mathcal{L}_{\mathrm{ori}}(t+1) - \mathcal{L}_{\mathrm{ori}}(t) = -\Re\left(\left\langle \Sigma - \frac{W(t+1) + W(t)}{2}, W(t+1) - W(t) \right\rangle\right)$$

$$= -\eta \sum_{j=1}^{N} \left\| W_{\prod_{L}, j+1}(t)^H \left(\Sigma - W(t)\right) W_{\prod_{R}, j-1}(t)^H \right\|_F^2$$

$$+ \eta^2 O\left(M^2 \sqrt{\mathcal{L}_{\mathrm{ori}}(t)} \cdot \max_{j \in \{1,2,3,4\}} \left\|\nabla_{W_j} \mathcal{L}(t)\right\|_F^2\right)$$

$$+ \eta^2 O\left(M^6 \cdot \max_{j \in \{1,2,3,4\}} \left\|\nabla_{W_j} \mathcal{L}_{\mathrm{ori}}(t)\right\|_F^2\right) \tag{285}$$

$$+ \eta^4 O\left(M^4 \cdot \max_{j \in \{1,2,3,4\}} \left\|\nabla_{W_j} \mathcal{L}(t)\right\|_F^4\right)$$

$$\leq -2\eta N \min_{j,k} |\sigma_k(W_j(t))|^{2(N-1)} \mathcal{L}_{\mathrm{ori}}(t)$$

$$+ \eta^2 O\left(M^8 \left(M^4 + \sqrt{\mathcal{L}_{\mathrm{ori}}(t)}\right) \mathcal{L}_{\mathrm{ori}}(t) + aM^4 \sqrt{\mathcal{L}_{\mathrm{ori}}(t)} \mathcal{L}_{\mathrm{reg}}(t)\right)$$

$$+ \eta^4 O\left(M^{16} \mathcal{L}_{\mathrm{ori}}(t)^2 + a^2 M^8 \mathcal{L}_{\mathrm{reg}}(t)^2\right).$$

$$\square$$

Below we further assume $\eta = O\left(\min\left(f_1^{-27} f_2^{-9} d^{-355/8} \epsilon^9 \sigma_1^{-15/4}(\Sigma), a^{-1} f_1^{-21} f_2^{-7} d^{-273/8} \epsilon^7 \sigma_1^{-9/4}(\Sigma)\right)\right)$ with appropriate small constant.

**Lemma 54.** *Bound of operator norms.*

*For $t \in [T_1, T_1 + T_2]$,*

$$\|\Sigma - W(t)\|_F \leq 1.01\sqrt{d}\sigma_1(\Sigma)$$

$$e_\Delta(t) \leq 1.01 \cdot 2^{-44} f_1^{-21} f_2^{-7} d^{-269/8} \epsilon^7 \sigma_1^{-5/4}(\Sigma)$$

$$ae_\Delta(t) \leq 1.01 \cdot 2^{-30} f_1^{-15} f_2^{-5} d^{-187/8} \epsilon^5 \sigma_1^{1/4}(\Sigma) \tag{286}$$

$$\|W\|_{op} \leq \|W\|_F \leq 3\sqrt{d}\sigma_1(\Sigma)$$

$$\max_j \|W_j\|_{op} \leq \max_j \|W_j\|_F \leq \sqrt{2}d^{1/8}\sigma_1^{1/4}(\Sigma).$$

*Proof.* We first prove that if the first three inequalities hold at some time $t$, then the rest follows. Then we prove the first three by mathematical induction.

1. For some $t$, it the first two hold, then

$$\|W(t)\|_{op} \leq \|W(t)\|_F \leq \|\Sigma - W(t)\|_F + \|\Sigma\|_F \leq 3\sqrt{d}\sigma_1(\Sigma). \tag{287}$$

For the last inequality, prove by contradiction. (Omit $t$ here)

Suppose $\max_j \|W_j\|_{op} \geq \sqrt{2}d^{1/8}\sigma_1^{1/4}(\Sigma)$, then

$$e_\Delta(t) \leq 1.01 e_\Delta(T_1) \leq 2^{-15} \max_j \|W_j\|_{op}^2. \tag{288}$$

Thus for $t > T_1$,

$$\|W\|_{op}^2 = \left\|W_4 W_3 W_2 W_1 W_1^H W_2^H W_3^H W_4^H\right\|_{op}$$

$$\geq \left\|W_4 W_4^H\right\|_{op} - \left\|W_4 W_3 W_2 \Delta_{12} W_2^H W_3^H W_4^H\right\|_{op}$$

$$- \left\|W_4 W_3 \Delta_{23} W_2 W_2^H W_3^H W_4^H\right\|_{op} - \left\|W_4 W_3 W_2 W_2^H \Delta_{23} W_3^H W_4^H\right\|_{op}$$

$$- \left\|W_4 \Delta_{34} \left(W_3 W_3^H\right)^2 W_4^H\right\|_{op} - \left\|W_4 W_3 W_3^H \Delta_{34} W_3 W_3^H W_4^H\right\|_{op} - \left\|W_4 \left(W_3 W_3^H\right)^2 \Delta_{34} W_4^H\right\|_{op}$$

$$\geq \left(\max_j \|W_j\|_{op}^2 - 3e_\Delta\right)^4 - 6e_\Delta \max_j \|W_j\|_{op}^6 > 15\sqrt{d}\sigma_1(\Sigma),$$

$$(289)$$

which contradicts inequality (287).

2. Mathematical induction.

For $t = T_1$,

$$\|\Sigma - W(T_1)\|_F \leq \|\Sigma\|_F + \|W(T_1)\|_F \leq \left(1 + 2^{-39}\right)\sqrt{d}\sigma_1(\Sigma). \qquad (290)$$

Suppose for $t' \in [T_1, t]$ ($T_1 \leq t < T_2$), the first two properties hold. Denote $M = \max_j \|W_j(t' \in [T_1, t])\|_{op}$. By invoking Lemma 53 and 29, at $t+1$,

$$\mathcal{L}_{\text{ori}}(t+1) = \mathcal{L}_{\text{ori}}(T_1) + \eta^2(t - T_1)O\left(M^8\left(M^4 + \sqrt{\mathcal{L}_{\text{ori}}(T_1)}\right)\mathcal{L}_{\text{ori}}(T_1) + aM^4\sqrt{\mathcal{L}_{\text{ori}}(T_1)}\mathcal{L}_{\text{reg}}(T_1)\right)$$

$$+ \eta^4(t - T_1)O\left(M^{16}\mathcal{L}_{\text{ori}}(T_1)^2 + a^2 M^8 \mathcal{L}_{\text{reg}}(T_1)^2\right)$$

$$= \mathcal{L}_{\text{ori}}(T_1) + \eta^2 T_2 O\left(d^2 \sigma_1(\Sigma)^4 + d\sigma_1(\Sigma)^2(ae_\Delta(T_1))^2\right) \leq 1.01^2 \sqrt{d}\sigma_1(\Sigma).$$

$$(291)$$

Note that $\mathcal{L}_{\text{ori}} = \frac{a}{4}e_\Delta^2$. Under $\eta = O\left(\min\left(f_1^{-27} f_2^{-9} d^{-355/8} \epsilon^9 \sigma_1^{-15/4}(\Sigma), a^{-1} f_1^{-21} f_2^{-7} d^{-273/8} \epsilon^7 \sigma_1^{-9/4}(\Sigma)\right)\right)$ with appropriate small constant,

$$\mathcal{L}_{\text{reg}}(t+1) \leq \mathcal{L}_{\text{reg}}(T_1) + \eta^2(t - T_1)O\left(a^2 M^4 \mathcal{L}_{\text{reg}}(t) + \sqrt{a\mathcal{L}_{\text{reg}}(t)}M^6 \mathcal{L}_{\text{ori}}(t)\right)$$

$$+ \eta^4(t - T_1)O\left(aM^{12}\mathcal{L}_{\text{ori}}(t)^2 + a^3 M^4 \mathcal{L}_{\text{reg}}(t)^2\right)$$

$$\leq \mathcal{L}_{\text{reg}}(T_1) + \eta^2 T_2 O\left(\sqrt{a\mathcal{L}_{\text{reg}}(t)}M^6 \mathcal{L}_{\text{ori}}(t)\right) + \eta^4 T_2 O\left(aM^{12}\mathcal{L}_{\text{ori}}(t)^2\right)$$

$$\leq \frac{1.01^2}{4}\min\left(a \cdot \left[2^{-44} f_1^{-21} f_2^{-7} d^{-269/8} \epsilon^7 \sigma_1^{-5/4}(\Sigma)\right]^2, \frac{1}{a} \cdot \left[2^{-30} f_1^{-15} f_2^{-5} d^{-187/8} \epsilon^5 \sigma_1^{1/4}(\Sigma)\right]^2\right).$$

$$(292)$$

This completes the proof.

$$\square$$

**Lemma 55.** *Bound of $\left\|W_2^{-1}\right\|_{op}$ and relevant term.*

*For $t \in [T_1, T_1 + T_2]$,*

$$\left\|W_2^{-1}(t)\right\|_{op} \leq 128 f_1^6 f_2^2 d^{77/8} \epsilon^{-2} \sigma_1^{1/4}(\Sigma), \qquad (293)$$

$$e_\Delta(t)\left\|W_2^{-1}(t)\right\|_{op}^2 \leq 1.01 \cdot 2^{-30} f_1^{-9} f_2^{-3} d^{-115/8} \epsilon^3 \sigma_1^{-3/4}(\Sigma). \qquad (294)$$

*Proof.* We begin with the update of $W_2^{-1}$. From Lemma 17,

$$\left\| W_2^{-1}(t+1) - W_2^{-1}(t) \right.$$
$$\left. - \eta \left[ -R(t)W_4(t)^H(\Sigma - W(t))W_1(t)^H W_2(t)^{-1} - a\Delta_{12}(t)W_2(t)^{-1} + aW_2(t)^{-1}\Delta_{23}(t) \right] \right\|_{op}$$
$$\leq \eta^2 \|W_2(t)^{-1}\|_{op}^2 \|W_2(t+1)^{-1}\|_{op} \|\nabla_{W_2}\mathcal{L}(t)\|_{op}^2. \tag{295}$$

By triangular inequality,

$$\left\| W_2(t+1)^{-1} \right\|_{op} - \left\| W_2(t)^{-1} \right\|_{op} \leq \eta \left\| R(t) \right\|_{op} \left\| W_4(t) \right\|_{op} \left\| \Sigma - W(t) \right\|_{op} \left\| W_1(t)^H W_2(t)^{-1} \right\|_{op}$$
$$+ \eta a \left\| \Delta_{12}(t) \right\|_{op} \left\| W_2(t)^{-1} \right\|_{op} + \eta a \left\| W_2(t)^{-1} \right\|_{op} \left\| \Delta_{23}(t) \right\|_{op}$$
$$+ \eta^2 \|W_2(t)^{-1}\|_{op}^2 \|W_2(t+1)^{-1}\|_{op} \|\nabla_{W_2}\mathcal{L}(t)\|_{op}^2. \tag{296}$$

From

$$\left\| R \right\|_{op} \leq \sqrt{1 + \frac{1}{\sigma_{\min}^2(W_2)} \cdot \left\| \Delta_{23} \right\|_{op}}$$

$$\left\| W_1^H W_2^{-1} \right\|_{op} = \sqrt{\left\| W_2^{H-1} W_1 W_1^H W_2^{-1} \right\|_{op}} = \sqrt{\left\| I + W_2^{H-1}\Delta_{12}W_2^{-1} \right\|} \leq \sqrt{1 + e_\Delta \left\| W_2^{-1} \right\|_{op}^2}. \tag{297}$$

Further we have

$$\left\| W_2(t+1)^{-1} \right\|_{op} - \left\| W_2(t)^{-1} \right\|_{op} \leq 2\sqrt{2}\eta \left( 1 + e_\Delta(t) \left\| W_2(t)^{-1} \right\|_{op}^2 \right) d^{5/8}\sigma_1^{5/4}(\Sigma)$$
$$+ \sqrt{2}\eta a e_\Delta(t) \left\| W_2(t)^{-1} \right\|_{op}$$
$$+ \eta^2 O \left( \|W_2(t)^{-1}\|_{op}^2 \|W_2(t+1)^{-1}\|_{op} \|\nabla_{W_2}\mathcal{L}(t)\|_{op}^2 \right). \tag{298}$$

Combine with Lemma 54, for $t \geq T_1$ such that (293) holds,

$$\left\| W_2(t+1)^{-1} \right\|_{op} - \left\| W_2(t)^{-1} \right\|_{op}$$
$$\leq 2\sqrt{2}(1 + 1.01 \cdot 2^{-30})\eta d^{5/8}\sigma_1^{5/4}(\Sigma) + 2^{-22}\eta f_1^{-9}f_2^{-3}d^{-55/4}\epsilon^3\sigma_1^{1/2}(\Sigma)$$
$$+ \eta^2 O \left( f_1^{18}f_2^6 d^{245/8}\epsilon^{-6}\sigma_1^{17/4}(\Sigma) \right)$$
$$\leq 2\sqrt{2}(1 + 2^{-20})\eta d^{5/8}\sigma_1^{5/4}(\Sigma). \tag{299}$$

From Theorem 48, $\max \left( \left\| W_2(T_1)^{-1} \right\|_{op}, \left\| W_3(T_1)^{-1} \right\|_{op} \right) \leq \frac{1}{\min_{j,k} |\sigma_k(W_j(T_1))|} \leq \frac{f_1\sqrt{d}}{(1-2^{-17})\epsilon}$, then the proof of the first inequality is completed via integration during the time interval $[T_1, T_1+T_2]$. The second inequality follows immediately.

□

**Remark 18.** *This Lemma verifies that $W_{2,3}^{-1}$ are bounded (consequently $W_{2,3}$ are full rank), then $R$ is well defined throughout this stage. For $t > T_1 + T_2$, further analysis shows that the minimum singular values of $W_2$ and $W_3$ are lower bounded by $\Omega(\sigma_1^{1/4}(\Sigma))$.*

**Lemma 56.** *Skew-Hermitian error in saddle avoidance stage, gradient descent. For $t \in [T_1, T_1 + T_2]$,*

$$\left\| W_1 - W_2^{-1}W_3^H W_4^H \right\|_F \leq 3f_1 d\epsilon. \tag{300}$$

*Proof.* From Lemma 55, for $t \in [T_1, T_1 + T_2]$,

$$
\max\left( \left\| R^H R - I \right\|_{op}, \left\| I - R R^H \right\|_{op} \right) \leq e_\Delta \left\| W_2^{-1} \right\|_{op}^2
$$
$$
\leq 1.01 \cdot 2^{-30} f_1^{-9} f_2^{-3} d^{-115/8} \epsilon^3 \sigma_1^{-3/4}(\Sigma), \tag{301}
$$

$$
\left\| M_1 - M_1' \right\|_{op} \leq \sqrt{6} \cdot \frac{\max_{j,k} \sigma_k^2(W_j)}{\sigma_{\min}^2(W_2)} e_\Delta
$$
$$
\leq 2^{-27} f_1^{-9} f_2^{-3} d^{-113/8} \epsilon^3 \sigma_1^{-1/4}(\Sigma), \tag{302}
$$

$$
\left\| M_2 - \frac{M_1 + M_1'}{2} \right\|_{op} \leq \left\| \Delta_{12} \right\|_{op} + \frac{1}{2} \left\| M_1 - M_1' \right\|_{op} \leq \left[ 1 + \frac{\sqrt{6}}{2} \cdot \frac{\max_{j,k} \sigma_k^2(W_j)}{\sigma_{\min}^2(W_2)} \right] e_\Delta \tag{303}
$$
$$
\leq 2^{-28} f_1^{-9} f_2^{-3} d^{-113/8} \epsilon^3 \sigma_1^{-1/4}(\Sigma).
$$

Consequently:

$$
\left\| R \right\|_{op} \leq \sqrt{1 + e_\Delta \left\| W_2^{-1} \right\|_{op}^2} \leq 1 + 1.01 \cdot 2^{-31} f_1^{-9} f_2^{-3} d^{-115/8} \epsilon^3 \sigma_1^{-3/4}(\Sigma), \tag{304}
$$

$$
\left\| W_1' \right\|_{op} \leq \left\| W_1' \right\|_F \leq \sqrt{2} d^{1/8} \sigma_1^{1/4}(\Sigma) \left\| R \right\|_{op} \leq \left( 1 + 1.01 \cdot 2^{-31} \right) \sqrt{2} d^{1/8} \sigma_1^{1/4}(\Sigma), \tag{305}
$$

$$
\left\| \frac{M_1 + M_1'}{2} \right\|_{op} \leq \left\| M_2 \right\|_{op} + \left\| M_2 - \frac{M_1 + M_1'}{2} \right\|_{op} \leq \left( 1 + 2^{-29} \right) 2 d^{1/4} \sigma_1^{1/2}(\Sigma), \tag{306}
$$

$$
\left\| M_1' M_2 M_1 - M_1 M_2 M_1' \right\|_{op} \leq \left\| M_1 - M_1' \right\| \left\| M_2 \right\| \left\| M_1 + M_1' \right\|
$$
$$
\leq \left( 1 + 2^{-29} \right) 2^{-25} f_1^{-9} f_2^{-3} d^{-109/8} \epsilon^3 \sigma_1^{3/4}(\Sigma). \tag{307}
$$

By combining all results above, for $t \in [T_1, T_1 + T_2 - 1]$ such that $\left\| W_1 - W_1' \right\|_F \leq 3 f_1 d \epsilon$ holds,

$$
\left\| W_1(t+1) - W_1'(t+1) \right\|_F^2 - \left\| W_1(t) - W_1'(t) \right\|_F^2
$$
$$
\leq -2\eta \sigma_1(\Sigma) \sigma_{\min}(W_2)^2 \left\| W_1(t) - W_1'(t) \right\|_F^2
$$
$$
+ \eta \| M_2(t) \|_F \left\| M_1'(t) - M_1(t) \right\|_{op} \| M_2(t) \|_{op} \left( \left\| W_1'(t) \right\|_{op} + \left\| W_1(t) \right\|_{op} \right) \left\| W_1(t) - W_1'(t) \right\|_F
$$
$$
+ 2\eta \left\| -M_1'(t) M_2(t) M_1(t) + M_1(t) M_2(t) M_1'(t) \right\|_{op} \left\| W_1'(t) \right\|_F \left\| W_1(t) - W_1'(t) \right\|_F
$$
$$
+ 2\eta \max_j \| W_j(t) \|_{op}^3 \| \Sigma - W(t) \|_F \left( \left\| R(t)^H R(t) - I \right\|_{op} + \left\| I - R(t) R(t)^H \right\|_{op} \right) \left\| W_1(t) - W_1'(t) \right\|_F
$$
$$
+ 2\eta a e_\Delta(t) \left\| W_1(t) - W_1'(t) \right\|_F^2
$$
$$
+ 4\eta a e_\Delta(t) \left\| W_2(t)^{-1} \right\|_{op} \| W_2(t) \|_F \left\| W_1'(t) \right\|_{op} \left\| W_1(t) - W_1'(t) \right\|_F
$$
$$
+ \eta^2 O\left( \left[ \max_{j \in \{1,2,3,4\}} \| W_j(t) \|_{op} \| \Sigma - W(t) \|_F + a e_\Delta(t) \left\| W_2(t)^{-1} \right\|_{op} \right]^2 \right.
$$
$$
\left. \cdot \max_{j \in \{1,2,3,4\}} \| W_j(t) \|_{op}^5 \cdot \| W_2(t+1)^{-1} \|_{op} \right)
$$
$$
\leq -2\eta \sigma_1(\Sigma) \sigma_{\min}(W_2)^2 \left\| W_1(t) - W_1'(t) \right\|_F^2 + 2^{-17} \eta f_1^{-8} f_2^{-3} d^{-25/2} \epsilon^4 \sigma_1(\Sigma). \tag{308}
$$

From Theorem 48, at $t = T_1$,

$$\|W_1(T_1) - W_1'(T_1)\|_F \le \|W_1(T_1)\|_F + \|W_1'(T_1)\|_F \le \|W_1(T_1)\|_F + \|W_4(T_1)\|_F \|R(T_1)\|_{op}$$
$$\le \left(1 + 2^{-20}\right) 2 f_1 d\epsilon. \tag{309}$$

Thus $\|W_1(t) - W_1'(t)\|_F^2 \le \sqrt{\left[(1 + 2^{-20}) 2 f_1 d\epsilon\right]^2 + 2^{-17} f_1^{-8} f_2^{-3} d^{-25/2} \epsilon^4 \sigma_1(\Sigma) \eta(t - T_1)}$, when both $t \in [T_1, T_1 + T_2]$ and $\|W_1(t) - W_1'(t)\|_F^2 \le 3 f_1 d\epsilon$ hold. Then

$$\|W_1(T_1 + T_2) - W_1'(T_1 + T_2)\|_F^2 \le \sqrt{\left[(1 + 2^{-20}) 2 f_1 d\epsilon\right]^2 + 2^{-17} f_1^{-8} f_2^{-3} d^{-25/2} \epsilon^4 \sigma_1(\Sigma) \eta T_2}$$
$$\le \sqrt{\left[(1 + 2^{-20}) 2 f_1 d\epsilon\right]^2 + 2^{-12} f_1^{-2} f_2^{-1} d^{-7/2} \epsilon^2} < 3 f_1 d\epsilon, \tag{310}$$

which completes the proof.

$\square$

**Lemma 57.** *The minimum eigenvalue of Hermitian term. For $t = T_1 + T_2$,*

$$\sigma_{\min}\left(W_1 + W_2^{-1} W_3^H W_4^H\right)|_{t=T_1+T_2} \ge 2^{3/4} \sigma_1^{1/4}(\Sigma). \tag{311}$$

*Proof.* We analyze the dynamics of $\lambda_{\min}\left((W_1 + W_1')^H (W_1 + W_1')\right) = \sigma_{\min}^2$.

From $\left\|M_2 - \frac{M_1 + M_1'}{2}\right\|_{op} \le 2^{-28} f_1^{-9} f_2^{-3} d^{-113/8} \epsilon^3 \sigma_1^{-1/4}(\Sigma)$ and $\left\|\frac{M_1 + M_1'}{2}\right\|_{op} \le \left(1 + 2^{-29}\right) 2 d^{1/4} \sigma_1^{1/2}(\Sigma)$, define

$$E(t) := \sigma_1(\Sigma) \left(M_2(t) - \frac{M_1(t) + M_1'(t)}{2}\right) - \left(M_2(t) \left(\frac{M_1(t) + M_1'(t)}{2}\right) M_2(t) - \left(\frac{M_1(t) + M_1'(t)}{2}\right)^3\right). \tag{312}$$

Then

$$\|E(t)\|_{op} \le 2^{-28} f_1^{-9} f_2^{-3} d^{-113/8} \epsilon^3 \sigma_1^{3/4}(\Sigma) + \left(1 + 2^{-28}\right) 2^{-24} f_1^{-9} f_2^{-3} d^{-109/8} \epsilon^3 \sigma_1^{3/4}(\Sigma)$$
$$\le \left(1 + 2^{-4} + 2^{-28}\right) 2^{-24} f_1^{-9} f_2^{-3} d^{-109/8} \epsilon^3 \sigma_1^{3/4}(\Sigma). \tag{313}$$

By Lemma 56, $\|W_1 - W_1'\|_{op} \le \|W_1 - W_1'\|_F \le 3 f_1 d\epsilon$, and under $\sigma_{\min}(t) \ge \frac{\epsilon}{2 f_1^3 f_2 d^{9/2}}$,

$$\sigma_{\min}(t+1)^2 \ge \lambda_{\min}\left(W_{\text{new}}(t)^H W_{\text{new}}(t)\right) - 2^{-18} \sigma_1(\Sigma) \sigma_{\min}(t)^4, \tag{314}$$

where

$$W_{\text{new}}(t) = \left(I + \eta \left[\sigma_1(\Sigma) \left(\frac{M_1(t) + M_1'(t)}{2}\right) - \left(\frac{M_1(t) + M_1'(t)}{2}\right)^3 + E(t)\right]\right) (W_1(t) + W_1'(t)). \tag{315}$$

Denote $P = \frac{W_1 + W_1'}{2}$, $Q = \frac{W_1 - W_1'}{2}$. Notice that $PP^H + QQ^H = \frac{M_1 + M_1'}{2}$. Then by invoking Lemma 20 (omit $t$ here) the first term becomes

$$\lambda_{\min}\left(W_{\text{new}}^H W_{\text{new}}\right) = \lambda_{\min}\left(W_{\text{new}} W_{\text{new}}^H\right)$$

$$= 4\lambda_{\min}\left(\left(I + \eta\left[\sigma_1(\Sigma)\left(PP^H + QQ^H\right) - \left(PP^H + QQ^H\right)^3 + E\right]\right) PP^H\right.$$

$$\left.\cdot\left(I + \eta\left[\sigma_1(\Sigma)\left(PP^H + QQ^H\right) - \left(PP^H + QQ^H\right)^3 + E\right]\right)\right)$$

$$\geq \sigma_{\min}^2 + 8\eta\left(\sigma_1(\Sigma) - 2\|Q\|_{op}^2\left\|\frac{M_1 + M_1'}{2}\right\|_{op}\right)\left(\frac{\sigma_{\min}^2}{4}\right)^2$$

$$- 8\eta\left\|\frac{M_1 + M_1'}{2}\right\|_{op}\left(\frac{\sigma_{\min}^2}{4}\right)^3$$

$$- 8\eta\left(\|E\|_{op} + \|Q\|_{op}^4\left\|\frac{M_1 + M_1'}{2}\right\|_{op}\right)\left(\frac{\sigma_{\min}^2}{4}\right)$$

$$+ \eta^2 O\left(\left(\sigma_1(\Sigma)^2\left\|\frac{M_1 + M_1'}{2}\right\|_{op}^2 + \left\|\frac{M_1 + M_1'}{2}\right\|_{op}^6 + \|E\|_{op}^2\right)\left\|\frac{M_1 + M_1'}{2}\right\|_{op}\right).$$

$$\tag{316}$$

Notice $\|Q\|_{op} = \frac{1}{2}\|W_1 - W_1'\|_F \leq \frac{3}{2}f_1 d\epsilon \leq \sigma_k \cdot 3f_1^4 f_2 d^{11/2}$, $\epsilon \leq \frac{1}{32 f_1^5 f_2 d^{53/8}}\sigma_1^{1/4}(\Sigma)$, then under $\sigma_{\min}(t) \geq \frac{\epsilon}{2 f_1^3 f_2 d^{9/2}}$,

$$\sigma_{\min}(t+1)^2 \geq \sigma_{\min}(t)^2 + (2^{-1} - 81(1 + 2^{-4})2^{-10})\eta\sigma_1(\Sigma)\sigma_{\min}(t)^4 - \frac{1}{32}\eta\sigma_{\min}(t)^8. \tag{317}$$

Notice that $\sigma_{\min}(t)$ is bounded by $O\left(d^{1/8}\sigma_1^{1/4}(\Sigma)\right)$. By taking reciprocal,

$$\frac{1}{\sigma_{\min}(t+1)^2} \leq \frac{1}{\sigma_{\min}(t)^2} + \frac{(2^{-1} - 81(1 + 2^{-4})2^{-10})\eta\sigma_1(\Sigma)\sigma_{\min}(t)^4 - \frac{1}{32}\eta\sigma_{\min}(t)^8}{\sigma_{\min}(t)^4 + (2^{-1} - 81(1 + 2^{-4})2^{-10})\eta\sigma_1(\Sigma)\sigma_{\min}(t)^6 - \frac{1}{32}\eta\sigma_{\min}(t)^{10}}$$

$$\leq \frac{1}{\sigma_{\min}(t)^2} + \frac{3}{8}\eta\sigma_1(\Sigma) - \frac{1}{32}\eta\sigma_{\min}(t)^4.$$

$$\tag{318}$$

This indicates that $\sigma_{\min}(t)$ takes at most time $\Delta t' = \frac{1}{\frac{1}{8}\eta\sigma_1(\Sigma)}\left[\frac{1}{\sigma_{\min}(t=0)^2} - \frac{1}{\left(2^{3/4}\sigma_1^{1/4}(\Sigma)\right)^2}\right] < T_2$

to increase to $2^{3/4}\sigma_1^{1/4}(\Sigma)$, and never decrease to less than $2^{3/4}\sigma_1^{1/4}(\Sigma)$ afterwards (in $t \in [T_1 + \Delta t', T_2]$).

$\square$

## I.3 STAGE 3: LOCAL CONVERGENCE STAGE

In this stage, we analysis the time to reach $\epsilon_{\text{conv}}$-convergence, that is

$$T(\epsilon_{\text{conv}}, \eta) = \inf_t\{\mathcal{L}(t) \leq \epsilon_{\text{conv}}\}. \tag{319}$$

**Theorem 58.** *Local convergence.*

*For $t \in [T_1 + T_2, +\infty)$,*

$$\mathcal{L}_{\text{ori}}(t) \leq \mathcal{L}_{\text{ori}}(T_1 + T_2) \exp\left(-\eta\sigma_1^{3/2}(\Sigma)(t - T_1 - T_2)\right)$$

$$\mathcal{L}_{\text{reg}}(t) \leq l_{\text{reg}} \exp\left(-\eta\sigma_1^{3/2}(\Sigma)(t - T_1 - T_2)\right)$$

$$\sigma_{\min}\left(W_1(t) + W_1'(t)\right) \geq 2^{3/4}\sigma_1^{1/4}(\Sigma)$$

$$\|W_1(t) - W_1'(t)\|_F \leq 3f_1 d\epsilon, \tag{320}$$

*where* $\mathcal{L}_{\text{ori}}(T_1 + T_2) = \frac{1.01^2}{2} \cdot d\sigma_1^2(\Sigma)$, *and* $l_{\text{reg}} = \min\left(\frac{a}{4}\left(1.01 \cdot 2^{-44}f_1^{-21}f_2^{-7}d^{-269/8}\epsilon^7\sigma_1^{-5/4}(\Sigma)\right)^2, \frac{1}{4a}\left(1.01 \cdot 2^{-30}f_1^{-15}f_2^{-5}d^{-187/8}\epsilon^5\sigma_1^{1/4}(\Sigma)\right)^2\right)$.

*Proof.* Prove by induction.

At $t = T_2$ these properties holds.

Suppose at some time $t \in [T_2, +\infty)$ they holds, then follow the same arguments in Lemma 54, $\max_j \|W_j(t)\|_{op} \leq \sqrt{2}d^{1/8}\sigma_1^{1/4}(\Sigma)$.

To address the bound of $\left\|W_2^{-1}\right\|_{op}$,

$$\left\|\frac{M_1(t) - M_1'(t)}{2}\right\|_{op} \leq \|W_1(t) - W_1'(t)\|_{op} \left\|\frac{W_1(t) + W_1'(t)}{2}\right\|_{op} \leq 8f_1 d^{9/8}\sigma_1^{1/4}(\Sigma)\epsilon$$

$$\left\|M_2(t) - \frac{M_1(t) + M_1'(t)}{2}\right\|_{op} \leq \|\Delta_{12}(t)\|_{op} + \left\|\frac{M_1(t) - M_1'(t)}{2}\right\|_{op} \leq 16f_1 d^{9/8}\sigma_1^{1/4}(\Sigma)\epsilon$$

$$\sigma_{\min}(W_2(t)) = \sqrt{\lambda_{\min}(M_2(t))} \geq \sqrt{\lambda_{\min}\left(\frac{M_1(t) + M_1'(t)}{2}\right) - 16f_1 d^{9/8}\sigma_1^{1/4}(\Sigma)\epsilon}$$

$$\geq \sqrt{\sigma_{\min}^2\left(\frac{W_1(t) + W_1'(t)}{2}\right) - 16f_1 d^{9/8}\sigma_1^{1/4}(\Sigma)\epsilon} \geq \frac{1}{2^{3/8}}\sigma_1^{1/4}(\Sigma). \tag{321}$$

Similarly, $\min_{j,k}(\sigma_k(W_j(t))) \geq \frac{1}{2^{3/8}}\sigma_1^{1/4}(\Sigma)$.

Then following the derivations in Lemma 56 and 57,

$$\|W_1(t+1) - W_1'(t+1)\|_F^2 \leq \left(1 - 2\eta\sigma_1(\Sigma)\sigma_{\min}(W_2)^2\right)\|W_1(t) - W_1'(t)\|_F^2 + 2^{-17}\eta f_1^{-8}f_2^{-3}d^{-25/2}\epsilon^4\sigma_1(\Sigma)$$

$$\leq \left(1 - \eta\sigma_1^{3/2}(\Sigma)\right)\|W_1(t) - W_1'(t)\|_F^2 + 2^{-17}\eta f_1^{-8}f_2^{-3}d^{-25/2}\epsilon^4\sigma_1(\Sigma) \leq 3f_1 d\epsilon$$

$$\frac{1}{\sigma_{\min}\left(W_1(t+1) + W_1'(t+1)\right)^2} \leq \frac{1}{\sigma_{\min}(t)^2} + \frac{3}{8}\eta\sigma_1(\Sigma) - \frac{1}{32}\eta\sigma_{\min}(t)^4 < \frac{1}{\left(2^{3/4}\sigma_1^{1/4}(\Sigma)\right)^2}. \tag{322}$$

Then by Theorem 53 and 29,

$$\mathcal{L}_{\text{ori}}(t+1) \leq \mathcal{L}_{\text{ori}}(t) - 2^{3/4}\eta\sigma_1^{3/2}(\Sigma)\mathcal{L}_{\text{ori}}(t)$$

$$+ \eta^2 O\left(\max_j \|W_j(t)\|_{op}^8 \left(\max_j \|W_j(t)\|_{op}^4 + \sqrt{\mathcal{L}_{\text{ori}}(t)}\right)\mathcal{L}_{\text{ori}}(t) + a\max_j \|W_j(t)\|_{op}^4 \sqrt{\mathcal{L}_{\text{ori}}(t)}\mathcal{L}_{\text{reg}}(t)\right)$$

$$+ \eta^4 O\left(\max_j \|W_j(t)\|_{op}^{16}\mathcal{L}_{\text{ori}}(t)^2 + a^2\max_j \|W_j(t)\|_{op}^8\mathcal{L}_{\text{reg}}(t)^2\right)$$

$$\leq \left(1 - \eta\sigma_1^{3/2}(\Sigma)\right)\mathcal{L}_{\text{ori}}(t), \tag{323}$$

$$\mathcal{L}_{\text{reg}}(t+1) \leq \left(1 - \frac{1}{3}\eta a d^{-1/4}\sigma_1^{1/2}(\Sigma)\right) \cdot \mathcal{L}_{\text{reg}}(t) + \eta^2 O\left(a^2 M^4 \mathcal{L}_{\text{reg}}(t) + \sqrt{a\mathcal{L}_{\text{reg}}(t)}M^6\mathcal{L}_{\text{ori}}(t)\right)$$

$$+ \eta^4 O\left(aM^{12}\mathcal{L}_{\text{ori}}(t)^2 + a^3 M^4 \mathcal{L}_{\text{reg}}(t)^2\right)$$

$$\leq \left(1 - \frac{1}{4}\eta a d^{-1/4}\sigma_1^{1/2}(\Sigma)\right) \cdot \mathcal{L}_{\text{reg}}(t) \leq \left(1 - \eta a d^{-1/4}\sigma_1^{3/2}(\Sigma)\right) \cdot \mathcal{L}_{\text{reg}}(t).$$

$$(324)$$

This completes the proof.

$\square$

By Combining the three-stage results, the global convergence guarantee of Theorem 47 is proved.

## J EXPLANATION OF MAIN RESULT

This section expands the discussion of main convergence result Theorem 47.

### J.1 PROOF OF EXAMPLE FOR TIGHTNESS

This section completes the proof of Example below Theorem 1 for tightness analysis.

Firstly, since all $w_j$ are initialized to the same value, from the property of balancedness all $w_j$ remain identical through the optimization.

To solve the differential equation of $\frac{\mathrm{d}w_j}{\mathrm{d}t} = (\sigma_1 - w_j^4)w_j^3$,

$$
\begin{aligned}
T(w_j = (1-\gamma)\sigma_1^{1/4}) &= \int_\epsilon^{(1-\gamma)\sigma_1^{1/4}} \frac{1}{(\sigma_1 - w_j^4)w_j^3}\mathrm{d}w_j \\
&= \sigma_1^{-3/2}\int_{\epsilon/\sigma_1^{1/4}}^{1-\gamma} \frac{1}{(1-x^4)x^3}\mathrm{d}x \\
&= \sigma_1^{-3/2}\left[\int_{\epsilon/\sigma_1^{1/4}}^{2^{-1/4}} \frac{1}{(1-x^4)x^3}\mathrm{d}x + \int_{2^{-1/4}}^{1-\gamma} \frac{1}{(1-x^4)x^3}\mathrm{d}x\right] \\
&= \sigma_1^{-3/2}\left[\Theta\left(\int_{\epsilon/\sigma_1^{1/4}}^{2^{-1/4}} \frac{1}{x^3}\mathrm{d}x\right) + \Theta\left(\int_{2^{-1/4}}^{1-\gamma} \frac{1}{1-x}\mathrm{d}x\right)\right] \\
&= \sigma_1^{-3/2}\left[\Theta\left(\sigma_1^{1/2}/\epsilon^2\right) + \Theta\left(\ln(1/\gamma)\right)\right]
\end{aligned}
$$

$$(325)$$

By setting $\gamma$ through $\epsilon_{\text{conv}} = \frac{1}{2}[1-(1-\gamma)^4]^2\sigma_1^2$, $\gamma = \Theta(\epsilon_{\text{conv}}/\sigma_1^2)$. Then it takes $T(\mathcal{L} \leq \epsilon_{\text{conv}}) = \left[\Theta\left(\sigma_1^{-1}\epsilon^{-2}\right) + \Theta\left(\sigma_1^{-3/2}\ln(1/\gamma)\right)\right]$ This completes the proof of tightness.

### J.2 ILLUSTRATION FOR THE EXPONENT OF $\sigma_1$ IN INITIALIZATION SCALE AND CONVERGENCE TIME

We consider arbitrary $N$-layer matrix factorization under gradient flow setting(gradient descent follows the same argument). Then for fixed condition number $\kappa := \sigma_1(\Sigma)/\sigma_d(\Sigma)$, the requirements for initialization scale $\epsilon \propto \sigma_1^{1/N}(\Sigma)$, while the training time scales by $\sigma_1^{-2(N-1)/N}(\Sigma)$.

Suppose the target matrix is scaled by a positive real constant $\lambda \in \mathbb{R}^+$, then the new dynamics becomes

$$\frac{\mathrm{d}}{\mathrm{d}t} W_j = \left( \prod_{k=N}^{j+1} W_k \right) (\lambda \Sigma - W) \left( \prod_{k=j-1}^{1} W_k \right). \tag{326}$$

By setting $W_j' = \lambda^{1/N} W_j$, $t' = \lambda^{-2(N-1)/N}$ (correspondingly the initialization scale $\epsilon' = \lambda^{1/N} \epsilon$), then the dynamics becomes the form of

$$\frac{\mathrm{d}}{\mathrm{d}t'} W_j' = \left( \prod_{k=N}^{j+1} W_k' \right) (\Sigma - W') \left( \prod_{k=j-1}^{1} W_k' \right). \tag{327}$$

Then $W_j'(t')$ shares *exactly the same dynamics with $W_j(t)$ before scaling* . Thus for fixed conditional number $\kappa := \sigma_1(\Sigma)/\sigma_d(\Sigma)$ (for Theorem 1 and 2, $\kappa = 1$) or to say, relative size of target singular values, the initialization scale $\epsilon \propto \sigma_1^{1/N}(\Sigma)$, convergence time $T \propto \sigma_1^{-2(N-1)/N}(\Sigma)$. For $N = 4$, $T \propto \sigma_1^{-3/2}(\Sigma)$; for $N = 2$, $T \propto \sigma_1^{-1}(\Sigma)$.

**Remark 19.** *This is intuitively similar to dimensional analysis, which is a powerful technique used to understand the relationships between different physical quantities by analyzing their dimensions and units. For example, when calculating the resonant period of a simple pendulum with mass $m$, pendulum length $l$ and gravitational acceleration $g$, by analyzing the units of target quantity $[T_{pendulum}] = T^1 = [m]^\alpha [l]^\beta [g]^\gamma$ ([·] denotes its dimension) along with variables $[m] = M^1$, $[l] = L^1$, $[g] = L^1 T^{-2}$. (Here $L$ is length, $T$ is time, $M$ is mass. ) Then by solving the coefficients, $\alpha = 0$, $\beta = -1/2$, $\gamma = 1/2$, we have $T_{pendulum} \propto \sqrt{l/g}$.*

*In our problem setting, if we view the dimension of the largest singular value of $\Sigma$ to be a unit (conditional number is dimensionless), then $[\mathcal{L}_{\mathrm{ori}}] = [\frac{1}{2} \| \Sigma - \prod_{j=N}^{1} W_j \|_F^2] = [\sigma_1(\Sigma)]^2 \left[ \frac{1}{2} \left\| (\sigma_1^{-1}(\Sigma)\Sigma) - \prod_{j=N}^{1} \left( \sigma_1^{-1/N}(\Sigma) W_j \right) \right\|_F^2 \right] = [\sigma_1(\Sigma)]^2$, so $\sigma_1^{-1/N}(\Sigma) W_j$ is dimensionless, $W_j$ has dimension $[\sigma_1(\Sigma)]^{1/N}$, then the initialization scale $\epsilon \propto \sigma_1^{1/N}(\Sigma)$. For the training time, $\frac{\mathrm{d}}{\mathrm{d}t} W_j = \left( \prod_{k=N}^{j+1} W_k \right) (\Sigma - W) \left( \prod_{k=j-1}^{1} W_k \right)$, then $[\frac{\mathrm{d}}{\mathrm{d}t}] = [\sigma_1^{2(N-1)/N}(\Sigma)]$, the training time is proportional to $\sigma_1^{-2(N-1)/N}(\Sigma)$.*

## K  NUMERICAL SIMULATIONS

Through out this section, we consider numerical simulations under four-layer matrix factorization on square matrices with dimension of 5.

### K.1  SADDLE AVOIDANCE DYNAMICS UNDER BALANCE INITIALIZATION

This section presents numerical simulations of the saddle avoidance stage under balanced initialization. In this experiment, $\epsilon = 0.05$, $\eta = 0.1$, $\Sigma_w(0) = \epsilon \cdot \mathrm{diag}(1, 0.8, 0.6, 0.5, 0.9)$.

We set the target matrix to $\Sigma = I$ in Figure 1 and to $\Sigma = \mathrm{diag}(2.00, 1.55, 1.10, 0.65, 0.20)$ in Figure 2. Each pair of solid and dashed lines of the same color represents the logarithms of the $k^{th}$ singular value of $\Sigma_W$ and that of $\frac{1}{2}(U+V)\Sigma_W$, respectively. (Here $U, V, \Sigma_w$ are defined by SVD of product matrix $W$: $W = U\Sigma_w^N V^\top$, or $\cdot^H$ for complex domain. ) Considering the numerical precision and for appropriate visualization, all values plotted are truncated at a small value. (Here the singular values are truncated at $1e - 5$ so the logarithms are truncated at around $-11.5$. )

These figures clearly exhibit the following properties:

- $\sigma_k \left( \frac{1}{2}(U+V)\Sigma_W \right)$ provides a tight lower bound for $\sigma_k (\Sigma_W)$, verifying the conclusion of Lemma 18.
- The spectral gap of the target matrix introduces non-smoothness and non-monotonicity into the original lower bound for singular values of the product matrix, leading to segmented

rather than global smoothness and monotonicity. This explains why the dynamics are easier to analyze when the target matrix is the identity.

- The $1/2$ failure probability of converging to a saddle point under real balanced initialization is a general phenomenon, even if the target matrix is not identity. This illustrates that the exact balancedness in real domain may hinder the convergence in matrix factorization, which is also discussed in Xiong et al. (2024). For the complex initialization, such $1/2$ failure probability of convergence does not occur. This indicates that the complex domain *does not suffer from the drawbacks of exact balancedness* at least under our framework, and thus merits further theoretical investigation.

  It is also interesting to notice that in the setting of Figure 2, initializations with $\det(U^\top V) = 1$ fail to converge but $\det(U^\top V) = -1$ converges, which contrasts with the identity target case (but still with a $1/2$ probability).

- The incremental learning of singular values. Through Figure 1 and Figure 2, we observe the incremental learning of singular values: the model learns features (here the singular values of target matrix) one by one. While we cannot explain why the larger singular values of target matrix converges at first then the smaller ones in Figure 2, and the proof of incremental learning itself is beyond the scope of this work, we still provide an explanation of Figure 1 under the scheme of balanced Gaussian initialization, gradient flow.

  Equation (11) provides both upper and lower bound for the $k^{th}$ singular value of product matrix $\sigma_k(W) = \sigma_k^4(\Sigma_w)$ by the term $\sigma_k((U + V)\Sigma_w)$, while Theorem 5 demonstrates that the increasing rate of this term is accurately bounded and *approximately independent of other components* $k' \neq k$. By invoking conclusions in random matrix theory, we may prove the gap of singular values at initialization, which leads to the explanation of incremental learning. This method can be applied to general random initialization under gradient flow. For gradient descent, more perturbation techniques are required.

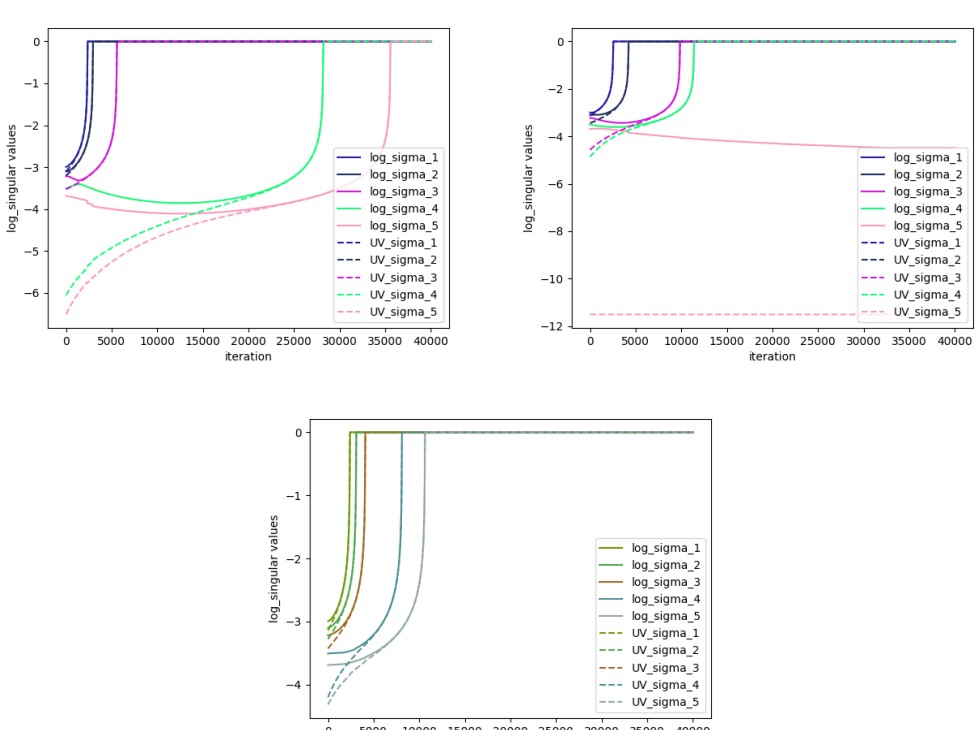

Figure 1: Dynamics of singular values (log scale) for an identity target matrix. From left to right, up to down: real initialization with $\det(U^\top V) = 1$, $\det(U^\top V) = -1$, and complex initialization.

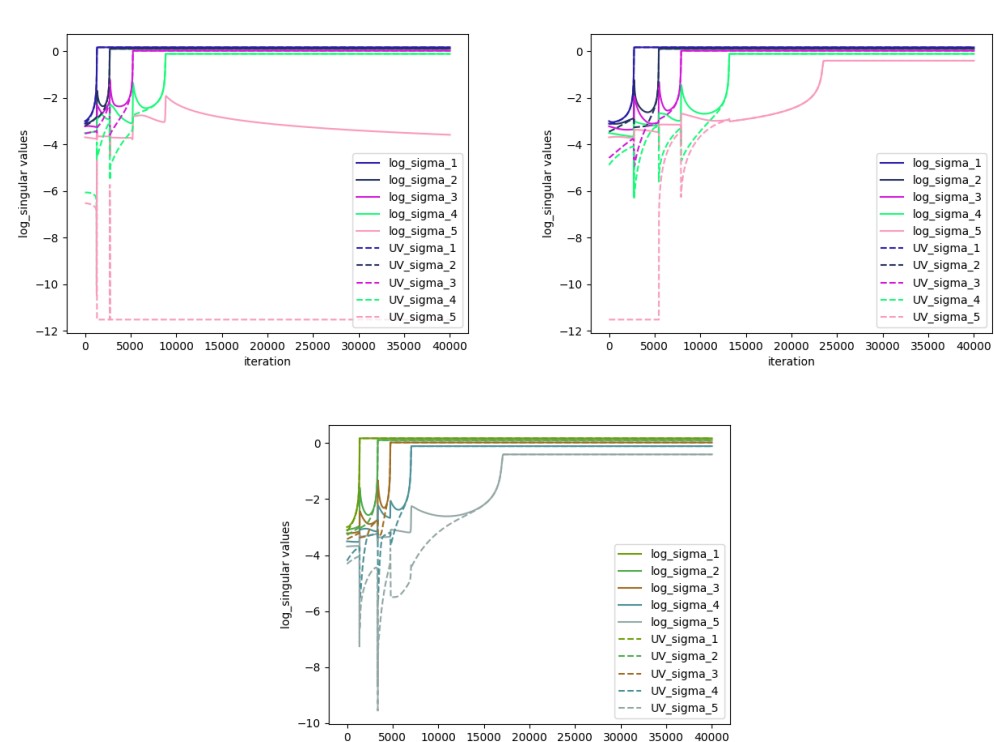

Figure 2: Dynamics of singular values (log scale) for a non-identity target matrix. From left to right, up to down: real initialization with $\det(U^\top V) = 1$, $\det(U^\top V) = -1$, and complex initialization.

## K.2 CONVERGENCE RATE OF DIFFERENT DEPTHS

This section presents examples showing the convergence rate of different depths. Specifically, we vary the depth from 2 to 6 under complex balanced Gaussian initialization, with other hyper-parameters fixed as $\epsilon = 0.05$, $\eta = 0.1$, $\Sigma_w(0) = \epsilon \cdot \mathrm{diag}(1, 0.8, 0.6, 0.5, 0.9)$, $\Sigma = I$. The plots of loss curves and singular values (with dashed line lower bounds which is the same in K.1) are presented in Figure 3.

From the experimental results we exhibit that:

- Generally, deeper $N$ takes more iterations to converge.

- For deeper $N$ the network stays at saddle for more time relative to local convergence phase, which is shown by the sharper change in the decrease of loss and the increase of singular values.

- For depth $N \geq 5$ the lower bound term $\sigma_k((U + V)\Sigma_w)$ still suffers from sudden change when one singular value converges. Furthermore, the monotonicity of this term may not hold anymore, see Figure 4 for result on real domain.

## K.3 ALIGNMENT DYNAMICS UNDER BALANCE REGULARIZATION TERM

This section exhibits the dynamics of weight matrices under regularization term. The original square loss $\mathcal{L}_{\mathrm{ori}}$ is omitted. Here $a = 1$, $\epsilon = 1$, $\eta = 0.001$.

Figure 5 illustrates the conclusion of Theorem 28 and 30. Clearly the maximum among all the singular values are non-increasing while the minimum is non-decreasing.

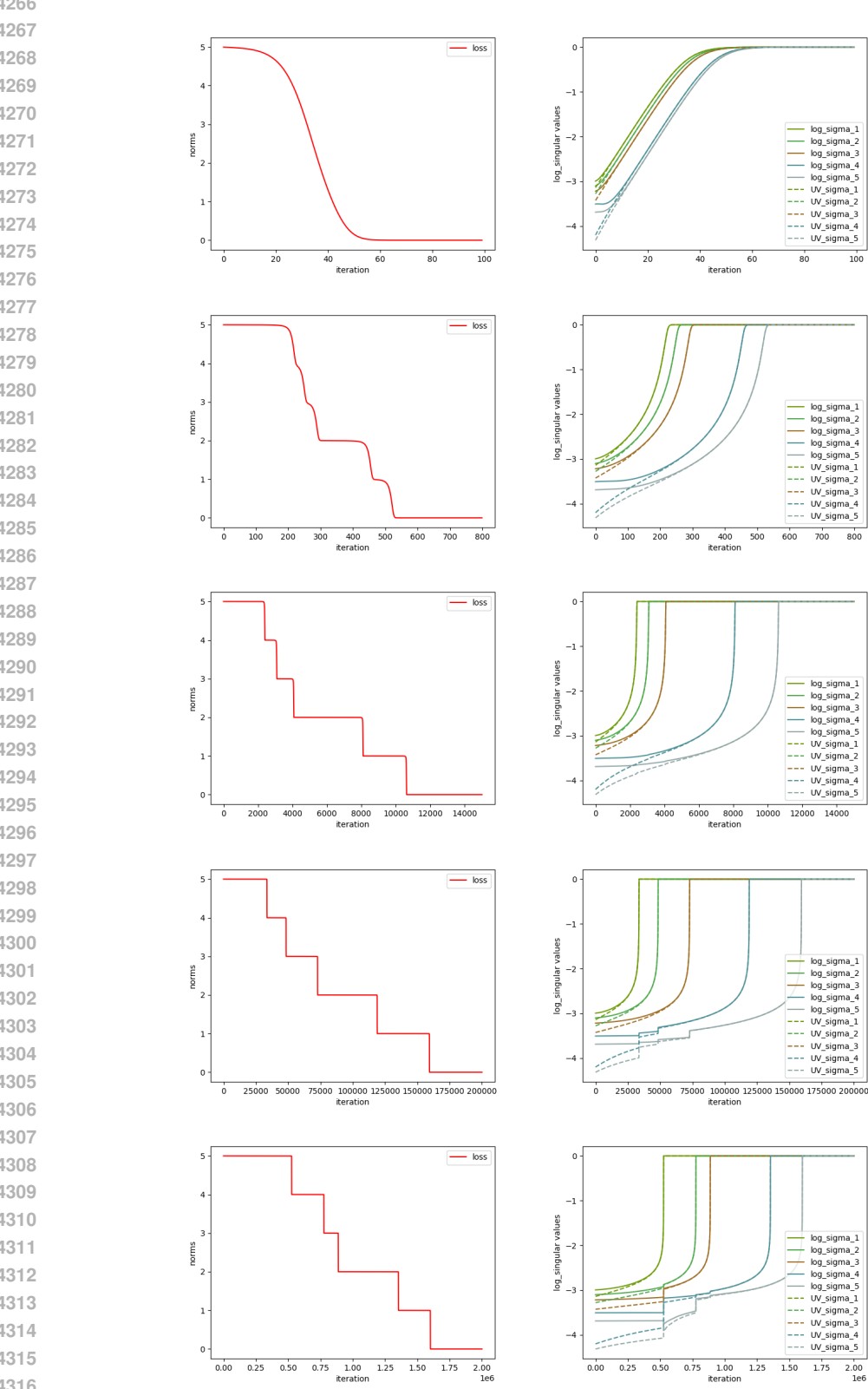

Figure 3: Dynamics of losses and log scale singular values for identity target matrix, under complex initialization, with depth from 2 to 6. Figures on the left are loss curves, the right ones are logarithms of singular values.

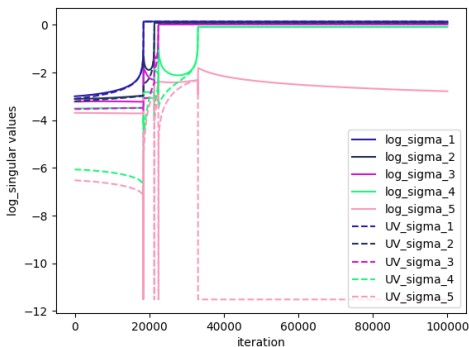

Figure 4: Dynamics of singular values (log scale) for identity target matrix, under real initialization, depth 5, $\det(U^\top V) = 1$.

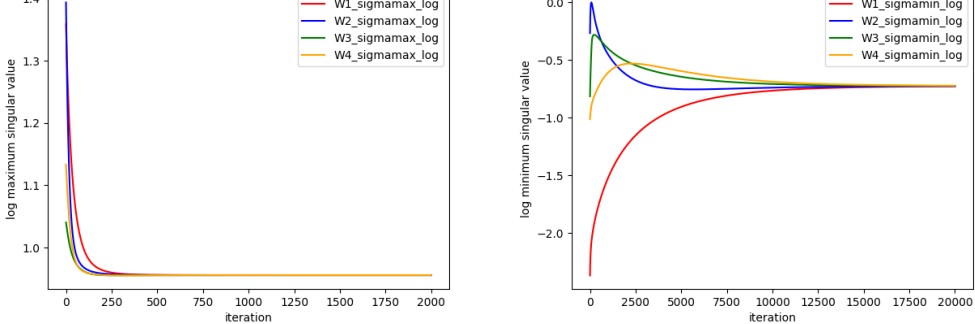

Figure 5: Dynamics of extreme singular values (log scale) for four weight matrices.

Figure 6 illustrates the dynamics of main term $\sigma_{\min}(W_1 + W_2^{-1}W_3^H W_4^H)$. For real initialization with $\det(W(0)) < 0$, $\sigma_{\min}(W_1 + W_2^{-1}W_3^H W_4^H)$ decays to 0 at a linear rate, while for $\det(W(0)) > 0$ and complex initialization it stays at a small value after some oscillation.

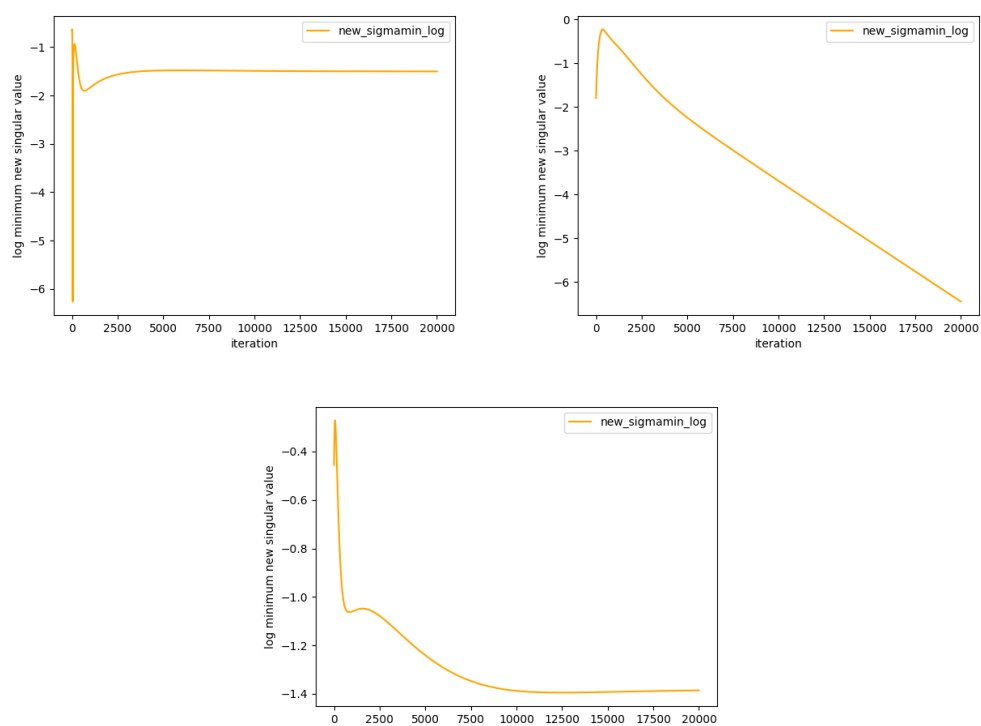

Figure 6: Dynamics of the minimum singular value of Hermitian main term $W_1 + W_2^{-1}W_3^H W_4^H$ (log scale). From left to right, up to down: real initialization with $\det(W) > 0$, $\det(W) < 0$, and complex initialization.

## L    LLM USAGE DECLARATION

In the preparation of this paper, large language models (LLMs) served only as an auxiliary tool for enhancing writing clarity, checking grammar, and assisting in the drafting and debugging of simulation code. These tasks were performed under the authors' complete oversight. The central scientific ideas, theoretical results, and research contributions are entirely the work of the authors.

