# OpenReview forum: "Global Convergence of Four-Layer Matrix Factorization under Random Initialization"
_ICLR.cc/2026/Conference — Submitted to ICLR 2026_

### Official Review · Reviewer_TMKW · 2025-10-25

**Soundness:** 3
**Presentation:** 3
**Contribution:** 3
**Rating:** 6
**Confidence:** 2

**Summary:**

This paper studies the 4-layer matrix factorization problem, with regularization, under gradient flow and infinitesimal learning rate gradient descent. It proves the convergence with probability around 1/2 to the global minimum and around 1/2 to saddles, under a special balanced initialization.

**Strengths:**

The theory seems technically rigorous and the analysis of deeper networks is important even just linear. It extends the results of matrix factorization to deeper layers, although under some strong requirements (see Weaknesses).

**Weaknesses:**

The analysis relies on a special initialization that seems to reduce the four-layer model to the one with approximately two-layer freedom. It also depends on strong regularization. The writing is a little technicle and lacks high-level intuitive explanations of the terms used in the analysis.

**Questions:**

- What is $c_1,c_2$?
- What are the requirements on the width $d$?
- The regularization coefficient is very large (polynomial in width). Could the authors comment on which parts of the analysis would fail if this coefficient were reduced or even set to zero? Is the exact balance between weights essential to the proof?
- The initialization seems very special in at least the following two senses. 1) It could with probability at least 1/2 make the dynamics fall into a degenerate submanifold such that GD/GF will converge to saddles, whose stable manifold is a measure-zero set, i.e., very rare case. 2) Adjacent weights share the same randomness up to canonicalization, and all weights share the same Gaussian randomness. This seems to reduce the model to a two-layer-like case. What, then, is special about the 4-layer setting under this initialization? Which parts of the analysis would fail under a more general (e.g., Gaussian for all weights) initialization?
- Reduce to diagonal target: based on my understanding, the dynamics is nonlinear and the transformation is not symmetric on all the four factors.  Also, it seems the analysis relies on the alignment between $U$ and $V$ as stated around line 280-290.  Please show this reduction does not affect the GD trajectory, beyond the isotropic matrix case.
- Please explain the origin or intuition behind the term $W+(WW^H)^{1/2}$ in the analysis
- Please explain why terms like $(U\pm V)\Sigma$ is useful in the convergence analysis instead of something like $U\Sigma\pm \Sigma V^\top$
- The analysis seems to rely a lot on even layers. What is the reason of bounding even layers and the thresholds of extending it to odd layers? Is this due to the initialization? Also, does this work for deeper layer? If not, what is the threshold of the extension?

---

> ### Author Response · Authors · 2025-11-21
>
> Thank you for your valuable comment and insightful response! Below we address the weaknesses and your questions respectively.
>
> **Weaknesses**: Thank you for your summary of the weaknesses. For the weaknesses of special initialization and strong regularization, please refer to the responses to Question 4 and 3 respectively. For the weakness of technical writing and lack of high-level intuitive explanations, we have moved supporting bounds such as Lemma 9 and 10 to Appendix and simplified Theorem 11, adding an example along with more explanations in Appendix J for Theorem 1 to explain the convergence rate, and revised the notations thoroughly for better understanding (also see the response to Q1).
>
> **Q1 (Definition of $c_1$, $c_2$)**: Thank you for your important question. We apologize for not specifying $c_1$ and $c_2$ beforehand. In the original submission, these quantities are defined in Theorem 6 ($c_1(\delta)$, $c_2(\delta)$), acting as upperbounds depending on the failure probability $\delta$. Previous works often treat them as constants ([1]) so we used $c_{(\cdot)}$ to denote them in the original submission.
>
> To specify the dependence on failure probability $\delta$, in the revised version we have changed the notations to $f_1(\delta)$ and $f_2(\delta)$ while specifying $f_1(\delta),f_2(\delta) = O({\rm poly}(1/\delta))$, then replaced ${\rm poly}(c_1,c_2)$ by ${\rm poly}(\delta)$ in Theorem 1 and 2. These modifications solves the notation problem while making the final result more rigorous. We hope this modification can address your concern.
>
> **Q2 (Requirements on the width $d$)**: Thank you for raising this important question. Throughout this paper there is no restiction on the width, so $d$ can be arbitrary positive integers (for $d=1$ it reduces to scalars). We now have explicitly claimed the restriction of $d \in \mathbb{N}^*$ in Section 1.
>
> In this work we simplify the framework into square matrix decomposition, where all the matrices are square with size $d$. A natural extension of this work is the generalization on dimensions $W_{j} \in \mathbb{F}^{d_{j+1}\times d_{j}}$, where we leave this for future work.
>
> **Q3 (Necessity of large regularization coefficient)**: Thank you for this crucial question.
>
> Balanced regularization is widely discussed in previous works (e.g. [2] [3] [4]), where the regularization generally is polynomial in matrix width $d$. We believe that this requirement of regularization coefficient is relatively reasonable. Generally we **do not require exact balancedness** (treated as a special case in Section 4), our main result is under a general random Gaussian initialization setting in Section 5 (requiring balanced regularization). The poly-size regularization coefficient ensures approximate balancedness.
>
> If this coefficient is reduced or set to $0$, the balanced error is not relatively small compared to the smallest singular values of the weight matrices, then: the Skew-Hermian error term and Hermitian main term may not share the same dynamics as discussed under the exact balancedness due to the relatively large balanced error, rasing technical challenges for stage 2 (saddle avoidance stage).
>
> **Q4 (Specialness of the initialization)**: Thank you for pointing out this important question. Our global convergence result is conducted under random Gaussian initialization, with a balanced regularization term. In Section 4 we also provide a result for balanced Gaussian initialization, mainly to show the method of convergence proof in a relatively simple setting.
>
> This balanced Gaussian initialization itself is the combination of random Gaussian initialization and exact balancedness (Corollary 13 in the Appendix). It acts as a prelude for the main result (Theorem 1). We have also included the intuition of special balanced (Gaussian) initialization at the beginning of Section 4.
>
> Generally, for $N$-layer matrix decomposition under any balanced initialization, their dynamics of the product matrix $W:= W_N \cdots W_2 W_1$ reduce to $\frac{\mathrm{d}}{\mathrm{d}t} W = f(W, W^H)$ (independent of layer weights $W_{1,2,\cdots, N}$), which simplifies the analysis (refer to [5]) but **does not reduces to two-layer like dynamic**. Initialization itself is not special for $N=4$, the main challenge towards general $N$ and $\Sigma$ is Theorem 5 (see the response to Q8).
>
> For the 1/2 of failure case (in real domain) under balanced initialization, it is due to the exact balancedness which causes the $\sigma_{\min}((U+V)\Sigma_w)$ to be zero with $1/2$ probability. For random Gaussian initialization, it is hard to characterize the dynamics of this $1/2$ case. Complex domain does not suffer from this $1/2$ failure probability.

---

> ### Author Response · Authors · 2025-11-21
>
> **Q5 (Reduction to diagonal target)**: Thank you for pointing out this interesting question. In our **Appendix A in the original submission (now Appendix B)**, we discussed that for general target matrix the GD trajectory can be reduced to the diagonal target case by applying the following transformation: if the SVD of the target matrix is $\Sigma = U_\Sigma \Sigma^\prime V_\Sigma^H$ by applying $W_N^\prime = U_\Sigma^H W_N$, $W_1^\prime = W_1 V_\Sigma$, then the whole training dynamics is the same as the dynamics of target matrix $\Sigma$ (diagonal).
>
> We also claimed that this reduction does not affect initialization. Please refer to Appendix B for detailed derivations.
>
> For line 280-290 in the original submission (Theorem 4), we discussed the formation of this theorem under general target matrix (or equivalently, before the diagonal reduction). Specifically, this theorem becomes $\frac{\mathrm{d}}{\mathrm{d} t} ||{\Sigma^\prime}^{1/2} (U_\Sigma^H U - V_\Sigma^H V) \Sigma_w ||_F^2 \le 0$, which is also related to the SVD of target matrix. We now have added reference for the Remark on Theorem 4 (around line 280-290 in the original submission).
>
> **Q6 (Intuition behind $W+(WW^H)^{1/2}$)**: Thank you for raising this crucial question. This term arises from the technical bridge required to generalize our analysis from the strictly balanced setting (Section 4) to the imbalanced random initialization case (Section 5).
>
> Briefly, we want to lower bound $\sigma_{\min}(W_1 + W_2^{-1} W_3^H W_4^H) = \sigma_{\min}((W_4^{-1}W_3^{-1}W_2^{-1})(W + W_4 W_3 W_3^H W_4^H))$ at the end of alignment stage $T_1$ as the transition to the next saddle avoidance stage. From $W(T_1)\approx W(0)$, $W_4 W_3 W_3^H W_4^H \approx (WW^H)^{1/2}$ at $T_1$, and $W_{2,3,4}^{-1}$ are bounded at $T_1$, we only need to lower bound $\sigma_{\min}(W(0)+(W(0)W(0)^H)^{1/2})$. We have added detailed claims in Section 5.2.2 for your reference.
>
> **Q7 (Why $(U+V)\Sigma_w$ is useful)**: Thank you for pointing out this crucial question. This is a key quantity in the exact balanced case and gradient flow (Section 4) rather than $U\Sigma_w + \Sigma_w V$ because:
>
> (1) It can be easily expressed by weight matrices. Under balanced initialization, to obtain $U,V,\Sigma_w$ we only have $W_N W_N^H = U\Sigma_w^2 U^H$, $W_1^H W_1 = V \Sigma_w^2 V^H$, $W = U \Sigma_w^N V^H$. We cannot directly obtain $U\Sigma_w + \Sigma_w V$, but for $2\mid N$. However $U\Sigma_w^2 V^H = (W_N W_N^H)^{-(N-2)/2} W$ (if $2\mid N$ we can take time derivative), so we have $(U+V)\Sigma_w^2(U+V)^H$. Section E in the Appendix provides more details for your reference.
>
> (2) $(U+V)\Sigma_w^2(U+V)^H$ is a key term in the proof of Theorem 5, which is essential in the saddle avoidance stage since the increasing rate of $\sigma_{\min}((U+V)\Sigma_w)$ is lower bounded.
>
> **Q8 (Extension to odd layers or deeper layers)**: Thank you for pointing out this insightful question. Below we answer your questions under the framework of the exact balanced case and gradient flow (in general case we only need to treat imbalancedness as error terms).
>
> (1) Extension to odd layers: Our convergence analysis requires the time derivatives of $||\Sigma^{1/2}(U-V)\Sigma_w||\_F$ (Theorem 4) and $\lambda_{min}((U+V)\Sigma_w^2 (U+V)^H)$ (Theorem 5). As explained in the response to Q7 part (1), if $2\nmid N$, their time derivatives cannot be expressed by the matrices directly.
>
> Although we proved Theorem 4 under odd $N$ by adopting techniques in [5], it is hard to generalize since this approach highly depend on the exact balancedness. We addressed this in the proof sketch of Theorem 4.
>
> (2) The main obstacle to general even layers is the Hermitian main term in Theorem 5. For $N \ge 6$, the summation term of equation (147) in the proof of Theorem 5 contains more terms, making the error analysis hard to establish and such term may not have nice properties as $N=4$. We have conducted numerical experiments in Appendix K.1 showing the term in Theorem 5 suffers from bizzare sudden change for your reference.
>
> Reference:
>
> [1] Tian Ye and Simon S.Du. Global convergence of gradient descent for asymmetric low-rank matrix factorization, 2021 https://arxiv.org/abs/2106.14289.
>
> [2] Dohyung Park, Anastasios Kyrillidis, Constantine Carmanis, and Sujay Sanghavi. Non-square matrix sensing without spurious local minima via the burer-monteiro approach. In Artificial Intelligence and Statistics, pp. 65–74. PMLR, 2017.
>
> [3] Rong Ge, Chi Jin, and Yi Zheng. No spurious local minima in nonconvex low rank problems: A unified geometric analysis, 2017. https://arxiv.org/abs/1704.00708.
>
> [4] Qinqing Zheng and John Lafferty. Convergence analysis for rectangular matrix completion using burer-monteiro factorization and gradient descent. arXiv preprint arXiv:1605.07051, 2016.
>
> [5] Sanjeev Arora, Nadav Cohen, Wei Hu, and Yuping Luo. Implicit regularization in deep matrix factorization, 2019b.https://arxiv.org/abs/1905.13655.

---

> > ### Comment · Reviewer_TMKW · 2025-11-26
> >
> > I would like to thank the authors for the detailed reply. My questions have been clarified. Based on my understanding, the analysis still relies on very special settings that depend on initialization, near-balancedness, and unifying all weight matrices; nevertheless, the proof itself is technically meaningful, and the results serve as a decent extension to current literature. I will keep my score.

---

### Official Review · Reviewer_QmG7 · 2025-10-29

**Soundness:** 2
**Presentation:** 1
**Contribution:** 2
**Rating:** 2
**Confidence:** 3

**Summary:**

In this paper, the authors study convergence of full-batch gradient descent (GD) on 4-layer matrix factorization with random initialization, i.e., GD on the loss

$\mathcal L (W_1,W_2,W_3,W_4) = 1/2 \Vert W_4W_3W_2W_1 - \Sigma \Vert_F^2 +  \mathcal{L}_{reg} (W_1,W_2,W_3,W_4)$

To prove convergence in this setting they let $\mathcal{L}_{reg}$ be a balancedness-regularization.

Their main result, Theorem 1, shows convergence of GD to a global minimum in polynomial time (depending on the ambient dimension and the operator norm of $\Sigma$).

**Strengths:**

+ Proving global convergence of GD in the general $L$-layer case is an interesting problem

**Weaknesses:**

- Quality of the writing makes it really hard to parse statements
- Quantities are used before they are defined
- Relevant literature is not taken into account
- 4-layer case with balancedness regularization is a very special setting and main theorem only applies to $\Sigma$ with flat singular spectrum, i.e., all singular values are identical

**Questions:**

I do not recommend this paper to be accepted. There are several reasons for this decision:

1.) Due to the noticeable low quality of writing, it is very difficult to fully parse mathematical statements and to check their correctness. Quantities are often used before they are defined, e.g., $\Delta_{j,j+1}$ used in line 161, but defined in line 187, or  balanced Gaussian initialization used in line 207, but defined in lines 221ff

2.) Relevant literature such as

Nguegnang, G.M., Rauhut, H. & Terstiege, U. Convergence of gradient descent for learning linear neural networks. Adv Cont Discr Mod 2024, 23 (2024). https://doi.org/10.1186/s13662-023-03797-x

has not been taken into account. This makes it hard to assess the novelty of the presented results.

3.) The main result only applies to a very restricted special case (4-layer, identical singular values of $\Sigma$, $\mathcal L$ including balancedness regularization).

Since the combination of these points requires substantial changes, I strongly recommend a thorough revision of the full manuscript and extension of the results before resubmission.

---

> ### Author Response · Authors · 2025-11-21
>
> We address your questions and concerns below. Thank you for pointing out the writing issue. We have undertaken a thorough revision of the manuscript to enhance clarity and rigor. We hope that these will clarify our contribution and lead to a more favorable reassessment.
>
> **W1 (Quality of the writing makes it really hard to parse statements)**:
>
> We apologize for the difficulties in parsing the statements. We have revised the manuscript thoroughly to improve clarity and organization.
>
> Besides refining the expressions (for example we have changed "balance error" to "balance difference" since imbalance can some time be benificial to convergence, see [1]), we also have:
>
> (1) Refined Explanations: We have modified the remarks and explanations for Theorem 1 and 2, to better connect the technical results with the overall training dynamics.
>
> (2) Notation Consistency: We have changed notations to avoid overload and ambiguity throughout the paper and the Appendix.
>
> (3) Accurate Framing: We have rephrased the abstract and introduction to describe our results and their scope in a more accurate way.
>
> (4) Improved Navigation: We have added Paper Roadmap sections at the end of the Introduction Section and in the Appendix for guiding the readers. This section specifies the assumptions made for main theorems.
>
> We hope these modifications can address your concerns.
>
> **W2&Q1 (Quantities are used before they are defined)**: We apologize for the confusion caused by the ordering of definitions. We have reorganized the manuscript to ensure all mathematical terms (*e.g.*, $\Delta_{j,j+1}, c_1, c_2$, definition of **balanced Gaussian initialization**, etc.) are defined prior to their use. We also add new Paper Organization parts for easy checking of definitions and the dependencies of mathematical statements.
>
> **W3&Q2 (Relevant literature is not taken into account)**: Thank you for bringing the relevant work [2] to our attention. The literature on Linear Networks is vast, and we apologize for missing this specific result. We have now added a discussion of this paper in Section 2.
>
> While [2] focus on general deep linear networks, they only analyze the convergence to a critical point for $N \ge 3$ under certain conditions. Their result does not provide a global convergence guarantee to a minimum in polynomial time under random initialization, which is the core contribution of our work.

---

> ### Author Response · Authors · 2025-11-21
>
> **W4&Q3 (The main result only applies to a very restricted special case)**: We demonstate why our setting is standard and important as below.
>
> (1) Balanced Regularization: Balancedness regularization term is a standard setting widely used in previous works (see [3], [4], [5]).
>
> (2) 4-Layer Restriction: As emphasized in Section 1, our paper presents the first global convergence guarantee that goes beyond $N=2$ layers under random Gaussian initialization beyond NTK regime. The $N=4$ case is the simplest deep architecture that allowed us to establish the necessary conditions for convergence.
>
> However, for the *non-increasing property of Skew-Hermitian error term* in Theorem 4, we have established this result for *general depth $N$ and both complex and real domain* under balanced initialization and gradient flow (in the revision version we have completed the proof for odd $N$ and complex domain, which requires  additional mathematical techniques). We believe this result can be extended to general even $N$ under random initialization and provide general insights to future analysis.
>
> As listed in the paper roadmap section, some of other results can also be applied to general $N$, including the properties of initializations, the impact of large regularization terms, etc.
>
> (3) Identical Singular Values: We demonstrate why this assumption is needed with simulations in Appendix K.1. Empirical results suggest that the monotonicity of the lower bound (the Hermitian main term in Theorem 5) does not hold anymore for general target matrices.
>
> We have now included a more detailed discussion of this challenge after Theorem 5 and in Appendix K.1.
>
> We would also like to emphasize that global convergence for deep matrix factorization is a crucial open problem significantly harder than shallow (two-layer) factorization. Previous works either did not establish a global convergence guarantee or require restricted initialization conditions. In our work, we have discovered an intrinsic non-increasing quantity (Theorem 4) and an approach to prove global convergence for $N=4$ with identical target matrices (Theorem 5), in both real and complex domains. While our result does not yet address the fully general case, we believe this is the important first step towards understanding the dynamics of general deep matrix factorization and will inspire future work to relax these restrictions.
>
>
> References:
>
> [1] N. Xiong, L. Ding, and S. S. Du, "How Over-Parameterization Slows Down Gradient Descent in Matrix Sensing: The Curses of Symmetry and Initialization", arXiv preprint arXiv:2310.01769, 2023.
>
> [2] Nguegnang, G.M., Rauhut, H. & Terstiege, U. Convergence of gradient descent for learning linear neural networks. Adv Cont Discr Mod 2024, 23 (2024). https://doi.org/10.1186/s13662-023-03797-x
>
> [3] Dohyung Park, Anastasios Kyrillidis, Constantine Carmanis, and Sujay Sanghavi. Non-square matrix sensing without spurious local minima via the burer-monteiro approach. In Artificial Intelligence and Statistics, pp. 65–74. PMLR, 2017.
>
> [4] Rong Ge, Chi Jin, and Yi Zheng. No spurious local minima in nonconvex low rank problems: A unified geometric analysis, 2017. https://arxiv.org/abs/1704.00708.
>
> [5] Qinqing Zheng and John Lafferty. Convergence analysis for rectangular matrix completion using burer-monteiro factorization and gradient descent. arXiv preprint arXiv:1605.07051, 2016.

---

> > ### Comment · Reviewer_QmG7 · 2025-11-24
> >
> > First of all, I thank the authors for their efforts to clarify the presentation. I furthermore acknowledge that it is hard to keep track of all relevant literature in such a fast evolving field and that the problem you investigate is inherently challenging.
> >
> > However, I still see severe issues in presentation and scope of the results.
> >
> > Regarding presentation:
> >
> > After first reading a theoretical paper, I expect to have learned some insights into the problem even without fully going through the proof details. The present paper leaves me behind more confused than before.
> >
> > 1.) Structure: At the moment, Sections 1-3 introduce the problem setting and provide an informal version of the main result (Theorem 1), Section 4 tries to explain the proof strategy for the simpler case of balanced Gaussian initialization, and Section 5 then tries to extend this to the general case. For me, this overall structure clearly misses its point by mixing high-level convergence results on the same level as Theorem 1 with technical lemmata bounding specific quantities used during the proof (e.g., Theorem 3).
> >
> > 2.) Level of detail: The results in Sections 4 and 5 are presented in very different level of detail (compare for instance Theorems 6 and 7)
> >
> > 3.) Formal statements: The results are often stated in an unclear/imprecise way. For instance, what shall the second sentence in Point 2 of Theorem 3 even mean? I simply cannot parse it grammatically.
> >
> > Regarding the identical singular values:
> >
> > 1.) This restriction means that the paper only treats target matrices which are scalar multiples of unitary matrices, right? This should be made clear from the beginning when the problem set-up is introduced.
> >
> > 2.) You wrote:
> >
> > „(3) Identical Singular Values: We demonstrate why this assumption is needed with simulations in Appendix K.1. Empirical results suggest that the monotonicity of the lower bound (the Hermitian main term in Theorem 5) does not hold anymore for general target matrices.“
> >
> > Do I understand correctly that this assumption is only needed by your specific proof technique? If yes, this suggests to me that the current proof technique might not be the right way to approach the more general problem and as thus also puts into question the use of the findings.
> >
> >
> > In the above explanation I only mentioned single examples, but I find these problems all over the (main) document. Hence, I see no way of resolving these issues by small cosmetic changes, but only by a substantial re-writing of the paper.
> >
> > Despite the fact that my evaluation deviates from the remaining reviewers, I thus keep my score and trust in the AC to objectively decide whether my comments are relevant for the final decision.

---

> > > ### Author Response · Authors · 2025-11-29
> > >
> > > Thank you for your thorough reading and further feedback. For the point 2 of Theorem 3, we have corrected a typo which may led to misunderstanding and have clarified the statement. We believe that level of detail and paper structure are the choice of writing to represent our results. We would also like to clarify that our paper first established a global convergence result for matrix factorization of depth over 2 beyond NTK regime, along with intermediate results for general depth, although the main result is presented under a specialized setting on target matrix and regularization.

---

### Official Review · Reviewer_dzdx · 2025-10-31

**Soundness:** 3
**Presentation:** 2
**Contribution:** 4
**Rating:** 6
**Confidence:** 4

**Summary:**

This paper provides a rigorous polynomial-time global convergence analysis for gradient descent on a four-layer linear network (matrix factorization), given the inclusion of a standard balanced regularization term and the condition that the target matrix has identical singular values. Previous results addressed only two-layer networks or required balanced initialization; this work achieves the first provable global convergence guarantee for a deeper architecture (N=4) under small random Gaussian initialization. The proof proceeds through a novel three-stage decomposition: alignment, saddle avoidance, and local convergence. The authors introduce two monotonic quantities (a non-increasing skew-Hermitian error and a non-decreasing Hermitian main term) that track training progress and ensure singular values remain bounded away from saddle points. The result holds with high probability in the complex setting and with probability close to 1/2 in the real setting.

**Strengths:**

1. Strong theoretical novelty: This work establishes the first polynomial-time global convergence guarantee for gradient descent on a deep matrix factorization problem (N>2) under random initialization, specifically targeting the four-layer (N=4) architecture. This result addresses a longstanding open question regarding general deep matrix factorization and Deep Linear Networks. The paper is technically impressive and represents a meaningful step forward in understanding deep gradient dynamics.

2. High mathematical rigor and completeness: The paper exhibits high mathematical rigor, supported by detailed proofs that are presented in full in the Appendix, allowing a reader to verify each step independently. The analysis requires complex technical developments, including the utilization of tools from random matrix theory, such as the Circular Ensembles, for analyzing the minimum singular value at initialization. Furthermore, to rigorously translate continuous-time intuition (Gradient Flow) into guarantees for the discrete Gradient Descent algorithm, the authors developed new perturbation bounds for eigenvalues. These auxiliary lemmas and novel technical approaches are considered to be of independent interest for deep linear networks analysis.

3. Conceptually structured framework: The three-phase view of training provides an intuitive narrative for how deep networks self-align. The construction of the skew-Hermitian and Hermitian terms offers a clean way to formalize this intuition, and could serve as a basis for analyzing even deeper or nonlinear architectures.

**Weaknesses:**

1. Limited scope and overstated generality: The analysis applies specifically to the four-layer case with identical singular values of the target matrix. The extension to general depth or non-isotropic targets is only conjectural. The paper’s framing (“global convergence for deep networks”) slightly overstates the reach of the results.

2. Dense presentation and navigational difficulty: Despite occasional intuitive remarks, the exposition remains heavy, with long theorem sequences spanning most of the paper. Assumptions vary subtly across theorems, and the logical dependencies are hard to track. A table summarizing the different assumptions made in each theorem and their dependencies would help improve readability. I also suggest moving some of the technical results (especially those that serve primarily as supporting bounds or spectral perturbation lemmas) to the appendix would free space for a higher-level discussion of the main theorems and their implications. This would help readers grasp the overarching logic without losing rigor and would also make room for intuitive commentary or schematic illustrations of the convergence stages.

3. Notation overload: The paper suffers from significant notation overload and inconsistency, which poses a challenge to readability despite the rigorous content. For instance, $\delta$ serves both as a failure probability parameter and as the minimal singular value in theorem 7. M serves as a scalar bound for maximum singular values and again as a placeholder for a matrix sum (M=S+D) in perturbation lemmas in the appendix.

**Questions:**

1. The discrete-time bounds include a higher-order term $a^2 \eta^2$ that appears to control the deviation between gradient flow and gradient descent. However, $\eta$ is defined as a function of $a$. Could the authors clarify the intended scaling between the initialization magnitude $a$ and step size $\eta$?

2. Some results (e.g., the main convergence theorem) explicitly assume $N=4$, while other constants and inequalities treat $N$ as variable. Could the authors clarify which theorems hold only for the four-layer case and which, if any, extend to general even depth? A brief discussion of whether the proof technique could plausibly generalize beyond N=4 would be very helpful.

3. $\delta$ denotes both a failure probability parameter and the minimal singular value in theorem 7. Could the authors clarify whether these are distinct quantities and consider using distinct notation or a notation summary to reduce ambiguity?

4. The paper establishes a deep linear convergence result under strong structural assumptions. Could the authors elaborate briefly on what aspects of the proof they believe might carry over to nonlinear or more general architectures?

---

> ### Author Response · Authors · 2025-11-21
>
> Thank you for your valuable comment and insightful response! Below we address the weaknesses and your questions respectively.
>
> **W1 (Limited scope and overstated generality)**:
>
> Thanks for pointing out this weakness. As emphasized in the title, our main focus is for four-layer matrix factorization, as special case of general deep matrix factorization (number of layers larger than two). The introduction for deep linear network is to justify the motivation of matrix factorization beyond two layer, and exhibit the connection between matrix factorization and deep network. We have rephrased the abstract and introduction to be more accurate due to your suggestion.
>
> **W2 (Dense presentation and navigational difficulty)**: Thank you for pointing out this weakness. While the assumption of main Theorem is clear, some of the intermediate results can be relaxed to more general settings, which may lead to complex local dependency as you emphasized. We apologize for the dense presentation and navigational difficulty.
>
> To present our proof strategy, we first consider balanced initialization and gradient flow in Section 4, where we first present properties of initialization stated in Theorem 3 then state Theorem 4 (non-increasing Skew-Hermitian error) under arbitrary number of layers $N$, both complex and real domain (in the submission we are missing $2 \nmid N$ for complex domain due to techinical reasons, now we have completed the result); Theorem 5 (non-decreasing Hermitian main term) only holds under $N=4$ and identical target, both real and complex domain are discussed. We then conduct similar results for random Gaussian initialization in Section 5, where Theorem 6 states the properties of initialization, Theorem 7 (convergence rate of the regularization term) only requires $N=4$, Theorem 8 (the change of the maximum and minimum singular values) holds for arbitrary $N$.
>
> For the suggestion of removing extra lemmas, we have removed supporting bounds of Lemma 9 and 10 into the Appendix because: (1) they only act as supporting bounds for the main in the saddle avoidance stage, (2) they are just adaptations of Theorem 4 and 5 for random Gaussian initialization scheme. We have also simplified the conclusions of Theorem 11 (local convergence stage) in the original submission (Now Theorem 9) by removing supporting bounds. The spectral perturbation lemmas are located in Appendix D.2 in the original submission (We have added reference to the corresponding lemmas in the main paper). Theorem 7 and 8 are crucial in demonstrating the effect of regularization term so we have kept them in the main paper.
>
> To We have added a concise paper organization at both the end of introduction section for better navigation, along with a table summarizing the different assumptions made in each theorem. We have also added a brief summary at the beginning of Appendix so that readers can quickly find the corresponding lemmas and theorems or verify the complete proof themselves.
>
> **W3 (Notation overload)**: We apologize for the notation overload. Due to your valuable suggestion, we revise the notation of the main paper and the appendix thoroughly. Although the overload of $M$ and $\delta$ in Theorem 7 only appears distinctively in the discussion of regularization term, and does not occur with the $\delta$ of failure probability and the $M$ in the perturbation lemmas together, we now modify the upper and lower bound for singular values in Theorem 7 as $\mu_{\max}$ and $\mu_{\min}$ in case of notation overload.
>
> For other intermediate quantities which may cause ambiguity, we also replace them with different notations, or add appropriate subscripts to avoid confusion. For example, for the main theorem where $\delta$ also appears as the upper bound for the final loss, we replace $\delta$ with $\epsilon_{\rm conv}$ following conventions while not violating the $\epsilon$ in the scale of small initialization. We also replace the $N$ appears in Appendix C.1 with $K$ to avoid notation overload with number of weight layers $N$, although they do not appear together.

---

> ### Author Response · Authors · 2025-11-21
>
> **Q1 (The dependence of $\eta$ on $a$)** Thank you for your important question. As you emphasized, there is a higher-order term $\eta^2 a^2$ due to the discrete optimization steps compared to continuous gradient flow setting. Furthermore, since the first order term contains a $\eta$, there should be some restrictions on learning rate such as $\eta = O(a^{-2}\cdots)$. However in other derivations $\eta$ have other restrictions which can not be merged together with this one. For simplification, we write $\eta \le {\rm poly}(a,d,\cdots)$, and refer to Theorem 47 for complete restrictions on initialization scale $\epsilon$, regularization coefficient $a$, learning rate $\eta$, along with convergence time $T$ with specified polynomial degrees. We now have explicitly guided the readers to refer to Theorem 47 for complete restrictions on $\eta$ and $a$ after Theorem 1 (and similar in Theorem 2).
>
> **Q2 (Extension to general even depths)** Thank you for your valuable question. As mentioned in the response to Weakness 2, Theorem 4 (Skew-Hermitian term) is established on arbitrary depth $N>2$ under both complex and real domain. Although under imbalanced initialization, this property is hard to characterize for odd depths, for even depths the following construction may helps: $$W_1 + W_{2}^{-1} \cdots W_{N}^{-1} W_{N+1}^H \cdots W_{2N}^H$$
>
> Theorem 5 (Hermitian term) does not hold for general $N$ nor non-indentical target matrix, which is the main obstacle towards global convergence proof for general even depth. We have added more discussions on this Theorem.
>
> For the training dynamics under regularization term (Theorem 7 and 8), they can be generalized to arbitrary $N$. Theorem 8 has already established for general $N$; while Theorem 7 is for $N=4$, we provided a proof for general $N$ under fradient flow (Theorem 27 and 28).
>
> **Q3 (Notation overload of $\delta$)** Thank you for your valuable suggestion. The $\delta$ in Theorem 7 in the original submission are distinctive quantities, and we have replaced the one in Theorem 7 with $\mu_{\min}$ in the revised version, which is also discussed in Weakness 3. Since we eliminate the overload of notations, we believe that the revised version is less ambiguous.
>
> **Q4 (Extension to nonlinear or more general architectures)** Thank you for your insightful question. This is a interesting future work direction. This work mainly focuses on linear neural networks, and since no concrete theoretical results for nonlinear neural networks has been established in our work, we cannot guarantee that which part of our results can be generalized to nonlinear structures. To our knowledge, previous work [1] has proved that the balancedness property holds for general deep networks under gradient flow. We also conjecture that similar property of Theorem 4 may hold for nonlinear networks.
>
> Reference:
>
> [1]  Simon S.Du, Wei Hu, and Jason D.Lee. Algorithmic regularization in learning deep homogeneous models: Layers are automatically balanced, 2018. https://arxiv.org/abs/1806.00900.

---

### Official Review · Reviewer_FFAL · 2025-11-01

**Soundness:** 3
**Presentation:** 3
**Contribution:** 4
**Rating:** 8
**Confidence:** 3

**Summary:**

The paper studies gradient flow under the balancedness condition and gradient descent without the balancedness assumption in a four-layer linear network, proving global convergence under random initialization. The authors consider both complex-valued and real-valued matrices and show that, with high probability under complex initialization and with probability $1/2$ under real initialization, the product matrix converges to a global minimum in polynomial time. The paper provides a proof sketch that proceeds through three distinct phases: first, the layers become balanced; second, the minimum eigenvalue of the Hermitian term remains positive; and finally, the loss decreases monotonically to zero.

**Strengths:**

- The paper is well-organized and, although technically challenging, is written in a way that makes the content as accessible as possible to readers. The problem settings and notations are clearly presented, which helps readers follow the subsequent sections. In addition, Section 4 (Gradient Flow under Balanced Gaussian Initialization) effectively serves as a warm-up, while Section 5 (Gradient Descent under Unbalanced Gaussian Initialization) presents the main results and provides a well-structured proof sketch.
- This paper is the first to provide results on four-layer matrix factorization under random Gaussian initialization beyond the NTK regime, which represents a significant contribution.It presents important and novel findings not only for the analytically tractable (though nontrivial) gradient flow case but also, building on this foundation, for gradient descent under unbalanced initialization, offering a comprehensive analysis of both settings.

**Weaknesses:**

- The paper lacks a clear explanation of the convergence rate derived in Theorems 1 and 2. A more detailed discussion of this rate, including how tight the bound is, would strengthen the analysis. It would also be helpful to compare the convergence rate with that of the depth-2 case (e.g., [1]). In addition, although the authors note that it is difficult to analyze odd factorizations theoretically, it would still be valuable to include an empirical comparison of the convergence behavior across depth-2, depth-3, depth-4, and even deeper networks.
- The paper does not demonstrate the incremental learning of singular values in matrix factorization, where the model gradually learns features one by one. This limitation may stem from the assumption that all target singular values are equal in most theorems. However, Figure 1 (identity target) suggests that, possibly due to random initialization, such incremental learning behavior still emerges. Could the authors provide a theoretical explanation or justification for this phenomenon under their setting?
- The paper does not address cases with different target singular values or rank-deficient targets. While this omission is understandable given the analytical difficulty, it would be helpful if the authors could briefly explain what makes these cases challenging to analyze.

[1] Global Convergence of Gradient Descent for Asymmetric Low-Rank Matrix Factorization

**Questions:**

- Can the analysis be extended to deeper (even) linear networks? If not, please clarify the main obstacles or limitations that prevent such an extension.
- It is also unclear why the complex domain is considered alongside the real domain. Typically, analyses are conducted only in the real domain, so it would be helpful to explain the specific motivation or advantage of including the complex case.
- Could the authors clarify why the convergence rate and initialization scale involve the term $\sigma_1(\Sigma)^{1/4}$? In particular, and how does it compare to the two-layer results?

---

> ### Author Response · Authors · 2025-11-21
>
> Thank you for your valuable comment and insightful response! Below we address the weaknesses and your questions respectively.
>
> **W1 (lack of explanation for convergence rate)**: We apologize for the missing of explanation in Theorem 1 and 2. Generally the tightness of our convergence rate bound is hard to prove, however we can provide a simple one-dimensional toy-model to illustrate that the bound is nearly tight, with a clear intuition behind. We have included this simple example after Theorem 1 in the formal version of this paper.
>
> For the convergence behaviour of different $N$, we have conducted numerical simulations for $N=2,3,4,5,6$ for balanced initialization, updated in Figure 3, Appendix K.2. Generally, for deeper $N$ the network stays at saddle for more time relative to local convergence phase, which is shown by the sharper decrease of loss.
>
> **W2 (lack of demonstration of the incremental learning of singular value)**: Thank you for pointing out this. Incremental learning is indeed a general observation in matrix factorization, which is also observed in our simulation.
>
> Previous works have discussed incremental learning under two layer matrix factorization ([1]). Generally proving incremental learning is more technically challenging than proving global convergence (the latter one is naturally implied by the former one), which is beyond the scope of this work. We will consider this as future work, but we also updated some elementary demonstrations to Appendix K.1.
>
> We also conjecture that for non-identical target matrix (conditional number $\kappa > 1$), characterizing incremental learning is helpful to prove the global convergence, since it is hard to find a universal lower bound with good properties such as (approximate) monotonicity (shown in Figure 2). We will consider this as future work.
>
> **W3 (lack of explanation for the obstacle in generalizing different target singular values)**: Thank you for pointing out this weakness. As shown in Figure 2, the convergence behaviour for different target singular values suffers from a sudden change when one component converges, so the monotonicity of Hermitian lower bound discussed in Theorem 5 (which is then extended to general random initialization) does not hold anymore. This is discussed in the Appendix K.1. For rank-deficient targets same problem also occurs, but we conjecture that one may tackle this through decomposing weight matrices into principle and complement space. We leave this as future work.
>
> We now have emphasized the discussion after Theorem 5 in the main paper, adding explicit reference to Appendix K.1, and have extended the explanation in the Appendix K.1 itself.

---

> ### Author Response · Authors · 2025-11-21
>
> **Q1 (Extension to deeper networks)**: Thank you for this important question. Our main convergence result cannot be directly extended to general deep even network, mainly because that the Hermitian main term (in Theorem 5) is hard to generalize to larger $N$. Specifically, the summation term of equation (147) in the Appendix E.2 contains more terms in the $N \ge 6$, and thus does not have monotonicity anymore.
>
> However, the analysis of Skew-Hermitian error term (Theorem 4) is more general. For balanced initialization, we have completed the proof for $2 \nmid N$ on complex domain (in the original paper we have only proved $2 \mid N$ for complex domain and general $N$ for real domain), now this Skew-Hermitian error term is proven to be **non-increasing for arbitrary deep linear networks $N \ge 2$ under balanced initialization for both real and complex domain**. For general random initialization (which does not have a decomposition $W = U\Sigma_w V^H$), the generalization of this term on even $N$ can be achieved by taking inverse (e.g. $W_1 + W_{2}^{-1} \cdots W_{N}^{-1} W_{N+1}^H \cdots W_{2N}^H$). For odd $N$ it still remains to discover an appropriate form to characterize this term.
>
>
> **Q2 (Motivation of Results on Complex Domain)**: Thank you for this crucial question. Typically neural networks are trained in real domain (while complex neural networks are also discussed in some areas). We consider the complex domain mainly because of the following two reasons:
>
> 1. The complex domain is a natural extension of real domain, at least in the field of matrix decomposition. Since most of the analysis is both applicable on both real and complex domain, we add this extension for completeness.
>
> 2. As discussed in Theorem 2, the convergence behaviour in the real domain suffers from a 1/2 in stricted balanced setting. This phenomenon, where **balancedness may hinder convergence** (as also mentioned in [2]), is directly caused by the minimum singular value of $(U+V)\Sigma_w$ having at least $1/2$ probability to be $0$ in $\mathbb{R}$. Crucially, this $1/2$ obstacle does not occur in the complex domain $\mathbb{C}$. This indicates that the complex domain **does not suffer from the drawbacks of exact balancedness** at least under our frmework of matrix factorization, and thus merits further theoretical investigation.
>
> We have emphasized this explanation in the final discussion part of the main paper and Appendix K.1.
>
> **Q3 (The reason of involving a 1/4 power term in the initialization scale and convergence rate)**: Thank you for raising this interesting question. This phenomena can be explained by scaling the target matrix. We can consider arbitrary $N$-layer matrix factorization instead of four-layer. By treating conditional number $\kappa = \sigma_1(\Sigma) / \sigma_d(\Sigma)$ as constant (or to say, relative size of target singular values is ixed), the initialization scale $\epsilon \propto \sigma_1^{1/N}(\Sigma)$, convergence time $T \propto \sigma_1^{-2(N-1)/N}(\Sigma)$. For $N=4$, $T \propto \sigma_1^{-3/2}(\Sigma)$; for $N=2$, $T \propto \sigma_1^{-1}(\Sigma)$.
> We also illustrate this through a **dimensional analysis** perspective, which we have specified in Remark 19.
>
> We have added a more detailed discussion in Appendix J.2 for your reference.
>
>
>
> Reference:
>
> [1] L. Jiang, Y. Chen, and L. Ding, "Algorithmic Regularization in Model-free Overparametrized Asymmetric Matrix Factorization", arXiv preprint arXiv:2203.02839, 2022.
>
> [2] N. Xiong, L. Ding, and S. S. Du, "How Over-Parameterization Slows Down Gradient Descent in Matrix Sensing: The Curses of Symmetry and Initialization", arXiv preprint arXiv:2310.01769, 2023.

---

### Author Response · Authors · 2025-12-03

**Rebuttal Summary for Area Chair**

We thank all reviewers for their valuable feedback and constructive suggestions. Below we summarize our main contributions and how the manuscript has been updated addressing the reviewers' concerns.

**Main Contributions**

Again we highlight our main contributions. Our work provides the first global convergence results for gradient descent on four-layer matrix factorization under random Gaussian initialization beyond the NTK regime, under target matrix with identical singular values, and standard balanced regularization term. While the main result is conducted under depth of four along with restrictions on target matrix, some of the intermediate lemmas are generalizable to arbitrary depth and target matrix.

**Paper Revisions**

Based on the reviewers' concerns, we have made the following revisions:

1. We have reorganized the manuscript, added roadmap sections in both the main paper and the appendix to enhance readability. We ensured all key concepts are properly explained and formally defined before their use. We have also provided expository illustrations and explanations of the main result, while moving supporting technical lemmas to the Appendix.

2. We adjusted some notations to avoid overloading, corrected typos, and clarified ambiguous statements, to address the concerns on ambiguity of notations and grammatical issues.

3. We have included additional examples and discussions in both the main paper and the Appendix to illustrate our convergence result and proof strategy, to address the lack of explanation in main results. Supplementary simulation results for more general settings beyond our theoretical scope have also been provided.

4. We have refined expressions in the abstract and introduction to ensure accuracy and avoid overstatement, regarding the concerns on overstated generality.

We believe these revisions present our results in a clear and well-organized manner, making the manuscript easy to follow with a clear proof outline.

Once again, we extend our sincere gratitude to all reviewers for their precious time and thoughtful comments, which have greatly helped us improve our work.

---

### Meta-Review · Area_Chair_bUir · 2025-12-28

**Summary:**

This paper shows that, in a highly specialized setting, a nonlinear objective arising from a four-layer linear network may admit global convergence with positive probability under gradient-based optimization.

Major concerns raised in the reviews relate to the presentation as well as the strong assumptions underlying the analysis, including unitary target matrices, a fixed layer number of four, constraints on the initialization, and the use of an additional balance-promoting regularizer. As clarified in the rebuttal, the authors confirm that these assumptions are intrinsic to their current proof framework and are not easily removable.

Following the reviewers’ suggestions, the authors made a substantial effort to improve the presentation in the revised manuscript. These revisions include adding roadmap sections, correcting several typos (although some issues remain, e.g., around lines 181 and 194), providing simple illustrative examples, and reorganizing the content.

Despite these limitations, the reviewers generally acknowledge that the optimization problem studied in the paper is highly nontrivial even under the above restrictive assumptions, and that establishing global convergence guarantees in this setting is technically challenging and of independent theoretical interest.

At the same time, concerns about clarity and accessibility remain. One reviewer, who provided post-rebuttal feedback, still finds the paper extremely difficult to follow and maintains a score of 2. Other reviewers, including those who assigned higher scores, had also noted varying degrees of difficulty in following the sequence of technical results in the main paper prior to the rebuttal, indicating that readability remains a nontrivial issue even when the technical contribution is viewed positively.

In summary, the paper addresses a difficult theoretical question and contains a meaningful result under a highly specialized set of assumptions. However, due to the limited scope and the remaining issues in clarity and accessibility, I do not believe the current version meets the ICLR standard for acceptance. I therefore recommend rejection.

**Reviewer Concerns:**

During the rebuttal, the authors clarified that the strong assumptions are intrinsic to their current framework, addressed the technical questions raised by the reviewers, and made a substantial effort to improve the presentation in the revised manuscript. However, the exposition remains challenging to follow for a broad ICLR audience.

**Reviewer Scores:**

Reviewer FFAL (score: 8) was positive about the paper, and given the already high initial score, a further increase is unlikely.
Reviewer dzdx (score: 6) and Reviewer TMKW (score: 6) are likely to maintain their scores, as their concerns regarding the limited scope and novelty of the work remain.
Reviewer QmG7 (score: 2) explicitly stated during the rebuttal that they would maintain their score, citing the severity of the presentation issues.

---

### Decision · Program_Chairs · 2026-01-26

Reject